# DOUBLE EQUIVARIANCE FOR INDUCTIVE LINK PREDICTION FOR BOTH NEW NODES AND NEW RELATION TYPES

## ABSTRACT

The task of inductive link prediction in knowledge graphs (KGs) generally focuses on test predictions with solely new nodes but not both new nodes and new relation types. In this work, we formally define the concept of *double permutation-equivariant representations* that are equivariant to permutations of both node identities and edge relation types. We then show how double-equivariant architectures are able to self-supervise pre-train on distinct KG domains and zero-shot predict links on a new KG domain (with completely new entities and new relation types). We also introduce the concept of *distributionally double equivariant positional embeddings* designed to perform the same task. Finally, we empirically demonstrate the capability of the proposed models against baselines on a set of novel real-world benchmarks. More interestingly, we show that self-supervised pre-training on more KG domains increases the zero-shot ability of our model to predict on new relation types over new entities on unseen KG domains.

## 1 INTRODUCTION

This work studies what we call a *doubly inductive* (node and relation) *link prediction* task to predict missing links in unseen knowledge graphs with completely new nodes and new relation types in test (i.e. none of them are seen in training). Doubly inductive link prediction can be seen as zero-shot meta-learning task, where training on knowledge graphs from domains A, B, and C allows us to zero-shot predict never-seen-before relations in a different domain D at test time without side information or fine-tuning. We note in passing that the outlined methodology could be applicable in other areas such as multilayer (Coscia et al., 2013) and heterogeneous (Chen et al., 2021a) network data.

*The main contribution of our work is a general theoretical framework for* **doubly inductive link prediction** *on knowledge graphs and a blueprint to create equivariant neural networks for this task (both from structural representations and from positional embeddings).* We will introduce the concept of double equivariant graph models and distributionally equivariant positional graph embedding models, which are equivariant to the overgroup of permutations of nodes and permutations of relations (we review the necessary group theory concepts in Section 2). The essence of double equivariance is to force the model to abstract and generalize across various domains (e.g., distinct knowledge graph domains such as Education, Health, Sports, Taxonomy, etc.), which traditional KG models are unable to do (Section 4 gives more details).

**Contributions.** This work makes the following *three* contributions:

1. Our work provides the first formal definition of the *doubly inductive link prediction task*, the concept of *double equivariance*, and that of *distributionally double equivariant positional embeddings* for knowledge graph models, whose node and pairwise representations are equivariant to the action of the permutation overgroup composed by the permutation subgroups of node identities, and edge types (relations).
2. Our work introduces ISDEA+, a fast and general *double equivariant* graph neural network model that is capable of performing *doubly inductive link prediction*. ISDEA+ is able to work on train and test knowledge graphs with disjoint sets of relations of distinct sizes. We also introduce an approximately double equivariant representation built from distributionally double equivariant positional embeddings.

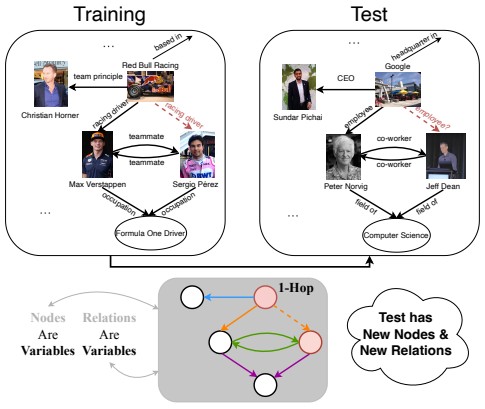

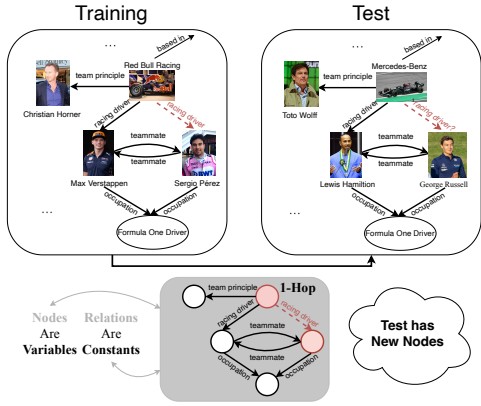

(a) **(Our task) Test over new nodes & relation types**

(b) **(Standard task) Test over new nodes only**

Figure 1: (a) **Doubly inductive link prediction:** This task learns (on training KGs from a set of domains) to inductively predict querying relation over a KG test over a different domain (with new nodes and new relation types). For instance, the local **relational structure** (gray rectangle) would be able to predict the same professional engagement relation types w.r.t. the structure of the training pair, as there is a common relational type structure (the colored links) in the training Sports KG that can be applied to the new nodes and new relation in the test Corporations KG. (b) **Traditional inductive link prediction:** The task predicts the same relation types seen in training (e.g., Sports) but over new nodes. The gray rectangle shows the relation types in the relational structure grounded in the relation types seen in training that could perform this task.

3. Our work introduces two novel tasks in two real-world benchmark datasets: PediaTypes and WikiTopics. These are pre-trained zero-shot meta-learning tasks, where the model is initially trained on a (diverse set of) KG(s) from distinct domains (e.g., Education, Health, Sports). The essence of this training is to imbue the model with the ability to abstract and generalize across domains. In test, the model is evaluated zero-shot on a KG of a completely new domain (e.g. Taxonomy). This setup challenges the model to apply its learned meta-knowledge to a new, unseen domain, demonstrating its capacity for cross-domain generalization and adaptation without prior exposure to the specific domain of the test KG. Our experiments show that ISDEA+ zero-shot performance increases with the number of self-supervised training KGs on diverse domains, to the point that it is sometimes better than *transductive training on the test KG domain itself* (e.g., the state-of-the-art baseline trains on Taxonomy KG to predict held out links on Taxonomy KG itself).

## 2 DOUBLY (NODE & RELATION) INDUCTIVE LINK PREDICTION

In what follows, we introduce the doubly inductive link prediction task and compare it with the traditional inductive link prediction task using two examples. We then proceed to theoretically describe the task in a general setting and propose our double equivariant modeling framework to handle doubly inductive link prediction task using structural representations and positional embeddings.

### 2.1 DOUBLY INDUCTIVE LINK PREDICTION EXAMPLES

We now introduce doubly inductive link prediction over both new nodes and new relation types and explain the difference between the traditional inductive link prediction task in Figure 1. The traditional inductive link prediction task focuses solely on predicting new nodes in the test. To this end, standard graph neural networks (GNNs) (Xu et al., 2019a; Morris et al., 2019) and their KG counterparts (e.g. (Hamilton et al., 2017; Galkin et al., 2021; Chen et al., 2022b; Zhu et al., 2021; Teru et al., 2020)) force the neural network to learn structural node representations (Srinivasan & Ribeiro, 2020), which —if used appropriately— allows GNNs for KGs to perform *inductive link prediction over new nodes*. As shown in Figure 1(b), these models are capable of learning the patterns in the gray box of Figure 1(b), but these patterns cannot extrapolate zero-shot to new relation types in test, because they rely on the relation type labels.

In other words, the equivariance in GNNs is not enough to perform the doubly inductive link prediction task in Figure 1(a). Specifically, to be able to *inductively* predict the "employee" relation on the test graph by learning from the "racing driver" relation on the training graph, the equivariance property

needs to go beyond just node permutations. To be able to represent the structural properties of the nodes and relations with respect to the structural properties of other nodes and relations, our work defines an equivariance also in relations. For instance, via double equivariance (we will define the concept in Section 2.3) it is possible to perform the task of predicting "employee" using the node and relation structural pattern shown at the bottom of Figure 1(a) in the gray box. Additional examples and a more detailed analysis of a family of logical statements implied by double equivariance can be found in Appendices A and B.

## 2.2 FORMALIZING THE DOUBLY INDUCTIVE LINK PREDICTION TASK

We now introduce notations and definitions used throughout this paper. First, we formally define our inductive link prediction task for both new nodes and new relation types, i.e., doubly inductive link prediction, over knowledge graphs. We denote $[n] := \{1, \ldots, n\}$ for any $n \in \mathbb{N}$. Let $\mathcal{G}^{(\text{tr})} = (\mathcal{V}^{(\text{tr})}, \mathcal{R}^{(\text{tr})}, \mathbf{A}^{(\text{tr})})$ be the training knowledge graph, where $\mathcal{V}^{(\text{tr})}$ is the set of $N^{(\text{tr})}$ training nodes, $\mathcal{R}^{(\text{tr})}$ is the set of $R^{(\text{tr})}$ training relation types. We also define two associated bijective mappings $v_.^{(\text{tr})} : [N^{(\text{tr})}] \to \mathcal{V}^{(\text{tr})}, r_.^{(\text{tr})} : [R^{(\text{tr})}] \to \mathcal{R}^{(\text{tr})}$ that enumerate the nodes and relation types in training. The tensor $\mathbf{A}^{(\text{tr})} \in \{0,1\}^{N^{(\text{tr})} \times R^{(\text{tr})} \times N^{(\text{tr})}}$ defines the adjacency of the training graph such that $\forall (i,k,j) \in [N^{(\text{tr})}] \times [R^{(\text{tr})}] \times [N^{(\text{tr})}], \mathbf{A}_{i,k,j}^{(\text{tr})} = 1$ indicates that the triplet $(v_i^{(\text{tr})}, r_k^{(\text{tr})}, v_j^{(\text{tr})})$ is present in the data (we denote $(i,k,i)$ as the $k$-th attribute of node $i$). To simplify notation, we further refer to the collection of all knowledge graphs of any sizes as $\mathbb{A} := \cup_{N=1}^{\infty} \cup_{R=2}^{\infty} \{0,1\}^{N \times R \times N}$.

**Definition 2.1** (Doubly inductive link prediction task). The task of doubly inductive link prediction learns a model on a set of $K$ training graphs $\{\mathcal{G}_1^{(\text{tr})}, \ldots, \mathcal{G}_K^{(\text{tr})}\}$ (often in a self-supervised fashion by masking links) and inductively applies it to predict missing links in a test graph $\mathcal{G}^{(\text{te})} = (\mathcal{V}^{(\text{te})}, \mathcal{R}^{(\text{te})}, \mathbf{A}^{(\text{te})})$ with *completely* new nodes and new relation types, i.e., $\mathcal{V}^{(\text{te})} \cap \mathcal{V}^{(\text{tr})} = \emptyset, \mathcal{R}^{(\text{te})} \cap \mathcal{R}^{(\text{tr})} = \emptyset$, without extra context given to the model.

While some real-world tasks may have overlapping relation types between training and test, Definition 2.1 forces the model to not rely on potential overlaps. In what follows, we use the superscript $(*)$ as a wildcard to describe both train and test data. For example, $\mathbf{A}^{(*)}$ is a wildcard variable for referring to either $\mathbf{A}^{(\text{tr})}$ or $\mathbf{A}^{(\text{te})}$. And since there are bijections $v_.^{(*)}, r_.^{(*)}$ between indices and nodes and relation types, we represent the triplet $(v_i^{(*)}, r_k^{(*)}, v_j^{(*)}) \in \mathcal{V}^{(*)} \times \mathcal{R}^{(*)} \times \mathcal{V}^{(*)}$ with indices $(i,k,j) \in [N^{(*)}] \times [R^{(*)}] \times [N^{(*)}]$, and mainly use $\mathbf{A}^{(*)}$ to denote the knowledge graph.

Without additional context such as textural description embeddings for the new relations or graph ontology (thoroughly discussed in Section 4), it is essential for our model to differentiate nodes and relations based only on their structural relationships in $\mathbf{A}^{(*)}$, rather than their labels in $\mathcal{V}^{(*)}, \mathcal{R}^{(*)}$, in order to make accurate predictions in doubly inductive link prediction as discussed in Section 2.1. Thus, we develop the double equivariant representations for knowledge graphs as follows.

## 2.3 DOUBLE EQUIVARIANT REPRESENTATIONS FOR KNOWLEDGE GRAPHS

In what follows, we provide definitions and theoretical statements of our proposed double equivariant knowledge graph representations in the main paper while referring all proofs to Appendix D. The proposal starts with defining the permutation actions on knowledge graphs as:

**Definition 2.2** (Node and relation permutation actions on knowledge graphs). For any knowledge graph $\mathbf{A}^{(*)} \in \mathbb{A}$ with number of nodes and relations $N^{(*)}, R^{(*)}$, a node permutation $\phi \in \mathbb{S}_{N^{(*)}}$ is an element of the symmetric group $\mathbb{S}_{N^{(*)}}$, a relation permutation $\tau \in \mathbb{S}_{R^{(*)}}$ is an element of the symmetric group $\mathbb{S}_{R^{(*)}}$, and the operation $\phi \circ \tau \circ \mathbf{A}^{(*)}$ is the action of $\phi$ and $\tau$ on $\mathbf{A}^{(*)}$, defined as $\forall (i,k,j) \in [N^{(*)}] \times [R^{(*)}] \times [N^{(*)}], (\phi \circ \tau \circ \mathbf{A}^{(*)})_{\phi \circ i, \tau \circ k, \phi \circ j} = \mathbf{A}_{i,k,j}^{(*)}$ where $\phi \circ i = \phi_i$ and $\tau \circ k = \tau_k$. The node and relation permutation actions on $\mathbf{A}^{(*)}$ are commutative, i.e., $\phi \circ \tau \circ \mathbf{A}^{(*)} = \tau \circ \phi \circ \mathbf{A}^{(*)}$.

To learn structural representation for both nodes and relations, we first design triplet representations that are invariant to the two permutation actions on nodes and relations, as shown below.

**Definition 2.3** (Double invariant triplet representations). For any knowledge graph $\mathbf{A}^{(*)} \in \mathbb{A}$ with number of nodes and relations $N^{(*)}, R^{(*)}$, a double invariant triplet representation is a function $\Gamma_{\text{tri}} : \cup_{N=1}^{\infty} \cup_{R=2}^{\infty} ([N] \times [R] \times [N]) \times \mathbb{A} \to \mathbb{R}^d, d \geq 1$, such that $\forall (i,k,j) \in [N^{(*)}] \times [R^{(*)}] \times [N^{(*)}], \forall \phi \in \mathbb{S}_{N^{(*)}}, \forall \tau \in \mathbb{S}_{R^{(*)}}, \Gamma_{\text{tri}}((i,k,j), \mathbf{A}^{(*)}) = \Gamma_{\text{tri}}((\phi \circ i, \tau \circ k, \phi \circ j), \phi \circ \tau \circ \mathbf{A}^{(*)})$.

To understand the property of our double invariant triplet representations, we first introduce the notion of knowledge graph isomorphism and triplet double isomorphism.

**Definition 2.4** (Knowledge graph isomorphism and Triplet isomorphism). We say two knowledge graphs $\mathbf{A}^{(G)}, \mathbf{A}^{(H)} \in \mathbb{A}$ with number of nodes and relations $N^{(G)}, R^{(G)}$ and $N^{(H)}, R^{(H)}$ respectively, are isomorphic (denoted as "$\mathbf{A}^{(G)} \simeq_{\mathrm{RL}} \mathbf{A}^{(H)}$") if and only if $\exists \phi \in \mathbb{S}_{N^{(G)}}, \exists \tau \in \mathbb{S}_{R^{(G)}}$, such that $\phi \circ \tau \circ \mathbf{A}^{(G)} = \mathbf{A}^{(H)}$. And we say two triplets $\left(i^{(G)}, k^{(G)}, j^{(G)}\right) \in [N^{(G)}] \times [R^{(G)}] \times [N^{(G)}]$, $\left(i^{(H)}, k^{(H)}, j^{(H)}\right) \in [N^{(H)}] \times [R^{(H)}] \times [N^{(H)}]$ are isomorphic triplets (denoted as "$\left(\left(i^{(G)}, k^{(G)}, j^{(G)}\right), \mathbf{A}^{(G)}\right) \simeq_{\mathrm{TRI}} \left(\left(i^{(H)}, k^{(H)}, j^{(H)}\right), \mathbf{A}^{(H)}\right)$") if and only if $\exists \phi \in \mathbb{S}_{N^{(G)}}, \exists \tau \in \mathbb{S}_{R^{(G)}}$, such that $\phi \circ \tau \circ \mathbf{A}^{(G)} = \mathbf{A}^{(H)}$ and $\left(i^{(H)}, k^{(H)}, j^{(H)}\right) = \left(\phi \circ i^{(G)}, \tau \circ k^{(G)}, \phi \circ j^{(G)}\right)$.

For example, in Figure 1(a), the training triplet (Red Bull Racing, racing driver, Sergio Pérez) and the test triplet (Google, employee, Jeff Dean) are isomorphic triplets by Definition 2.4. It is clear that our double invariant triplet representations are able to output the same representations for these isomorphic triplets, enabling doubly inductive link prediction, where the model trained to predict the missing (Red Bull Racing, racing driver, Sergio Pérez) in the training graph is able to predict the missing (Google, employee, Jeff Dean) in test. The connection between Definition 2.3 and logical reasoning can be found in Appendix B. In what follows, we define the structure double equivariant representations for the whole knowledge graph $\mathbf{A}^{(*)}$ (akin to how GNNs provide representations for a whole graph).

**Definition 2.5** (Double equivariant knowledge graph representations). For any knowledge graph $\mathbf{A}^{(*)} \in \mathbb{A}$ with number of nodes and relations $N^{(*)}, R^{(*)}$, a function $\Gamma_{\mathrm{gra}} : \mathbb{A} \to \cup_{N=1}^{\infty} \cup_{R=2}^{\infty} \mathbb{R}^{N \times R \times N \times d}, d \geq 1$ is double equivariant w.r.t. arbitrary node $\phi \in \mathbb{S}_{N^{(*)}}$ and relation $\tau \in \mathbb{S}_{R^{(*)}}$ permutations, if $\Gamma_{\mathrm{gra}}(\phi \circ \tau \circ \mathbf{A}^{(*)}) = \phi \circ \tau \circ \Gamma_{\mathrm{gra}}(\mathbf{A}^{(*)})$. Moreover, valid mappings of $\Gamma_{\mathrm{gra}}$ must map a domain element to an image element with the same number of nodes and relations.

Finally, we connect Definitions 2.3 and 2.5 by showing how to build double equivariant graph representations from double invariant triplet representations in Theorem 2.6, and vice-versa.

**Theorem 2.6.** *For all $\mathbf{A}^{(*)} \in \mathbb{A}$ with number of nodes and relations $N^{(*)}, R^{(*)}$, given a double invariant triplet representation $\Gamma_{tri}$, we can construct a double equivariant graph representation as $\left(\Gamma_{gra}(\mathbf{A}^{(*)})\right)_{i,k,j} := \Gamma_{tri}((i, k, j), \mathbf{A}^{(*)}), \forall (i, k, j) \in [N^{(*)}] \times [R^{(*)}] \times [N^{(*)}]$, and vice-versa.*

Next, we consider positional graph embeddings that are equivariant in distribution.

## 2.4 DISTRIBUTIONALLY DOUBLE EQUIVARIANT POSITIONAL GRAPH EMBEDDINGS

To the best of our knowledge, InGram (Lee et al., 2023) is the first and only existing work capable of performing our doubly inductive link prediction task (Definition 2.1), but it does so with what we now define as *distributionally double equivariant positional embeddings*, which are permutation sensitive, as we will show in Section 3.2:

**Definition 2.7** (Distributionally double equivariant positional embeddings). For any knowledge graph $\mathbf{A}^{(*)} \in \mathbb{A}$ with number of nodes and relations $N^{(*)}, R^{(*)}$, the distributionally double equivariant positional embeddings of $\mathbf{A}^{(*)}$ are defined as joint samples of random variables $\mathbf{Z}|\mathbf{A}^{(*)} \sim p(\mathbf{Z}|\mathbf{A}^{(*)})$, where the tensor $\mathbf{Z}$ is defined as $\mathbf{Z}_{i,k,j} \in \mathbb{R}^d, d \geq 1, \forall (i, k, j) \in [N^{(*)}] \times [R^{(*)}] \times [N^{(*)}]$, where we say $p(\mathbf{Z}|\mathbf{A}^{(*)})$ is a double equivariant probability distribution on $\mathbf{A}^{(*)}$ defined as $\forall \phi \in \mathbb{S}_{N^{(*)}}, \forall \tau \in \mathbb{S}_{R^{(*)}}, p(\mathbf{Z}|\mathbf{A}^{(*)}) = p(\phi \circ \tau \circ \mathbf{Z}|\phi \circ \tau \circ \mathbf{A}^{(*)})$.

Prior work on (standard) link prediction tasks has shown the advantages of equivariant representations over positional embeddings (Zhang et al., 2021). Moreover, Srinivasan & Ribeiro (2020) establishes the equivalence between positional embeddings and structural representations for simple graphs by proving that representations based on an expectation of the positional embeddings are equivariant to node permutations. In what follows, we extend this result to the double equivariant setting:

**Theorem 2.8** (From distributional double equivariant positional embeddings to double equivariant representations). *For any knowledge graph $\mathbf{A}^{(*)} \in \mathbb{A}$, the average $\mathbb{E}_{p(\mathbf{Z}|\mathbf{A}^{(*)})}[\mathbf{Z}|\mathbf{A}^{(*)}]$ is a double equivariant knowledge graph representation (Definition 2.5) for any distributional double equivariant positional embeddings $\mathbf{Z}|\mathbf{A}^{(*)}$ (Definition 2.7).*

Later in Section 3.2, we use the result in Theorem 2.8 to introduce DEq-InGram, a double equivariant representation that builds upon InGram's distributionally double equivariant positional embeddings (Definition 2.7) that is shown to significantly outperforms the original InGram in Section 5.

# 3 DOUBLE EQUIVARIANT NEURAL ARCHITECTURE

This section introduces two double equivariant neural architectures based on Sections 2.3 and 2.4. First, Section 3.1 introduces an Inductive Structural Double Equivariant Architecture Plus (ISDEA+), a model guaranteed to produce double equivariant representations (Definition 2.5). Then, Section 3.2 introduces a Monte Carlo estimate of a double equivariant representation built from a distributionally double equivariant positional graph embedding (Lee et al., 2023).

## 3.1 INDUCTIVE STRUCTURAL DOUBLE EQUIVARIANT ARCHITECTURE PLUS (ISDEA+)

We start revisiting Definition 2.4. Consider an arbitrary knowledge graph $\mathbf{A}^{(*)} \in \mathbb{A}$ with number of nodes and relations $N^{(*)}, R^{(*)}$, and denote $A^{(*,k)}$ as the adjacency matrix such that $A_{i,j}^{(*,k)} := \mathbf{A}_{i,k,j}^{(*)}$, $\forall (i, k, j) \in [N^{(*)}] \times [R^{(*)}] \times [N^{(*)}]$. For each adjacency matrix $A^{(*,k)}$, it will correspond to a graph without edge relation types, thus we can also consider $A^{(*,k)}$ as an unattributed graph containing only edges with relation type $r_k^{(*)}$. Then, the knowledge graph $\mathbf{A}^{(*)}$ can be equivalently expressed as a collection of unattributed graphs $A^{(*)} := \{\!\!\{ A^{(*,1)}, \ldots, A^{(*,R^{(*)})} \}\!\!\}$. Since the actions of the two permutation groups $\mathbb{S}_{N^{(*)}}$ and $\mathbb{S}_{R^{(*)}}$ commute, the double equivariance of $\mathbf{A}^{(*)}$ (Definition 2.4) can be described as two (single) equivariances: A (graph) equivariance $\phi \in \mathbb{S}_{N^{(*)}}$ over each graph $A^{(*,k)}, k = 1, \ldots, R^{(*)}$, and a (set) equivariance $\tau \in \mathbb{S}_{R^{(*)}}$ (over the set of graphs). Hence, our double equivariance can make use of the general framework using DSS layers on learning sets of symmetric elements proposed by Maron et al. (2020). We first define a double equivariant layer composed by a Siamese layer (Bromley et al., 1993) as follows, $L : \mathbb{A} \to \cup_{N=1}^{\infty} \cup_{R=2}^{\infty} \mathbb{R}^{N \times R \times N \times d}$, for each $k = 1, ..., R^{(*)}$:

$$\left( L\left(\mathbf{A}^{(*)}\right) \right)_{:,k} = L_1\left(A^{(*,k)}\right) + L_2\left(\text{AGG}_{k' \neq k}^{R^{(*)}}\left(A^{(*,k')}\right)\right), \tag{1}$$

where $d$ is the output dimension, $L_1, L_2 : \mathbb{A} \to \cup_{N=1}^{\infty} \mathbb{R}^{N \times N \times d}$ can be any (node) equivariant layers that output pairwise representations (Zhang & Chen, 2018; Zhu et al., 2021; Zhang et al., 2021), and the aggregation term $\text{AGG}_{k' \neq k}^{R^{(*)}}$ can be any set aggregators such as sum, mean, max, DeepSets (Zaheer et al., 2017), etc.. Note that the proposed layer is similar to the $H$-equivariant layer proposed by Bevilacqua et al. (2021) for increasing the expressiveness of GNN using sets of subgraphs (a markedly different task than ours). We create our double equivariant model with double equivariant layers.

### 3.1.1 IMPLEMENTATION DETAILS

We use GNN layers for constructing $L_1, L_2$. Since most-expressive pairwise representations are computationally expensive, we propose Inductive Structural Double Equivariant Architecture Plus (ISDEA+) and trade-off expressivity in the implementation of Equation (1) for speed and memory by using node representation GNN layers (Xu et al., 2019a; Veličković et al., 2017; Morris et al., 2019). Specifically, for a knowledge graph $\mathbf{A}^{(*)}$ with number of nodes and relations $N^{(*)}, R^{(*)}$, at each iteration $t = 1, ..., T$, for each relation type $k \in [R^{(*)}]$, all nodes $i \in [N^{(*)}]$ are associated with two learned vectors $h_{i,k}^{(t)} \in \mathbb{R}^{d_t}, h_{i,\neg k}^{(t)} \in \mathbb{R}^{d_t}, d_t \geq 1$. If there are no node attributes, we initialize $\forall k \in [R^{(*)}], h_{i,k}^{(0)} = h_{i,\neg k}^{(0)} = \mathbb{1}$. Then we recursively compute the update, $\forall i \in [N^{(*)}], \forall k \in [R^{(*)}]$,

$$h_{i,k}^{(t+1)} = \text{GNN}_1^{(t)}\left(h_{i,k}^{(t)}, \{\!\!\{ h_{j,k}^{(t)} \big| j \in \mathcal{N}_k(i) \}\!\!\}\right), \qquad h_{i,\neg k}^{(t+1)} = \text{GNN}_2^{(t)}\left(h_{i,\neg k}^{(t)}, \{\!\!\{ h_{j,\neg k}^{(t)} \big| j \in \bigcup_{k' \neq k} \mathcal{N}_k(i) \}\!\!\}\right), \quad \text{if } t = 0,$$

$$h_{i,k}^{(t+1)} = \text{GNN}_1^{(t)}\left(h_{i,k}^{(t)}, \{\!\!\{ h_{j,k}^{(t)} \big| j \in \bigcup_{k' \in [R^{(*)}]} \mathcal{N}_{k'}(i) \}\!\!\}\right), h_{i,\neg k}^{(t+1)} = \text{GNN}_2^{(t)}\left(h_{i,\neg k}^{(t)}, \{\!\!\{ h_{j,\neg k}^{(t)} \big| j \in \bigcup_{k' \in [R^{(*)}]} \mathcal{N}_{k'}(i) \}\!\!\}\right), \text{if } t > 0,$$

where $\text{GNN}_1^{(t)}$ and $\text{GNN}_2^{(t)}$ denote two GNN layers and $\mathcal{N}_k(i) := \left\{ j \big| \mathbf{A}_{j,k,i}^{(*)} = 1 \right\}$ denotes the neighborhood set of node $i$ with relation $k$ in the unattributed graph $A^{(*,k)}$. To get the final representation $X_{i,k}$ for the node $i$ with respect to relation $k$. We define $h_{i,k} = h_{i,k}^{(0)} \big\| h_{i,k}^{(1)} \big\| \cdots \big\| h_{i,k}^{(T)}$,

$h_{i,\neg k} = h_{i,\neg k}^{(0)} \big\| h_{i,\neg k}^{(1)} \big\| \cdots \big\| h_{i,\neg k}^{(T)}$, and combine the two embeddings as illustrated in Equation (1),

$$X_{i,k} = \mathrm{MLP}_1(h_{i,k}) + \mathrm{MLP}_2(h_{i,\neg k}), \forall i \in [N^{(*)}], \forall k \in [R^{(*)}], \tag{2}$$

where $\mathrm{MLP}_1, \mathrm{MLP}_2$ are two multi-layer perceptrons, $\|$ as the concatenation operation.

As shown by Srinivasan & Ribeiro (2020); You et al. (2019), structural node representations are not most expressive for link prediction in unattributed graphs. Hence, we concatenate $i$ and $j$ (double equivariant) node representations with the shortest distance between $i$ and $j$ in the observed graph as our triplet representations (appending distances is also adopted in the representations of prior work (Teru et al., 2020; Galkin et al., 2021)). Finally, we obtain the triplet representation,

$$\Gamma_{\mathrm{ISDEA+}}((i,k,j), \mathbf{A}^{(*)}) = \left( X_{i,k} \big\| X_{j,k} \big\| d(i,j) \big\| d(j,i) \right), \forall (i,k,j) \in [N^{(*)}] \times [R^{(*)}] \times [N^{(*)}], \tag{3}$$

where we denote $d(i,j)$ as the length of shortest path from $i$ to $j$ without considering $(i,k,j)$. Since our graph is directed, we concatenate them in both directions. For more implementation details and complexity analysis of ISDEA+, please refer to Appendix C.

**Lemma 3.1.** $\Gamma_{ISDEA+}$ *in Equation* (3) *is a double invariant triplet representation as per Definition 2.3.*

As in Yang et al. (2015); Schlichtkrull et al. (2018); Zhu et al. (2021), we use negative sampling in our training with the difference that we account for both predicting missing nodes and relation types (Definition 2.1). Specifically, for each positive training triplet $(i,k,j)$ such that $\mathbf{A}_{i,k,j}^{(\mathrm{tr})} = 1$, we first randomly corrupt either the head or the tail $n_{\mathrm{nd}}$ times to generate the negative (node) examples $(i,k,j')$. Additionally, we also want our model to learn the correct relation type $(i,?,j)$ between a pair of nodes. Thus, we corrupt relation $n_{\mathrm{rl}}$ times to generate negative (relation) examples $(i,k',j)$. In our training, $n_{\mathrm{nd}} = n_{\mathrm{rl}} = 2$; while in evaluation, $n_{\mathrm{nd}} = 50, n_{\mathrm{rl}} = 0$ for node evaluation, and $n_{\mathrm{nd}} = 0, n_{\mathrm{rl}} = 50$ for relation evaluation. Following Schlichtkrull et al. (2018), we use cross-entropy loss to encourage the model to score positive examples higher than corresponding negative examples:

$$\mathcal{L} = - \sum_{(i,k,j) \in \mathcal{S}} \left( \log \left( \Gamma_{\mathrm{tri}}((i,k,j), \mathbf{A}^{(\mathrm{tr})}) \right) - \frac{1}{n_{\mathrm{nd}} + n_{\mathrm{rl}}} \sum_{p=1}^{n_{\mathrm{nd}} + n_{\mathrm{rl}}} \log \left( 1 - \Gamma_{\mathrm{tri}} \left( (i_p', k_p', j_p'), \mathbf{A}^{(\mathrm{tr})} \right) \right) \right), \tag{4}$$

where $S = \left\{ (i,k,j) \big| \mathbf{A}_{i,k,j}^{(\mathrm{tr})} = 1 \right\}$, and $(i_p', k_p', j_p')$ are the $p$-th negative node or relation example corresponding to $(i,k,j)$.

## 3.2 Double Equivariant InGram (DEq-InGram)

ISDEA+ directly obtains double equivariant representations for knowledge graphs. Alternatively, one can build these double equivariant representations from distributionally double equivariant positional embeddings (Theorem 2.8). To this end, we investigate obtaining double equivariant representations from the positional embeddings of InGram (Lee et al., 2023), as discussed in Section 2.4.

InGram (Lee et al., 2023) constructs a *relation graph* as a weighted graph consisting of relations and a heuristic to construct affinity weights between them. It then employs a GNN on the relation graph to generate relation embeddings, which are then fed into another GNN on the original knowledge graph to generate node embeddings. Finally, InGram uses a variant of DistMult (Yang et al., 2015) to compute triplet scores from the node and relation embeddings. These embeddings, however, are permutation sensitive due to their reliances on Glorot initialization (Glorot & Bengio, 2010) in each training epoch and test-time inference.

**Lemma 3.2.** *The triplet representations generated by InGram (Lee et al., 2023) output distributionally double equivariant positional embeddings (Definition 2.7).*

Theorem 2.8 suggests that averaging InGram's positional embeddings can be used to construct double equivariant knowledge graph representations. Hence, we propose a Monte Carlo method to estimate these double equivariant graph representations and denote it as DEq-InGram. Specifically, given InGram's triplet score function $\mathbf{Z}_{\mathrm{InGram}}((i,k,j), \mathbf{A}^{(\mathrm{te})}, \boldsymbol{V}^{(0)}, \boldsymbol{R}^{(0)})$ over a test knowledge graph $\mathbf{A}^{(\mathrm{te})}$, the initial random node embeddings $\boldsymbol{V}^{(0)} \in \mathbb{R}^{N^{(\mathrm{te})} \times d}$, and the initial random relation embeddings $\boldsymbol{R}^{(0)} \in \mathbb{R}^{R^{(\mathrm{te})} \times d'}$ (where $d$ and $d'$ are the dimension sizes), our DEq-InGram produces the following

triplet scores:

$$\Gamma_{\text{DEq-InGram}}((i, k, j), \mathbf{A}^{(\text{te})}) = \frac{1}{M} \sum_{m=1}^{M} \mathbf{Z}_{\text{InGram}}((i, k, j), \mathbf{A}^{(\text{te})}, \boldsymbol{V}_m^{(0)}, \boldsymbol{R}_m^{(0)}) \quad (5)$$

where $\{\boldsymbol{V}_m^{(0)}\}_{m=1}^{M}$ and $\{\boldsymbol{R}_m^{(0)}\}_{m=1}^{M}$ are $M$ i.i.d. samples drawn from the distribution of initial node and initial relation embeddings respectively (via Glorot initialization).

## 4 RELATED WORK

A more comprehensive discussion of related work can be found in Appendix E.

**Transductive link prediction.** In transductive link prediction task, missing links are predicted over a fixed set of nodes and relation types as in training (Bordes et al., 2013; Yang et al., 2015; Trouillon et al., 2016). These (positional) embeddings can be made inductive via Srinivasan & Ribeiro (2020)'s theory but are not designed for predicting new relation types.

**Inductive link prediction over new nodes (but not new relations).** Rule-induction methods (Yang et al., 2015; 2017; Meilicke et al., 2018; Sadeghian et al., 2019) are inherently node-independent which aim to extract First-order Logical Horn clauses from the attributed multigraph. Recently, with the advancement of GNNs, various works (Schlichtkrull et al., 2018; Teru et al., 2020; Galkin et al., 2021; Zhu et al., 2021; Chen et al., 2022b) have applied the idea of GNN in relational prediction to learn structural node/pairwise representation. Although all these methods can be used to perform *inductive link prediction over solely new nodes*, they can not handle new relation types in test.

**Inductive link prediction over both new nodes and new relations (with extra context).** Existing methods for querying triplets involving both new nodes and new relations generally assume access to extra context, such as generating language embedding for textual descriptions of unseen relation types (Qin et al., 2020; Geng et al., 2021; Zha et al., 2022; Wang et al., 2021a), a shared background graph connecting seen and unseen relations (e.g., test graph has training relations (Huang et al., 2022; Chen et al., 2021b; 2022a)), or access to graph ontology (Geng et al., 2023). Hence, these methods cannot be directly applied to test graphs that neither contain meaningful descriptive information of the unseen relation types (e.g., url links) nor connection with nodes and relation types seen in training.

**Inductive link prediction over both new nodes and new relations (no extra context).** We focus on this most general doubly inductive link prediction task without additional context data (just the test graph structure is available during inference). To the best of our knowledge, InGram (Lee et al., 2023) is the first and only existing method capable of performing this task. The connection between InGram and our work has been described in Sections 2.4 and 3.2.

## 5 EXPERIMENTAL RESULTS

In this section, we aim to answer two questions: **Q1**: Can double equivariant models (ISDEA+ and DEq-InGram) perform doubly inductive link prediction over knowledge graphs more accurately (and faster) than InGram (Lee et al., 2023)? **Q2**: We introduce a self-supervised zero-shot meta-learning task, where we pre-train (in a self-supervised fashion) on an increasing number of KGs from different domains (e.g., Education, Health, Sports) and then ask the model to zero-shot predict links on a test KG from an unseen domain (e.g. Taxonomy) with *completely new entities and relationship types*. Will the performance of ISDEA+ improve as the number of pre-training KG domains increase?

**Datasets.** To the best of our knowledge, there are no existing real-world benchmarks that are specially designed to test a model's extrapolation capability for doubly inductive link prediction task with training and test graphs coming from distinct domains with distinct characteristics. Existing datasets such as NL-100, WK-100, and FB-100 from Lee et al. (2023) are typically created by randomly splitting a larger graph (e.g., NELL-995 (Xiong et al., 2018), Wikidata68K (Gesese et al., 2022), FB15K237 (Toutanova & Chen, 2015)) into disjoint node and relation sets, implying that the test and training graphs still come from the same distribution. In contrast, we purposefully create two doubly inductive link prediction benchmark datasets: PediaTypes and WikiTopics, sampled respectively from the OpenEA library (Sun et al., 2020b) and WikiData-5M (Wang et al., 2021b), where by design, the test and training graphs are either from different domains, and are likely to possess different characteristics to fully test model's capability for doubly inductive link prediction.

Table 1: **Relation & Node Hits@10 performance on Doubly Inductive Link Prediction over PediaTypes.** We report standard deviations over 5 runs. A higher value means better doubly inductive link prediction performance. The dataset name "$X$-$Y$" means training on graph $X$ and testing on graph $Y$. The best values are shown in bold font, while the second-best values are underlined. The highest standard deviation within each task is highlighted in red color. "Rand" column contains unbiased estimations of the performance from a random predictor. **Both ISDEA and DEq-InGram consistently achieve better results than the baselines with generally smaller standard deviations.** N/A*: Not available due to constant crashes.

(a) **Relation prediction** $(i, ?, j)$ **performance in %. Higher ↑ is better.**

| Models | EN-FR | FR-EN | EN-DE | DE-EN | DB-WD | WD-DB | DB-YG | YG-DB |
|---|---|---|---|---|---|---|---|---|
| Rand | $19.60_{\pm00.00}$ | $19.60_{\pm00.00}$ | $19.60_{\pm00.00}$ | $19.60_{\pm00.00}$ | $19.60_{\pm00.00}$ | $19.60_{\pm00.00}$ | $19.60_{\pm00.00}$ | $19.60_{\pm00.00}$ |
| GAT | $18.58_{\pm00.52}$ | $18.93_{\pm00.33}$ | $19.40_{\pm00.28}$ | $18.87_{\pm00.19}$ | $18.78_{\pm00.28}$ | $18.76_{\pm00.33}$ | $19.78_{\pm01.39}$ | $19.15_{\pm00.35}$ |
| GIN | $19.34_{\pm00.32}$ | $19.34_{\pm00.29}$ | $18.98_{\pm00.27}$ | $18.88_{\pm00.47}$ | $19.30_{\pm00.52}$ | $18.86_{\pm00.35}$ | $18.69_{\pm00.75}$ | $18.92_{\pm00.68}$ |
| GraphConv | $19.18_{\pm00.27}$ | $19.02_{\pm00.64}$ | $19.19_{\pm00.24}$ | $18.93_{\pm00.60}$ | $19.46_{\pm00.38}$ | $19.13_{\pm00.54}$ | $19.13_{\pm01.24}$ | $18.89_{\pm00.57}$ |
| NBFNet | $21.93_{\pm02.53}$ | $22.20_{\pm02.92}$ | $18.98_{\pm02.75}$ | $7.01_{\pm01.43}$ | $23.51_{\pm07.06}$ | $23.05_{\pm03.55}$ | $31.50_{\pm04.82}$ | $35.17_{\pm05.13}$ |
| RMPI | $27.91_{\pm07.48}$ | $28.62_{\pm03.75}$ | $27.51_{\pm06.48}$ | $25.59_{\pm06.48}$ | N/A* | $16.76_{\pm04.03}$ | $39.03_{\pm20.28}$ | $11.77_{\pm07.07}$ |
| InGram | $78.74_{\pm07.48}$ | $62.11_{\pm13.60}$ | $48.72_{\pm08.94}$ | $65.60_{\pm14.42}$ | $77.75_{\pm06.60}$ | $63.32_{\pm02.78}$ | $67.98_{\pm25.45}$ | $64.98_{\pm26.69}$ |
| DEq-InGram (Ours) | $\underline{87.94}_{\pm05.68}$ | $\underline{80.47}_{\pm09.90}$ | $\underline{68.89}_{\pm05.45}$ | $\underline{80.79}_{\pm10.51}$ | $\underline{91.47}_{\pm01.53}$ | $\underline{77.03}_{\pm04.09}$ | $\underline{77.72}_{\pm21.92}$ | $\underline{89.30}_{\pm05.53}$ |
| ISDEA+ (Ours) | $\mathbf{99.12}_{\pm00.24}$ | $\mathbf{98.84}_{\pm00.06}$ | $\mathbf{99.20}_{\pm00.13}$ | $\mathbf{98.99}_{\pm00.12}$ | $\mathbf{98.56}_{\pm00.12}$ | $\mathbf{98.03}_{\pm00.17}$ | $\mathbf{88.78}_{\pm03.23}$ | $\mathbf{96.45}_{\pm00.24}$ |

(b) **Node prediction** $(i, k, ?)$ **performance in %. Higher ↑ is better.**

| Models | EN-FR | FR-EN | EN-DE | DE-EN | DB-WD | WD-DB | DB-YG | YG-DB |
|---|---|---|---|---|---|---|---|---|
| Rand | $19.60_{\pm00.00}$ | $19.60_{\pm00.00}$ | $19.60_{\pm00.00}$ | $19.60_{\pm00.00}$ | $19.60_{\pm00.00}$ | $19.60_{\pm00.00}$ | $19.60_{\pm00.00}$ | $19.60_{\pm00.00}$ |
| GAT | $89.77_{\pm00.41}$ | $86.83_{\pm00.41}$ | $66.24_{\pm02.81}$ | $69.08_{\pm00.66}$ | $31.08_{\pm01.07}$ | $77.05_{\pm00.36}$ | $53.51_{\pm00.29}$ | $64.13_{\pm00.31}$ |
| GIN | $90.10_{\pm00.61}$ | $85.32_{\pm01.18}$ | $73.32_{\pm03.35}$ | $75.66_{\pm04.85}$ | $34.87_{\pm09.12}$ | $78.67_{\pm02.46}$ | $56.87_{\pm00.44}$ | $65.27_{\pm01.14}$ |
| GraphConv | $92.97_{\pm00.11}$ | $\mathbf{90.56}_{\pm00.04}$ | $83.58_{\pm00.68}$ | $82.64_{\pm00.65}$ | $40.59_{\pm01.72}$ | $79.28_{\pm01.29}$ | $68.91_{\pm00.51}$ | $\underline{76.50}_{\pm00.14}$ |
| NBFNet | $87.64_{\pm01.81}$ | $\underline{89.77}_{\pm00.80}$ | $85.56_{\pm02.07}$ | $59.78_{\pm03.73}$ | $63.23_{\pm03.65}$ | $78.24_{\pm00.90}$ | $49.97_{\pm01.44}$ | $66.36_{\pm02.64}$ |
| RMPI | $89.59_{\pm06.61}$ | $81.79_{\pm02.17}$ | $82.93_{\pm03.56}$ | $81.38_{\pm06.19}$ | N/A* | $65.76_{\pm07.45}$ | $55.67_{\pm06.61}$ | $71.03_{\pm02.12}$ |
| InGram | $92.32_{\pm01.00}$ | $83.71_{\pm03.53}$ | $90.82_{\pm01.84}$ | $92.15_{\pm00.90}$ | $61.44_{\pm09.84}$ | $87.60_{\pm01.21}$ | $54.79_{\pm08.81}$ | $67.84_{\pm06.38}$ |
| DEq-InGram (Ours) | $\underline{94.47}_{\pm00.60}$ | $88.90_{\pm02.06}$ | $\underline{93.85}_{\pm00.36}$ | $\underline{94.02}_{\pm00.74}$ | $71.94_{\pm07.37}$ | $\mathbf{91.47}_{\pm00.62}$ | $\mathbf{71.53}_{\pm04.78}$ | $\mathbf{80.53}_{\pm07.96}$ |
| ISDEA+ (Ours) | $\mathbf{95.39}_{\pm00.30}$ | $81.57_{\pm03.17}$ | $\mathbf{97.66}_{\pm00.19}$ | $\mathbf{95.03}_{\pm00.44}$ | $\mathbf{86.60}_{\pm00.59}$ | $90.93_{\pm00.24}$ | $\underline{69.62}_{\pm01.10}$ | $73.16_{\pm00.82}$ |

**Baselines.** To the best of our knowledge, InGram (Lee et al., 2023) is the first and only work capable of performing doubly inductive link prediction without needing significant modification to the model. We also run RMPI (Geng et al., 2023), which is capable of reasoning over new nodes and new relations but requires extra context at test time (test graphs either contain training relations or ontology about unseen relations). In addition, we consider the state-of-the-art link prediction model NBFNet (Zhu et al., 2021) capable of generalizing to new nodes but not new relations and modify its architecture to work with new relations at test time (following Lee et al. (2023)'s approach). We also compare our models with message-passing GNNs, including GAT (Veličković et al., 2017), GIN (Xu et al., 2019a), GraphConv (Morris et al., 2019), which treats the graph as a homogeneous graph by ignoring the relation types. For fair comparisons, we add distance features as in Equation (3) to increase the expressiveness of these GNNs. Additional baseline details are in Appendix F.

**Relation and Node Prediction Tasks.** We report the Hits@10 performances over 5 runs of different random seeds for all models on both the relation prediction task of $(i, ?, j)$ and the more traditional node prediction task of $(i, k, ?)$. For each task, we sample 50 negative triplets for each ground-truth positive target triplet during test evaluation by corrupting the relation type or the tail node respectively. *Further experiment details on synthetic tasks, additional datasets from Lee et al. (2023), baseline implementations, ablation studies, and other metrics (e.g., MRR, Hits@1) can be found in Appendix F.*

### 5.1 DOUBLY INDUCTIVE LINK PREDICTION OVER PEDIATYPES DATASET

The OpenEA library (Sun et al., 2020b) contains multiple knowledge graphs of relational databases (i.e., knowledge graphs) from different domains on similar topics, such as DBPedia (Lehmann et al., 2015) in different languages (English, French and German), YAGO (Rebele et al., 2016) and Wikidata (Vrandečić & Krötzsch, 2014). We create a new dataset **PediaTypes** (details in Appendix F.1.2) by sampling from the OpenEA library (Sun et al., 2020b), including pairs of knowledge graphs such as English-to-French DBPedia (denoted as EN-FR), DBPedia-to-YAGO (denoted as DB-YG), etc.. In each graph, triplets are randomly divided into 80% training, 10% validation, and 10% test. We then train and validate the model on one of the graphs (e.g., EN) and directly apply it to another graph (e.g., DE), which has completely new nodes and new relation types.

Table 1a shows the results on the relation prediction task, and Table 1b shows the node prediction task on PediaTypes. Across all scenarios on both tasks, our models, ISDEA+ and DEq-InGram, obtain significantly better average performance, achieving up to 50.48% absolute improvement in relation prediction and up to 23.37% absolute improvement in node prediction compared to the best-performing baseline. Furthermore, ISDEA+ tends to have smaller standard deviations than DEq-InGram, and both demonstrate much smaller standard deviations than InGram in almost all scenarios, corroborating our theoretical predictions in Section 2 that a model directly producing

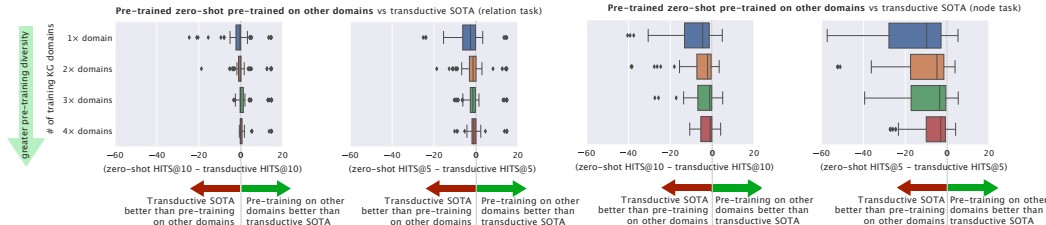

Figure 2: **ISDEA+ zero-shot self-supervised meta-learning performance on WikiTopics.** We report the Hits@10 and Hits@5 gains on node prediction $(i, k, ?)$ and relation prediction $(i, ?, j)$ tasks respectively over a standard transductive SOTA method trained on the test KG domain (denoted *transductive SOTA*). **ISDEA+ consistently improves its relative performance as it gets trained on increasingly more domains.**

double equivariant representations will be more stable than positional embeddings, which are only double equivariant in expectation.

Interestingly, we observe that in the node prediction task, the message-passing GNNs (GAT, GIN, and GraphConv) achieve quite excellent performances, even though *they completely disregard the information carried by different relation types and treat the knowledge graph as a homogeneous graph.* This observation corroborates with the conclusions of Jambor et al. (2021). Indeed, only 4 out of 8 scenarios did InGram outperform the message-passing GNNs on this task, suggesting the node prediction task might be too easy because a homogeneous link prediction model can do decently well.

### 5.2 WIKITOPICS: TESTING SELF-SUPERVISED PRE-TRAINED ZERO-SHOT META-LEARNING CAPABILITIES

WikiData-5M (Wang et al., 2021b) is a large knowledge graph dataset containing over 4M entities, 20M triplets, and 822 relation types from the Wikipedia website. The vast number of relation types span a wide range of domains, such as arts and media, education and academics, sports and gaming, etc.. Hence, we create another dataset that we denote **WikiTopics** by splitting the relation types into 11 different non-overlapping topic groups (which we refer as *domains* henceforth) and construct 11 different knowledge graphs respectively, each containing relation types specific to a particular domain (details and statistics in Appendix F.1.3). We then create a self-supervised pre-training zero-shot meta-learning task, where we train ISDEA+ on a randomly sampled increasing number of KGs of distinct domains, and test on KGs of unseen domains (i.e., over new nodes and relationship types).

This benchmark assesses the impact of increasing the number of KG domains in our self-supervised pre-training of ISDEA+ on our model's zero-shot capability to predict new relation types and new entities over unseen KG domains. Figure 2 shows the Hits@10 and Hits@5 performance of ISDEA+ as a percentage gain against the best standard state-of-the-art KG model trained on the test topic for a transductive task (i.e., test uses held out edges of the same training KG), but ISDEA+ is only pre-trained on KGs of different domains with relation types not present on the test domain. The results show that as ISDEA+ is trained over more (and diverse) KGs and domains, the performance gap between pre-training and a model trained specifically for the test topic narrows. Even more surprisingly, in some test datasets the pre-trained ISDEA+ is able to even outperform the transductive model trained on the test domain. Further discussions and more detailed results on individual training and test KG pairs on single-domain training KGs are relegated to Appendix F.1.3.

## 6 CONCLUSION

This work formally introduced the doubly inductive link prediction task defined over both new nodes and new relation types in the test data. It also defined *double equivariant models* and *distributionally double equivariant positional embedding* models for this task. We showed that, similar to how node equivariances impose learning structural node representations in unattributed graphs, double (node and relation) equivariances impose relational structure learning for knowledge graphs. We then introduced ISDEA+, a fast, accurate, and consistent double equivariant architecture that outperforms baselines in nearly all tasks, and our theory allowed us to improve an existing baseline via Monte Carlo averaging. Finally, we proposed two real-world doubly inductive link prediction benchmarks, and empirically verified the ability of our proposed approach to perform zero-shot meta-learning tasks where pre-training on more KG topics increases the zero-shot ability of our model to predict on new relation types over new entities on unseen KG topics.

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

## A    ADDITIONAL EXAMPLE FOR DOUBLY INDUCTIVE LINK PREDICTION

This example depicts an even harder scenario than the example in Figure 1, obtained from a fictional alien civilization. Knowing nothing about alien languages, we note that in training, all adjacent relations are different. Minimally, we could predict the missing relation in red in test data is not "$\not\prec$". By introducing equivariance in relations, it is possible for a model to predict relation types uniformly over the

Figure 3: **Alien knowledge graph:** The task is to predict the missing relation "?" in red. Training only tells us that relations do not repeat in a path.

set of other $(R-1)$ relations except for the existing relation "$\not\prec$", which is all we know about the aliens.

## B    CONNECTION TO DOUBLE EQUIVARIANT LOGICAL REASONING

In what follows, we follow the literature and connect link prediction in knowledge graph to logical induction (Teru et al., 2020; Zhu et al., 2022; Qiu et al., 2023). Existing logical induction requires all involved relations to be observed at least once, thus, such logical reasoning can not generalize to new relation types. We propose the Universally Quantified Entity and Relation (UQER) Horn clause, a double equivariant extension of conventional logical reasoning, which is capable of generalizing to new relation types, and show that the double invariant triplet representation in Definition 2.4 is capable of encoding such set of UQER Horn Clauses.

**Definition B.1** (Universally Quantified Entity and Relation (UQER) Horn clause). An UQER Horn clause involving $M$ nodes and $K$ relations is defined by an indicator tensor $\mathbf{B} \in \{0,1\}^{M \times K \times M}$:

$$\forall E_1 \in \mathcal{V}^{(*)}, \left(\forall E_u \in \mathcal{V}^{(*)} \setminus \{E_1, \ldots, E_{u-1}\}\right)_{u=2}^{M}, \forall C_1 \in \mathcal{R}^{(*)}, \left(\forall C_c \in \mathcal{R}^{(*)} \setminus \{C_1, \ldots, C_{c-1}\}\right)_{c=2}^{K},$$

$$\bigwedge_{\substack{u,u'=1,\ldots,M,c=1,\ldots,K, \\ \mathbf{B}_{u,c,u'}=1}} (E_u, C_c, E_{u'}) \implies (E_1, C_1, E_h),$$

(6)

for any node set $\mathcal{V}^{(*)}$ and relation set $\mathcal{R}^{(*)}$ with number of nodes and relations $N^{(*)}, R^{(*)}$ s.t. $N^{(*)} \geq M, R^{(*)} \geq K, h \in \{1,2\}$ (where $h=1$ indicates a self-loop relation or a relational node attribute), where if $M > h$, $\forall u \in \{h+1, \ldots, M\}$, $\sum_{u'=1}^{M} \sum_{c=1}^{K} \mathbf{B}_{u,c,u'} + \mathbf{B}_{u',c,u} \geq 1$, and if $K \geq 2, \forall c \in \{2, \ldots, K\}$, $\sum_{u=1}^{M} \sum_{u'=1}^{M} \mathbf{B}_{u,c,u'} + \mathbf{B}_{u',c,u} \geq 1$ (every variable should appear at least once in the formula).

Note that our definition of UQER Horn clauses (Definition B.1) is a generalization of the First Order Logic (FOL) clauses in Yang et al. (2017); Meilicke et al. (2018); Sadeghian et al. (2019); Teru et al. (2020) such that the relations in the Horn clauses are also universally quantified rather than predefined constants. UQER can be used to predict new relations in the test knowledge graph with *pattern matching*, i.e., if the left-hand-side (condition) of a UQER can be satisfied in the test knowledge graph, then the right-hand-side (implication) triplet should be present. In Figure 4, we illustrate two examples using UQER to predict new relations at test time.

We now connect our double equivariant representations (Definition 2.3) with the UQER Horn clauses.

**Theorem B.2.** *For any UQER Horn clause defined by $\mathbf{B} \in \{0,1\}^{M \times K \times M}$ (Definition B.1), there exists a double invariant triplet predictor $\Gamma_{tri} : \cup_{N=1}^{\infty} \cup_{R=2}^{\infty} ([N] \times [R] \times [N]) \times \mathbb{A} \to \{0,1\}$ (Definition 2.3), such that for any set of truth statements $\mathcal{S} \subseteq \mathcal{V}^{(*)} \times \mathcal{R}^{(*)} \times \mathcal{V}^{(*)}$ and their equivalent tensor representation $\mathbf{A}^{(*)} \in \mathbb{A}$ (where $\mathbf{A}_{i,k,j}^{(*)} = 1$ iff $(v_i^{(*)}, r_k^{(*)}, v_j^{(*)}) \in \mathcal{S}$), it satisfies $\Gamma_{tri}((i,k,j), \mathbf{A}^{(*)}) = 1$ iff $(i,k,j) \in \mathcal{S}'$, where $\mathcal{S}' = \left\{ (i,k,j) \,\middle|\, \forall (i,k,j), \text{such that} (E_1, C_1, E_2) = \left(v_i^{(*)}, r_k^{(*)}, v_j^{(*)}\right) \in \mathcal{V}^{(*)} \times \mathcal{R}^{(*)} \times \mathcal{V}^{(*)}, \exists^{M-2} E_3, \ldots, E_M \in \mathcal{V}^{(*)} \setminus \{E_1, E_2\}, \exists^{K-1} C_2, \ldots, C_K \in \mathcal{R}^{(*)} \setminus \{C_1\}, \text{where} \forall (u,c,u') \in [M] \times [K] \times [M], \mathbf{B}_{u,c,u'} = 1 \Rightarrow (E_u, C_c, E_{u'}) \in \mathcal{S} \right\}$ is the set of true statements induced by modus*

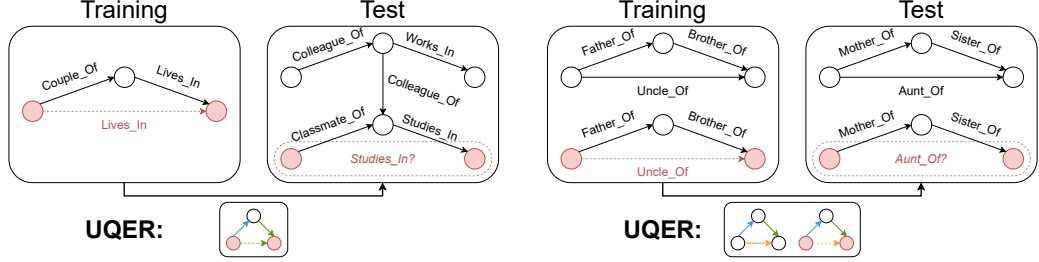

(a) **A Simple UQER Application**  (b) **A Complex UQER Application**

Figure 4: (a) The UQER (bottom) learned from training can be used to predict missing new relation "Studies_In" in red since an assignment of left-hand-side of the UQER $(E_1, \text{Classmate\_Of}, E_3) \wedge (E_3, \text{Studies\_In}, E_2)$ is satisfied in test. (b) UQER can contain disconnected components, giving more freedom to its application. For example, the UQER (bottom) can be learned from training to repeat arbitrary logical chain, which makes it possible to deal with new female relations at test time and will predict "Aunt_Of" in test just as "Uncle_Of" (red) in training.

*ponens by the truth statements $\mathcal{S}$ and the UQER Horn clause, where the existential quantifier $\exists^k$ means exists at least $k$ distinct values.*

The full proof is in Appendix D, showing how the universal quantification in Definition B.1 is a double invariant predictor.

## C  DETAILED MODEL DESIGN FOR ISDEA AND ISDEA+

The initial double equivariant neural network we propose is called ISDEA. And we propose a more efficient variant ISDEA+ with carefully designed message passing scheme for model speed and expressiveness improvement.

### C.0.1  IMPLEMENTATION DETAILS OF ISDEA

Specifically, for an knowledge graph $\mathbf{A}^{(*)}$ with number of nodes and relations $N^{(*)}, R^{(*)}$, at each iteration $t = 1, ..., T$, all nodes $i \in [N^{(*)}]$ are associated with a learned vector $h_i^{(t)} \in \mathbb{R}^{R^{(*)} \times d_t}, d_t \geq 1$. If there are no node attributes, we initialize $h_i^{(0)} = \mathbb{1}$ and $d_0 = 1$. Then we recursively compute the update, $\forall i \in [N^{(*)}], \forall k \in [R^{(*)}]$,

$$h_{i,k}^{(t+1)} = \text{GNN}_1^{(t)}\left(h_{i,k}^{(t)}, \left\{\!\!\left\{ h_{j,k}^{(t)} \middle| j \in \mathcal{N}_k(i) \right\}\!\!\right\}\right) + \text{GNN}_2^{(t)}\left(\text{AGG}_{k' \neq k}^{R^{(*)}}\left(h_{i,k'}^{(t)}\right), \left\{\!\!\left\{ \text{AGG}_{k' \neq k}^{R^{(*)}}\left(h_{j,k'}^{(t)}\right) \middle| j \in \bigcup_{k' \neq k} \mathcal{N}_{k'}(i) \right\}\!\!\right\}\right),$$

where $\text{GNN}_1^{(t)}$ and $\text{GNN}_2^{(t)}$ denote two GNN layers and $\mathcal{N}_k(i) := \left\{ j \middle| \mathbf{A}_{j,k,i}^{(*)} = 1 \right\}$ denotes the neighborhood set of node $i$ with relation $k$ in the unattributed graph $A^{(*,k)}$. At the final layers, we use standard MLPs instead of GNNs to output a final prediction. We use *mean* as our aggregators.

As shown by Srinivasan & Ribeiro (2020); You et al. (2019), structural node representations are not most expressive for link prediction in unattributed graphs. Hence, we concatenate $i$ and $j$ (double equivariant) node representations with the shortest distance between $i$ and $j$ in the observed graph as our triplet representations (appending distances is also adopted in the representations of prior work (Teru et al., 2020; Galkin et al., 2021)). Finally, we obtain the triplet representation,

$$\Gamma_{\text{ISDEA}}((i, k, j), \mathbf{A}^{(*)}) = \left( h_{i,k}^{(T)} \middle\| h_{j,k}^{(T)} \middle\| d(i,j) \middle\| d(j,i) \right), \forall (i, k, j) \in [N^{(*)}] \times [R^{(*)}] \times [N^{(*)}], \quad (7)$$

where we denote $d(i, j)$ as the length of shortest path from $i$ to $j$ without considering $(i, k, j)$, $\|$ as the concatenation operation. Since our graph is directed, we concatenate them in both directions.

**Lemma C.1.** $\Gamma_{ISDEA}$ *in Equation* (3) *is a double invariant triplet representation as per Definition 2.3.*

We choose GCN (Kipf & Welling, 2017) as our GNN kernel. In the implementation, we also follow SIGN (Frasca et al., 2020) to not have activation function between GNN layers to make the method faster and more scalable, and only have activation functions in the final MLPs in Equation (2). In the message passing scheme, we only use DSS-GNN in the first layer, while using the whole graph adjacency matrix in following layer updates. This procedure guarantees the double equivariant property of ISDEA+ and increases the expressiveness of ISDEA+ to capture more diverse relation paths. In training, we carefully design the training batch, so that the each gradient step considers only one relation vs all others. Another way to improve the computation complexity is via parallelization.

The time complexity analysis for ISDEA and ISDEA+ is detailed in Appendix F.2.

# D PROOFS

**Theorem 2.6.** *For all* $\mathbf{A}^{(*)} \in \mathbb{A}$ *with number of nodes and relations* $N^{(*)}, R^{(*)}$*, given a double invariant triplet representation* $\Gamma_{tri}$*, we can construct a double equivariant graph representation as* $\left(\Gamma_{gra}(\mathbf{A}^{(*)})\right)_{i,k,j} := \Gamma_{tri}((i,k,j), \mathbf{A}^{(*)}), \forall (i,k,j) \in [N^{(*)}] \times [R^{(*)}] \times [N^{(*)}]$*, and vice-versa.*

*Proof.* ($\Rightarrow$) For any knowledge graph $\mathbf{A}^{(*)} \in \mathbb{A}$ with number of nodes and relations $N^{(*)}, R^{(*)}$, $\Gamma_{tri} : \cup_{N=1}^{\infty} \cup_{R=2}^{\infty} ([N] \times [R] \times [N]) \times \mathbb{A} \rightarrow \mathbb{R}^d, d \geq 1$ is a double invariant triplet representation as in Definition 2.3. Using the double invariant triplet representation, we can define a function $\Gamma_{gra} : \mathbb{A} \rightarrow \cup_{N=1}^{\infty} \cup_{R=2}^{\infty} \mathbb{R}^{N \times R \times N \times d}$ such that $\forall (i,k,j) \in [N^{(*)}] \times [R^{(*)}] \times [N^{(*)}]$, $(\Gamma_{gra}(\mathbf{A}^{(*)}))_{i,k,j,:} = \Gamma_{tri}((i,k,j), \mathbf{A}^{(*)})$. Then $\forall \phi \in \mathbb{S}_{N^{(*)}}, \forall \tau \in \mathbb{S}_{R^{(*)}}$, $(\Gamma_{gra}(\phi \circ \tau \circ \mathbf{A}^{(*)}))_{\phi \circ i, \tau \circ k, \phi \circ j,:} = \Gamma_{tri}((\phi \circ i, \tau \circ k, \phi \circ j), \phi \circ \tau \circ \mathbf{A}^{(*)})$. We know $\Gamma_{tri}((i,k,j), \mathbf{A}) = \Gamma_{tri}((\phi \circ i, \tau \circ k, \phi \circ j), \phi \circ \tau \circ \mathbf{A})$. Thus we conclude, $\forall \phi \in \mathbb{S}_{N^{(*)}}, \forall \tau \in \mathbb{S}_{R^{(*)}}, \forall (i,k,j) \in [N^{(*)}] \times [R^{(*)}] \times [N^{(*)}]$, $(\phi \circ \tau \circ \Gamma_{gra}(\mathbf{A}^{(*)}))_{\phi \circ i, \tau \circ k, \phi \circ j,:} = (\Gamma_{gra}(\mathbf{A}^{(*)}))_{i,k,j,:} = \Gamma_{tri}((i,k,j), \mathbf{A}^{(*)}) = \Gamma_{tri}((\phi \circ i, \tau \circ k, \phi \circ j), \phi \circ \tau \circ \mathbf{A}^{(*)}) = (\Gamma_{gra}(\phi \circ \tau \circ \mathbf{A}^{(*)}))_{\phi \circ i, \tau \circ k, \phi \circ j,:}$. In conclusion, we show that $\phi \circ \tau \circ \Gamma_{gra}(\mathbf{A}^{(*)}) = \Gamma_{gra}(\phi \circ \tau \circ \mathbf{A}^{(*)})$, which proves the constructed $\Gamma_{gra}$ is a double equivariant representation as in Definition 2.5.

($\Leftarrow$) For any knowledge graph $\mathbf{A}^{(*)} \in \mathbb{A}$ with number of nodes and relations $N^{(*)}, R^{(*)}$, assume $\Gamma_{gra} : \mathbb{A} \rightarrow \cup_{N=1}^{\infty} \cup_{R=2}^{\infty} \mathbb{R}^{N \times R \times N \times d}$ is a double equivariant representation as Definition 2.5. Since $\Gamma_{gra}(\phi \circ \tau \circ \mathbf{A}^{(*)}) = \phi \circ \tau \circ \Gamma_{gra}(\mathbf{A}^{(*)})$, then $\forall (i,k,j) \in [N^{(*)}] \times [R^{(*)}] \times [N^{(*)}]$, $(\Gamma_{gra}(\phi \circ \tau \circ \mathbf{A}^{(*)}))_{\phi \circ i, \tau \circ k, \phi \circ j} = (\phi \circ \tau \circ \Gamma_{gra}(\mathbf{A}^{(*)}))_{\phi \circ i, \tau \circ k, \phi \circ j} = (\Gamma_{gra}(\mathbf{A}))_{i,k,j}$. Then we can define $\Gamma_{tri} : \cup_{N=1}^{\infty} \cup_{R=2}^{\infty} ([N] \times [R] \times [N]) \times \mathbb{A} \rightarrow \mathbb{R}^d, d \geq 1$, such that $\forall (i,k,j) \in [N^{(*)}] \times [R^{(*)}] \times [N^{(*)}]$, $\Gamma_{tri}((i,k,j), \mathbf{A}^{(*)}) = (\Gamma_{gra}(\mathbf{A}^{(*)}))_{i,k,j}$. It is clear that $\Gamma_{tri}((i,k,j), \mathbf{A}^{(*)}) = (\Gamma_{gra}(\mathbf{A}^{(*)}))_{i,k,j} = (\Gamma_{gra}(\phi \circ \tau \circ \mathbf{A}^{(*)}))_{\phi \circ i, \tau \circ k, \phi \circ j} = \Gamma_{tri}((\phi \circ i, \tau \circ k, \phi \circ j), \phi \circ \tau \circ \mathbf{A}^{(*)})$. Thus, we show $\Gamma_{tri}$ is a double invariant triplet representation as in Definition 2.3. $\square$

**Theorem 2.8** (From distributional double equivariant positional embeddings to double equivariant representations). *For any knowledge graph* $\mathbf{A}^{(*)} \in \mathbb{A}$*, the average* $\mathbb{E}_{p(\mathbf{Z}|\mathbf{A}^{(*)})}[\mathbf{Z}|\mathbf{A}^{(*)}]$ *is a double equivariant knowledge graph representation (Definition 2.5) for any distributional double equivariant positional embeddings* $\mathbf{Z}|\mathbf{A}^{(*)}$ *(Definition 2.7).*

*Proof.* Based on Definition 2.7, for any knowledge graph $\mathbf{A}^{(*)} \in \mathbb{A}$ with number of nodes and relations $N^{(*)}, R^{(*)}$, the distributionally double equivariant positional embeddings of $\mathbf{A}^{(*)}$ are defined as joint samples of random variables $\mathbf{Z}|\mathbf{A}^{(*)} \sim p(\mathbf{Z}|\mathbf{A}^{(*)})$, where the tensor $\mathbf{Z}$ is defined as $\mathbf{Z}_{i,k,j} \in \mathbb{R}^d, d \geq 1, \forall (i,k,j) \in [N^{(*)}] \times [R^{(*)}] \times [N^{(*)}]$, where we say $p(\mathbf{Z}|\mathbf{A}^{(*)})$ is a double equivariant probability distribution on $\mathbf{A}^{(*)}$ defined as $\forall \phi \in \mathbb{S}_{N^{(*)}}, \forall \tau \in \mathbb{S}_{R^{(*)}}, p(\mathbf{Z}|\mathbf{A}^{(*)}) = p(\phi \circ \tau \circ \mathbf{Z}|\phi \circ \tau \circ \mathbf{A}^{(*)})$.

The tensor $\mathbf{Z}$ is defined as $\mathbf{Z}_{i,k,j} \in \mathbb{R}^d, \forall (i,k,j) \in [N^{(*)}] \times [R^{(*)}] \times [N^{(*)}]$, thus $\mathbf{Z} \in \mathbb{R}^{N^{(*)} \times R^{(*)} \times N^{(*)} \times d}$. So we can consider $\mathbb{E}_{p(\mathbf{Z}|\mathbf{A}^{(*)})}[\mathbf{Z}|\mathbf{A}^{(*)}]$ as a function on $\mathbf{A}^{(*)}$, and output a representation in $\mathbb{R}^{N^{(*)} \times R^{(*)} \times N^{(*)} \times d}$. Since $\forall \phi \in \mathbb{S}_{N^{(*)}}, \forall \tau \in \mathbb{S}_{R^{(*)}}, p(\mathbf{Z}|\mathbf{A}^{(*)}) = p(\phi \circ \tau \circ \mathbf{Z}|\phi \circ \tau \circ \mathbf{A}^{(*)})$,

it is clear to have $\forall \phi \in \mathbb{S}_{N^{(*)}}, \forall \tau \in \mathbb{S}_{R^{(*)}}, \phi \circ \tau \circ \mathbb{E}_{p(\mathbf{Z}|\mathbf{A}^{(*)})}[\mathbf{Z}|\mathbf{A}^{(*)}] = \phi \circ \tau \circ \int z p(\mathbf{Z} = z|\mathbf{A}^{(*)}) dz = \int \phi \circ \tau \circ z p(\mathbf{Z} = z|\mathbf{A}^{(*)}) dz = \int \phi \circ \tau \circ z p(\phi \circ \tau \circ \mathbf{Z} = \phi \circ \tau \circ z|\phi \circ \tau \circ \mathbf{A}^{(*)}) d(\phi \circ \tau \circ z) = \mathbb{E}_{p(\phi \circ \tau \circ \mathbf{Z}|\phi \circ \tau \circ \mathbf{A}^{(*)})}[\phi \circ \tau \circ \mathbf{Z}|\phi \circ \tau \circ \mathbf{A}^{(*)}]$. Since the permutation $\phi, \tau$ only changes the ordering of the output representation element-wise, we can interchange the permutations with the integral.

Finally, for any knowledge graph $\mathbf{A}^{(*)} \in \mathbb{A}$ with number of nodes and relations $N^{(*)}, R^{(*)}$, we can define $\Gamma_{\mathrm{gra}}(\mathbf{A}^{(*)}) : \mathbb{A} \to \cup_{N=1}^{\infty} \cup_{R=2}^{\infty} \mathbb{R}^{N \times R \times N \times d}, d \geq 1$ such that $\Gamma_{\mathrm{gra}}(\mathbf{A}^{(*)}) := \mathbb{E}_{p(\mathbf{Z}|\mathbf{A}^{(*)})}[\mathbf{Z}|\mathbf{A}^{(*)}]$. And we can derive $\phi \circ \tau \circ \Gamma_{\mathrm{gra}}(\mathbf{A}^{(*)}) = \phi \circ \tau \circ \mathbb{E}_{p(\mathbf{Z}|\mathbf{A}^{(*)})}[\mathbf{Z}|\mathbf{A}^{(*)}] = \mathbb{E}_{p(\phi \circ \tau \circ \mathbf{Z}|\phi \circ \tau \circ \mathbf{A}^{(*)})}[\phi \circ \tau \circ \mathbf{Z}|\phi \circ \tau \circ \mathbf{A}^{(*)}] = \Gamma_{\mathrm{gra}}(\phi \circ \tau \circ \mathbf{A}^{(*)})$. Thus, $\Gamma_{\mathrm{gra}}(\mathbf{A}^{(*)}) := \mathbb{E}_{p(\mathbf{Z}|\mathbf{A}^{(*)})}[\mathbf{Z}|\mathbf{A}^{(*)}]$ is a double equivariant knowledge graph representation as per Definition 2.5. $\square$

**Lemma 3.1.** $\Gamma_{ISDEA+}$ *in Equation* (3) *is a double invariant triplet representation as per Definition 2.3.*

*Proof.* From the ISDEA+ model architecture (Equation (7)), $\Gamma_{\mathrm{ISDEA+}}((i, k, j), \mathbf{A}^{(*)}) = (X_{i,k} \parallel X_{j,k} \parallel d(i, j) \parallel d(j, i))$. Using DSS layers, we can guarantee the node representations $X_{i,k}$ we learn are double invariant under the node and relation permutations, where $X_{i,k}$ in $\mathbf{A}^{(*)}$ is equal to $X_{\phi \circ i, \tau \circ k}$ in $\phi \circ \tau \circ \mathbf{A}^{(*)}$. It is also clear that the distance function is invariant to node and relation permutations, i.e. $\forall i, j \in [N^{(*)}]$, $d(i, j)$ in $\mathbf{A}^{(*)}$ is the same as $d(\phi \circ i, \phi \circ j)$ in $\phi \circ \tau \circ \mathbf{A}^{(*)}$. Thus $\Gamma_{\mathrm{ISDEA+}}((i, k, j), \mathbf{A}^{(*)}) = \Gamma_{\mathrm{ISDEA+}}((\phi \circ i, \tau \circ k, \phi \circ j), \phi \circ \tau \circ \mathbf{A}^{(*)})$ is a double invariant triplet representation as in Definition 2.3. $\square$

**Lemma C.1.** $\Gamma_{ISDEA}$ *in Equation* (3) *is a double invariant triplet representation as per Definition 2.3.*

*Proof.* From the ISDEA model architecture (Equation (7)), $\Gamma_{\mathrm{ISDEA}}((i, k, j), \mathbf{A}^{(*)}) = (h_{i,k}^{(T)} \parallel h_{j,k}^{(T)} \parallel d(i, j) \parallel d(j, i))$. Using DSS layers, we can guarantee the node representations $h_{i,k}^{(T)}$ we learn are double invariant under the node and relation permutations, where $h_{i,k}^{(T)}$ in $\mathbf{A}^{(*)}$ is equal to $h_{\phi \circ i, \tau \circ k}^{(T)}$ in $\phi \circ \tau \circ \mathbf{A}^{(*)}$. It is also clear that the distance function is invariant to node and relation permutations, i.e. $\forall i, j \in [N^{(*)}]$, $d(i, j)$ in $\mathbf{A}^{(*)}$ is the same as $d(\phi \circ i, \phi \circ j)$ in $\phi \circ \tau \circ \mathbf{A}^{(*)}$. Thus $\Gamma_{\mathrm{ISDEA}}((i, k, j), \mathbf{A}^{(*)}) = \Gamma_{\mathrm{ISDEA}}((\phi \circ i, \tau \circ k, \phi \circ j), \phi \circ \tau \circ \mathbf{A}^{(*)})$ is a double invariant triplet representation as in Definition 2.3. $\square$

**Lemma 3.2.** *The triplet representations generated by InGram (Lee et al., 2023) output distributionally double equivariant positional embeddings (Definition 2.7).*

*Proof.* To solve doubly inductive link prediction, InGram (Lee et al., 2023) first constructs a *relation graph*, in which the relation types are treated as nodes, and the edges between them are weighted by the affinity scores, a measure of co-occurrence between relation types in the original knowledge graph. It then employs a variant of the GATv2 (Veličković et al., 2017; Brody et al., 2021) on the relation graph to propagate and generate embeddings for the relation types. These relation embeddings, together with another GATv2, are applied to the original knowledge graph to generate embeddings for the nodes. Finally, a variant of DistMult (Yang et al., 2015) is used to compute the scores for individual triplets from the embeddings of the head and tail nodes and the embedding of the relation.

If the input node and relation embeddings to the InGram model were to be the same across all nodes and across all relation types respectively (such as vectors of all ones), then InGram would have produced double structural representations for the triplets (definition 2.3). Simply put, this is because the relation graphs proposed by Lee et al. (2023) encode only the structural features of the relation types (their mutual structural affinity), which is double equivariant to the permutation of relation type and node indices. Since the same initial embeddings for all nodes and relations are naively double equivariant, and the GATv2 (Veličković et al., 2017; Brody et al., 2021) is a message-passing neural network (Gilmer et al., 2017) that also produces equivariant representations, the final relation embeddings would be double equivariant. Same analysis will also show the final node embeddings are double equivariant.

However, to improve the expressivity of the model, Lee et al. (2023) chose to randomly re-initialize the input embeddings for all node and relation types using Glorot initialization (Glorot & Bengio, 2010) *for each epoch during training*, a technique inspired by recent studies on

the expressive power of GNNs (Abboud et al., 2020; Sato et al., 2021; Murphy et al., 2019). Unfortunately, random initial features break the double equivariance of the generated representations, making them sensitive to the permutation of node and relation type indices. However, since the initial node $\boldsymbol{V}^{(0)}$ and relation embeddings $\boldsymbol{R}^{(0)}$ are randomly initialized, and by design of InGram architecture, we have $\forall(i,k,j) \in [N^{(*)}] \times [R^{(*)}] \times [N^{(*)}], \forall \phi \in \mathbb{S}_{N^{(*)}}, \tau \in \mathbb{S}_{R^{(*)}}, \mathbf{Z}_{\text{InGram}}((i,k,j), \mathbf{A}^{(*)}, \boldsymbol{V}^{(0)}, \boldsymbol{R}^{(0)}) = \mathbf{Z}_{\text{InGram}}((\phi \circ i, \tau \circ k, \phi \circ j), \phi \circ \mathbf{A}^{(*)}, \boldsymbol{V}^{(0)}, \boldsymbol{R}^{(0)})$ for any random samples of node and relation embeddings $v^{(0)}, r^{(0)}$. We define $\mathbf{Z}_{\text{InGram}}|\mathbf{A}^{(*)} = [\mathbf{Z}_{\text{InGram}}((i,k,j), \mathbf{A}^{(*)}, \boldsymbol{V}^{(0)}, \boldsymbol{R}^{(0)}))]_{(i,k,j) \in [N^{(*)}] \times [R^{(*)}] \times [N^{(*)}]}$, and $\phi \circ \tau \circ \mathbf{Z}_{\text{InGram}}|\phi \circ \tau \circ \mathbf{A}^{(*)} = [\mathbf{Z}_{\text{InGram}}((\phi \circ i, \tau \circ k, \phi \circ j), \phi \circ \tau \circ \mathbf{A}^{(*)}, \boldsymbol{V}^{(0)}, \boldsymbol{R}^{(0)}))]_{(\phi \circ i, \tau \circ k, \phi \circ j) \in [N^{(*)}] \times [R^{(*)}] \times [N^{(*)}]}$. Since $\boldsymbol{V}^{(0)}, \boldsymbol{R}^{(0)}$ random variables that do not change with permutations, we can easily derive $p(\phi \circ \tau \circ \mathbf{Z}_{\text{InGram}}|\phi \circ \tau \circ \mathbf{A}^{(*)}) = p(\mathbf{Z}_{\text{InGram}}|\mathbf{A}^{(*)})$. Thus, InGram is a distributionally double equivariant positional graph embedding of $\mathbf{A}^{(*)}$ as per Definition 2.7. $\qquad\square$

**Theorem B.2.** *For any UQER Horn clause defined by* $\mathbf{B} \in \{0,1\}^{M \times K \times M}$ *(Definition B.1), there exists a double invariant triplet predictor* $\Gamma_{tri} : \cup_{N=1}^{\infty} \cup_{R=2}^{\infty} ([N] \times [R] \times [N]) \times \mathbb{A} \to \{0,1\}$ *(Definition 2.3), such that for any set of truth statements* $\mathcal{S} \subseteq \mathcal{V}^{(*)} \times \mathcal{R}^{(*)} \times \mathcal{V}^{(*)}$ *and their equivalent tensor representation* $\mathbf{A}^{(*)} \in \mathbb{A}$ *(where* $\mathbf{A}_{i,k,j}^{(*)} = 1$ *iff* $(v_i^{(*)}, r_k^{(*)}, v_j^{(*)}) \in \mathcal{S}$*), it satisfies* $\Gamma_{tri}((i,k,j), \mathbf{A}^{(*)}) = 1$ *iff* $(i,k,j) \in \mathcal{S}'$*, where* $\mathcal{S}' = \left\{ (i,k,j) \,\middle|\, \forall (i,k,j), \text{such that} (E_1, C_1, E_2) = \left( v_i^{(*)}, r_k^{(*)}, v_j^{(*)} \right) \in \mathcal{V}^{(*)} \times \mathcal{R}^{(*)} \times \mathcal{V}^{(*)}, \exists^{M-2} E_3, ..., E_M \in \mathcal{V}^{(*)} \setminus \{E_1, E_2\}, \exists^{K-1} C_2, ..., C_K \in \mathcal{R}^{(*)} \setminus \{C_1\}, \text{where} \forall (u,c,u') \in [M] \times [K] \times [M], \mathbf{B}_{u,c,u'} = 1 \Rightarrow (E_u, C_c, E_{u'}) \in \mathcal{S} \right\}$ *is the set of true statements induced by modus ponens by the truth statements* $\mathcal{S}$ *and the UQER Horn clause, where the existential quantifier* $\exists^k$ *means exists at least* $k$ *distinct values.*

*Proof.* Recall that we have two different cases $h = 1$ and $h = 2$ for Equation (6) in Definition B.1 of UQER. For the ease of proof, we will focus on the case where $h = 2$ in the following content, and for the case $h = 1$, the proof will be the same.

Given $h = 2$, any UQER is defined by $\mathbf{B} \in \{0,1\}^{M \times K \times M}$ as

$$\forall E_1 \in \mathcal{V}^{(*)}, \left( \forall E_u \in \mathcal{V}^{(*)} \setminus \{E_1, \ldots, E_{u-1}\} \right)_{u=2}^{M}, \forall C_1 \in \mathcal{R}^{(*)}, \left( \forall C_c \in \mathcal{R}^{(*)} \setminus \{C_1, \ldots, C_{c-1}\} \right)_{c=2}^{K},$$
$$\bigwedge_{\substack{u,u'=1,\ldots,M, c=1,\ldots,K, \\ \mathbf{B}_{u,c,u'}=1}} (E_u, C_c, E_{u'}) \implies (E_1, C_1, E_h),$$

(8)

for any node set $\mathcal{V}^{(*)}$ and relation set $\mathcal{R}^{(*)}$ with number of nodes and relations $N^{(*)}, R^{(*)}$ s.t. $N^{(*)} \geq M, R^{(*)} \geq K$, where if $M > 2, \forall u \in \{3, \ldots, M\}, \sum_{u'=1}^{M} \sum_{c=1}^{K} \mathbf{B}_{u,c,u'} + \mathbf{B}_{u',c,u} \geq 1$, and if $K \geq 2, \forall c \in \{2, \ldots, K\}, \sum_{u=1}^{M} \sum_{u'=1}^{M} \mathbf{B}_{u,c,u'} + \mathbf{B}_{u',c,u} \geq 1$ (every variable should appear at least once in the formula).

For all sets of truth statements $\forall \mathcal{S} \subseteq \cup_{N=1}^{\infty} \cup_{R=2}^{\infty} \mathcal{V}^{(*)} \times \mathcal{R}^{(*)} \times \mathcal{V}^{(*)}$, it has an equivalent tensor representation $\mathbf{A}^{(*)} \in \{0,1\}^{N^{(*)} \times R^{(*)} \times N^{(*)}}$ such that $\mathbf{A}_{i,k,j} = 1 \iff (v_i^{(*)}, r_k^{(*)}, v_j^{(*)}) \in \mathcal{S}$. We can then define a triplet representation $\Gamma_{\text{tri}}$ based on the given UQER as, $\forall (i,k,j) \in [N^{(*)}] \times [R^{(*)}] \times [N^{(*)}]$,

$$\Gamma_{\text{tri}}((i,k,j), \mathbf{A}^{(*)}) = \begin{cases} 1 & \text{if } (i,k,j) \in \mathcal{S}' \\ 0 & \text{otherwise}, \end{cases}$$

(9)

where we define $\mathcal{S}' = \left\{ (i,k,j) \,\middle|\, \forall (i,k,j) \in [N^{(*)}] \times [R^{(*)}] \times [N^{(*)}], \text{such that} (E_1, C_1, E_2) = \left( v_i^{(*)}, r_k^{(*)}, v_j^{(*)} \right) \in \mathcal{V}^{(*)} \times \mathcal{R}^{(*)} \times \mathcal{V}^{(*)}, \exists^{M-2} E_3, ..., E_M \in \mathcal{V}^{(*)} \setminus \{E_1, E_2\}, \exists^{K-1} C_2, ..., C_K \in \mathcal{R}^{(*)} \setminus \{C_1\}, \text{where} \forall (u,c,u') \in [M] \times [K] \times [M], \mathbf{B}_{u,c,u'} = 1 \Rightarrow (E_u, C_c, E_{u'}) \in \mathcal{S} \right\}$ is the set of true statements induced by modus ponens from the truth statements $\mathcal{S}$ and the UQER Horn Clause, where the existential quantifier $\exists^k$ means exists at least $k$ distinct values.

All we need to show is that Equation (9) is a double invariant triplet representation. For any node permutation $\phi \in \mathbb{S}_{N^{(*)}}$ and relation permutation $\tau \in \mathbb{S}_{R^{(*)}}$ of $\mathbf{A}^{(*)}$, we define $\phi \circ \tau \circ \mathcal{S} =$

$\{(v_{\phi \circ i}^{(*)}, r_{\tau \circ k}^{(*)}, v_{\phi \circ i}^{(*)}) | (v_i^{(*)}, r_k^{(*)}, v_j^{(*)}) \in \mathcal{S}\}$ which corresponds to their equivalent tensor representation $\phi \circ \tau \circ \mathbf{A}^{(*)}$, where $(\phi \circ \tau \circ \mathbf{A}^{(*)})_{\phi \circ i, \tau \circ k, \phi \circ j} = 1 \iff (v_i^{(*)}, r_k^{(*)}, v_j^{(*)}) \in \mathcal{S}$ otherwise 0. Similarly, we have $\phi \circ \tau \circ \mathcal{S}' = \{ (\phi \circ i, \tau \circ k, \phi \circ j) \mid \forall (i, k, j) \in [N^{(*)}] \times [R^{(*)}] \times [N^{(*)}]$, such that $(E_1, C_1, E_2) = \left( v_{\phi \circ i}^{(*)}, r_{\tau \circ k}^{(*)}, v_{\phi \circ j}^{(*)} \right) \in \mathcal{V}^{(*)} \times \mathcal{R}^{(*)} \times \mathcal{V}^{(*)}, \exists^{M-2} E_3, ..., E_M \in \mathcal{V}^{(*)} \setminus \{E_1, E_2\}, \exists^{K-1} C_2, ..., C_K \in \mathcal{R}^{(*)} \setminus \{C_1\}$, where $\forall (u, c, u') \in [M] \times [K] \times [M], \mathbf{B}_{u,c,u'} = 1 \Rightarrow (\phi \circ E_u, \tau \circ C_c, \phi \circ E_{u'}) \in \phi \circ \tau \circ \mathcal{S}\}$.

By definition, we have that for any $(i, k, j) \in \mathcal{S}'$,

$$\Gamma_{\text{tri}}((\phi \circ i, \tau \circ k, \phi \circ j), \phi \circ \tau \circ \mathbf{A}^{(*)}) = \begin{cases} 1 & \text{if } (\phi \circ i, \tau \circ k, \phi \circ j) \in \phi \circ \tau \circ \mathcal{S}' \\ 0 & \text{otherwise,} \end{cases}.$$

Now we show that $(i, k, j) \in \mathcal{S}'$ if and only if $(\phi \circ i, \tau \circ k, \phi \circ j) \in \phi \circ \tau \circ \mathcal{S}'$. If $(i, k, j) \in \mathcal{S}'$, then $E_1 = v_i^{(*)}, E_2 = v_j^{(*)}, C_1 = r_k^{(*)}, \exists^{M-2} E_3, ..., E_M \in \mathcal{V}^{(*)} \setminus \{E_1, E_2\}, \exists^{K-1} C_2, ..., C_K \in \mathcal{R}^{(*)} \setminus \{C_1\}$, such that $\mathbf{B}_{u,c,u'} = 1 \implies (E_u, C_c, E_{u'}) \in \mathcal{S}$. Since $(E_u, C_c, E_{u'}) \in \mathcal{S}$ if and only if $(\phi \circ E_u, \tau \circ C_c, \phi \circ E_{u'}) \in \phi \circ \tau \circ \mathcal{S}$ by definition, we have $(\phi \circ i, \tau \circ k, \phi \circ j) \in \phi \circ \tau \circ \mathcal{S}'$. Similarly we can prove if $(\phi \circ i, \tau \circ k, \phi \circ j) \in \phi \circ \tau \circ \mathcal{S}'$, then $(i, k, j) \in \mathcal{S}'$ with the same reasoning.

In conclusion, for any $\mathbf{A}^{(*)} \in \mathbb{A}$ with number of nodes and relations $N^{(*)}, R^{(*)}$, since $(i, k, j) \in \mathcal{S}'$ if and only if $(\phi \circ i, \tau \circ k, \phi \circ j) \in \phi \circ \tau \circ \mathcal{S}'$, then by definition $\Gamma_{\text{tri}}((\phi \circ i, \tau \circ k, \phi \circ j), \phi \circ \tau \circ \mathbf{A}^{(*)}) = \Gamma_{\text{tri}}((i, k, j), \mathbf{A}^{(*)})$ holds $\forall (i, k, j) \in [N^{(*)}] \times [R^{(*)}] \times [N^{(*)}]$, which proves $\Gamma_{\text{tri}}$ is a double invariant triplet representation (Definition 2.3).

$\square$

## E  ADDITIONAL RELATED WORK

Link prediction in knowledge graphs, which are commonly used to represent relational data in a structured way by indicating different types of relations between pairs of nodes in the graph, involves predicting not only the existence of missing edges but also the associated relation types.

**Transductive link prediction.**  In transductive link prediction, missing links are predicted over a fixed set of nodes and relation types as in training. Traditionally, factorization-based methods (Sutskever et al., 2009; Nickel et al., 2011; Bordes et al., 2013; Wang et al., 2014; Yang et al., 2015; Trouillon et al., 2016; Nickel et al., 2016; Trouillon et al., 2017; Dettmers et al., 2018; Sun et al., 2019) have been proposed to obtain latent embedding of nodes and relation types to capture their relative information in the graph. These models try to score all combinations of nodes and relations with embeddings as factors, similar to tensor factorization. Although excellence in transductive tasks, these positional embeddings (Srinivasan & Ribeiro, 2020) (a.k.a. permutation-sensitive embeddings) require extensive retraining to perform inductive tasks over new nodes or relations (Teru et al., 2020). However, in real-world applications, relational data is often evolving, requiring link prediction over new nodes and new relation types, or even entirely new graphs.

**Inductive link prediction over new nodes (but not new relations) with GNN-based model.**  In recent years, with the advancement of graph neural networks (GNNs) (Defferrard et al., 2016; Kipf & Welling, 2017; Hamilton et al., 2017; Veličković et al., 2017; Bronstein et al., 2017; Murphy et al., 2019), in graph machine learning fields, various works has applied the idea of GNN in relational prediction to ensure the inductive capability of the model, including RGCN (Schlichtkrull et al., 2018), CompGCN (Vashishth et al., 2019), GraIL (Teru et al., 2020), NodePiece (Galkin et al., 2021), NBFNet (Zhu et al., 2021), ReFactorGNNs (Chen et al., 2022b) etc.. Specifically, RGCN (Schlichtkrull et al., 2018) and CompGCN (Vashishth et al., 2019) were initially designed for transductive link prediction tasks. As GNNs are node permutation equivariant (Xu et al., 2019a; Srinivasan & Ribeiro, 2020), these models learn structural node/pairwise representation, which can be used to perform *inductive link prediction over solely new nodes*, while most of the GNN performance are worse than FM-based methods (Ruffinelli et al., 2020; Chen et al., 2022b). Specifically, Teru et al. (2020) extends the idea from (Zhang & Chen, 2018) to use local subgraph representations for knowledge graph link prediction. Chen et al. (2022b) aims to build the connection between FM and

GNNs, where they propose an architecture to cast FMs as GNNs. Galkin et al. (2021) uses anchor-nodes for parameter-efficient architecture for knowledge graph completion. Zhu et al. (2021) extends the Bellman-Ford algorithm, which learns pairwise representations by all the path representations between nodes. (Barcelo et al., 2022) analyzes knowledge graph-GNNs expressiveness by connecting it with the Weisfeiler-Leman test in knowledge graph.

**Inductive link prediction over new nodes (but not new relations) with logical induction.** The relation prediction problem in relational data represented by knowledge graph can also be considered as the problem of learning first-order logical Horn clauses (Yang et al., 2015; 2017; Sadeghian et al., 2019; Teru et al., 2020) from the relational data, where one aims to extract logical rules on binary predicates. These methods are inherently node-independent and are able to perform *inductive link prediction over solely new nodes*. Barceló et al. (2020) discusses the connection between the expressiveness of GNNs and first-order logical induction, but only on node GNN representation and logical node classifier. Qiu et al. (2023) further analyzes the logical expressiveness of GNNs for knowledge graph by showing GNNs are able to capture logical rules from graded modal logic and provides a logical explanation of why pairwise GNNs (Zhang et al., 2021; Zhu et al., 2021) can achieve SOTA results. In our paper, we try to build the connection between triplet representation and logical Horn clauses. Traditionally, logical rules are learned through statistically enumerating patterns observed in knowledge graph (Lao & Cohen, 2010; Galárraga et al., 2013). Neural LP (Yang et al., 2017) and DRUM (Sadeghian et al., 2019) learn logical rules in an end-to-end differentiable manner using the set of logic paths between two nodes with sequence models. Cheng et al. (2022) follows a similar manner, which breaks a big sequential model into small atomic models in a recursive way. Galkin et al. (2022) aims to inductively extract logical rules by devising NodePiece (Galkin et al., 2021) and NBFNet (Zhu et al., 2021). However, all these methods are not able to deal with new relation types in test.

**Inductive link prediction over both new nodes and new relations (with extra context)** Few-shot and zero-shot relational reasoning (Xiong et al., 2018; Lv et al., 2019; Qin et al., 2020; Zhao et al., 2020; Geng et al., 2021; Wang et al., 2021a; Huang et al., 2022; Chen et al., 2023; Geng et al., 2023) aim to query triplets involving unseen relation types with access to few or zero support triplets of these unseen relation types at test time. Recent methods (Qin et al., 2020; Zhao et al., 2020; Huang et al., 2022; Geng et al., 2023) can even query over unseen nodes. Yet, they often need extra context in the test graph, such as textual descriptions and/or ontological information of the unseen relation types or a shared background graph between the training and test graph, i.e., the test nodes and relation types are connected to the training ones. For instance, zero-shot link prediction methods such as Qin et al. (2020) employ a generative adversarial network (Goodfellow et al., 2015) to utilize the additional textual information to bridge the semantic gap between seen and unseen relations. Later, Geng et al. (2021) presented an ontology-enhanced zero-shot learning approach that incorporates both ontology structural and textural information. Similarly, TACT (Chen et al., 2021b) aims to model the topological correlations between the target relations and their adjacent relations (assumes there are relations that are seen in train) using a relational correlation network to learn more expressive representations of the target relations. A recent work is RMPI (Geng et al., 2023) that extracts enclosing subgraphs around the target triplet, which are assumed to contain triplets of some relation types seen in training and uses graph ontology to bridge the unseen relation types to the seen ones. Zhao et al. (2020) uses attention-based GNNs and convolutional transition for link prediction over new nodes and new relations assuming a shared background graph between training and test (i.e., new relations in test are connected with existing nodes and relations in training). MaKEr (Chen et al., 2022a) also uses the local graph structure to handle new nodes and new relation types using a meta-learning framework, assuming the test graph has overlapping relations and entities with the training graph. On the other hand, few-shot relational reasoning methods learn representations of the unseen relation types from the few support triplets, which are generally assumed to connect to existing nodes and relations seen in training (Xiong et al., 2018; Chen et al., 2019; Zhang et al., 2020). For example, Xiong et al. (2018) was the first to solve the one-shot task by proposing to compute matching scores between the new relation types observed in the support set to those training relation types. Later, Zhang et al. (2020) extends Xiong et al. (2018) by using an attention-based aggregation to take advantage of information from all support triplets. Recently, Huang et al. (2022) proposed a hypothesis testing method that matches the new relation types to the training ones by learning to compare the similarity between the connection subgraph patterns surrounding the target triplets. Another line of research is to solve few-shot relational reasoning via meta-learning. For instance,

Chen et al. (2019) updates a meta representation over the relation types, and Lv et al. (2019) adopts MAML (Finn et al., 2017) to learn meta parameters for frequently occurring relations, which can then be adapted to few-shot relations. All of these few-shot learning methods, however, require that the few-shot triplets are connected to a background graph observed during training in order to learn about the relationship between new relation types and existing ones. Hence, all these methods cannot be directly applied to test graphs that neither contain textual descriptions of the unseen relation types nor triplets involving those relation types seen in training.

**Inductive link prediction over both new nodes and new relations (no extra context)**   In this paper, we focus on the most general task, i.e., inductive link prediction over both new nodes and new relations on entirely new test graphs without textual descriptions, which we call *doubly inductive link prediction*. To the best of our knowledge, InGram (Lee et al., 2023) is the first and only existing method capable of performing this task. In contrast to Lee et al. (2023) that designed a specific architecture, i.e., InGram, our work proposes a general theoretical framework for designing an entire class of models capable of solving the doubly inductive link prediction task, which encompasses InGram as a specific instantiation. Modeling details of InGram have been substantially discussed in the main paper.

**Knowledge graph alignment.**   Knowledge graph alignment tasks (Sun et al., 2018; 2020a; Yan et al., 2021; Sun et al., 2020b) are very common in heterogeneous, cross-lingual, and domain-specific relational data, where the task aims to align nodes among different domains. For example, matching nodes with their counterparts in different languages (Wang et al., 2018; Xu et al., 2019b). It is intrinsically different than our task, where we aim to inductively apply on completely new nodes and relations, possibly with no clear alignments between them.

## F   EXPERIMENTS

Our code is available at `https://anonymous.4open.science/r/ISDEA-Fix-B3D7`.

### F.1   DOUBLY INDUCTIVE LINK PREDICTION TASK OVER BOTH NEW NODES AND NEW RELATION TYPES

In this section, we provide more detailed experiment results and analysis for our method on inductively doubly inductive link prediction on both new nodes and new relation types.

**Datasets.**   To the best of our knowledge, there are no existing real-world benchmarks that are specially designed to test a model's extrapolation capability for doubly inductive link prediction task by training the model on one graph and testing it on another completely new graph coming from different domains and distributions. Existing datasets such as NL-100, WK-100, and FB-100 from Lee et al. (2023) are typically created by randomly splitting a larger graph (e.g. NELL-995 (Xiong et al., 2018), Wikidata68K (Gesese et al., 2022), FB15K237 (Toutanova & Chen, 2015)) into disjoint node and relation sets, implying that the test and training graphs still come from the same distribution. In contrast, we purposefully create two doubly inductive link prediction benchmark datasets: PediaTypes and WikiTopics, sampled respectively from the OpenEA library (Sun et al., 2020b) and WikiData-5M (Wang et al., 2021b), where by design the test and training graphs are either from different domains or different topic groups and are likely to possess different characteristics to fully test model's capability for doubly inductive link prediction. We also propose another task with modifications of the NL-$k$, WK-$k$, and FB-$k$ datasets from InGram (Lee et al., 2023) and one synthetic task FD2 to study the expressive power of ISDEA.

#### F.1.1   EXPERIMENT SETUP

**Baselines.**   To the best of our knowledge, InGram (Lee et al., 2023) is the first and only work capable of performing doubly inductive link prediction without needing significant modification to the model. Hence, we chose InGram as one baseline. We also run RMPI (Geng et al., 2023), which is capable of reasoning over new nodes and new relations but requires extra context at test time (test graphs either contain training relations or ontological information of unseen relations). We simply provide randomized embeddings of unseen relations at test time following Lee et al. (2023). In addition, we

consider the state-of-the-art link prediction model NBFNet (Zhu et al., 2021) capable of generalizing over to new nodes but not new relations and modifying its architecture to work with new relations at test time by providing randomized embeddings of unseen relations at test time following Lee et al. (2023). We also compare our models with message-passing GNNs including GAT (Veličković et al., 2017), GIN (Xu et al., 2019a), GraphConv (Morris et al., 2019) which treats the graph as a homogeneous graph by ignoring the relation types. For fair comparisons, we add distance features as in Equation (3) to increase the expressiveness of these GNNs. For training of each single run, we augment each triplet $(i, k, j)$ by its inversion $(i, k^{-1}, j)$, and sample 2 negative (node) triplets $(i', k, j')$ and 2 negative (relation) triplets $(i, k', j)$ per positive in training as Sun et al. (2019); Zhu et al. (2021). Training was performed on NVidia A100s, L4s, GeForce RTX 2080 Ti, and TITAN V GPUs.

**Evaluation Metrics.** We sample 50 negative triplets for each test positive triplet during test evaluation by corrupting either nodes or relation types (Equation (4)), and use Nodes Hits@$k$ and Relation Hits@$k$ separately which counts the ratio of positive triplets ranked at or above the $k$-th place against the 50 negative samples as evaluation metric over 5 runs. Specifically, for Node prediction evaluation, we sample without replacement 50 negative tail (or head) nodes, and for Relation prediction evaluation, we sample with replacement 50 negative relation types (can also handle cases where the number of test relations is less than 50). We also report other widely used metrics such as MRR.

**Hyperparameters and Implementation Details.** For homogeneous GNN methods, NBFNet and ISDEA+, We follow the same configuration as Teru et al. (2020) such that the hidden layers have 32 neurons. We use Adam optimizer with grid search over learning rate $\alpha \in \{0.01, 0.001, 0.0001\}$, and over weight decay $\beta \in \{0.0005, 0\}$. For all datasets, we train these models for 10 epochs with a mini-batch size of 16. For the GNN kernel we choose GCN (Kipf & Welling, 2017) of ISDEA+. For these models, the number of hops and number of layers are 2 on FD-2, and 3 on all other datasets to ensure fair comparison.

Since NBFNet is designed to only perform inductive link prediction with solely new nodes and utilizes trained relation embeddings, we use randomly initialized embeddings for the unseen relation types at test time to enable it for performing doubly inductive link prediction.

To run InGram (Lee et al., 2023) on PediaTypes and WikiTopics, we conduct hyperparameter search over the configurations of ranking loss margin $\gamma \in \{1.0, 2.0\}$, learning rate $\alpha \in \{0.0005, 0.001\}$, number of entity layers $L \in \{2, 3, 4\}$, and number of entity layers $\hat{L} \in \{2, 3, 4\}$. For other hyperparameters, we use the suggested values from Lee et al. (2023) and their codebase, such as the number of bins $B = 10$ and the number of attention heads $K = 8$. We then use the overall best-performing hyperparameters on PediaTypes and the best-performing hyperparameters on WikiTopics to run InGram on all tasks in PediaTypes and all tasks in WikiTopics respectively. For running on the (modified) NL-$k$, WK-$k$, and FB-$k$ datasets from Lee et al. (2023), we use the provided hyperparameters for each task from the authors.

To run DEq-InGram, we use the same trained checkpoints of InGram. The difference is at inference time, where instead of a single forward pass with one sample of randomly initialized entity and relation embeddings for InGram, we draw 10 samples of initial entity and relation embeddings and run 10 forward passes. This yields 10 Monte Carlo samples of the triplet scores, which we then use to compute the DEq-InGram triplet scores according to Equation (5).

For RMPI (Geng et al., 2023), we use the provided hyperparameters from the codebase and run the RMPI-NE version of the model with a concatenation-based fusion function, which generally has the best performance reported in Geng et al. (2023). We note that, since our knowledge graph does not contain ontological information over the unseen relation types of the test graphs, we instead provide the model with randomly initialized embeddings for the unseen relation types to perform doubly inductive link prediction.

### F.1.2 DOUBLY INDUCTIVE LINK PREDICTION OVER PEDIATYPES

As discussed in Section 5, we create our own doubly inductive link prediction benchmark dataset PediaTypes. Each graph in PediaTypes is sampled from a graph in the OpenEA library (Sun

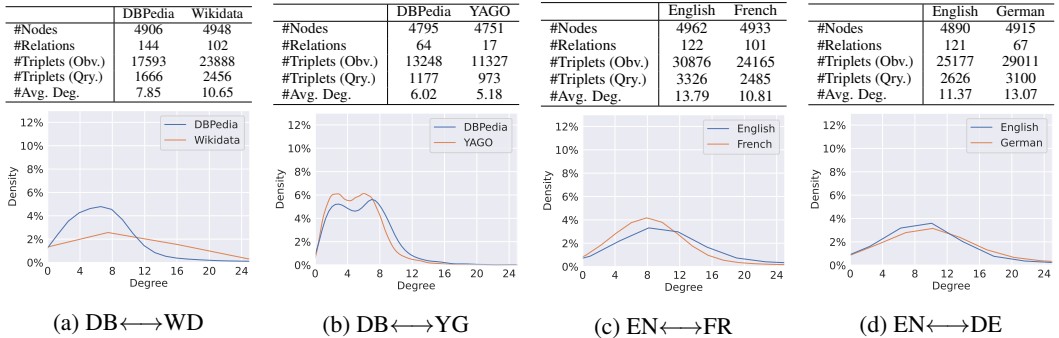

| | DBPedia | Wikidata | | DBPedia | YAGO | | English | French | | English | German |
|---|---|---|---|---|---|---|---|---|---|---|---|
| #Nodes | 4906 | 4948 | #Nodes | 4795 | 4751 | #Nodes | 4962 | 4933 | #Nodes | 4890 | 4915 |
| #Relations | 144 | 102 | #Relations | 64 | 17 | #Relations | 122 | 101 | #Relations | 121 | 67 |
| #Triplets (Obv.) | 17593 | 23888 | #Triplets (Obv.) | 13248 | 11327 | #Triplets (Obv.) | 30876 | 24165 | #Triplets (Obv.) | 25177 | 29011 |
| #Triplets (Qry.) | 1666 | 2456 | #Triplets (Qry.) | 1177 | 973 | #Triplets (Qry.) | 3326 | 2485 | #Triplets (Qry.) | 2626 | 3100 |
| #Avg. Deg. | 7.85 | 10.65 | #Avg. Deg. | 6.02 | 5.18 | #Avg. Deg. | 13.79 | 10.81 | #Avg. Deg. | 11.37 | 13.07 |

(a) DB⟷WD  (b) DB⟷YG  (c) EN⟷FR  (d) EN⟷DE

Figure 5: **Statistics of PediaTypes:** We report graph statistics including the number of nodes, number of relations, observed (obv.) triplets, querying (qry.) triplets, and average degree for each graph pair, e.g., (a) corresponds to DBPedia-and-Wikidata pair, and will be used to construct DB2WD and WD2DB tasks. We also report (in & out) degree distribution on each graph at the bottom. We omit tail distribution larger than 25 since they are too small and almost flat.

et al., 2020b) (under GPL-3.0 license). OpenEA (Sun et al., 2020b) library provides multiple pairs of knowledge graph, each pair of which is a database containing similar topics. Each node of a graph corresponds to the Universal Resource Identifier (URI) of an entity in the database, e.g., "*http://dbpedia.org/resource/E399772*" from English DBPedia. Each relation type of a graph corresponds to the URI of a relation in the database, e.g., "*http://dbpedia.org/ontology/award*" from English DBPedia. Moreover, since each pair of graphs describes similar topics, most entities and relations are highly related, e.g., "*http://dbpedia.org/resource/E678522*" from English and "*http://fr.dbpedia.org/resource/E415873*" from French are indeed the same thing, except that the labeling is different. These multilingual KGs predominantly use English for relation labels, which causes an overlap in relations. However, in our experimental setup, we treat relations as if they were in different languages and do not leverage this overlapping information during model training. Thus, we would expect a powerful model that is insensitive to node and relation type labelings to be able to learn on one graph of the pair and perform well on the other graph of the same pair.

To control the size under a feasible limitation, we use the same subgraph sampling algorithm as GraIL (Teru et al., 2020), which proposes link prediction benchmarks over solely new nodes. Details are provided in Algorithm 1. For each pair of graphs from the OpenEA library, e.g., English-to-French DBPedia, we first apply the sampling algorithm as in Algorithm 1 on each graph to reduce the size of each graph. Then we randomly split querying triplets given by the Algorithm 1 into 80% training, 10% validation, and 10% test for each graph. Finally, to construct the task where we learn on English DBPedia but test on French DBPedia (denoted as EN-FR), we pick training and validation triplets from the English graph for model tuning, and only use test triplets from the French graph for model evaluation; Similarly, for task from French to English (FR-EN), we pick training and validation triplets from French graph for model tuning, and only use test triplets from English graph for model evaluation. The **dataset statistics** for PediaTypes are summarized in Figure 5.

**Additional Results** We present the Node & Relation Hits@10 performance in the main paper. We provide more results including MRR, Hits@1, Hits@5 in Tables 2 to 4. We can see that our proposed ISDEA+ and DEq-InGram perform consistently and significantly better than the baselines in the much harder relation prediction task, showing their power to generalize to both new nodes and new relations. The structural double equivariant model ISDEA+ performs worse on node prediction over some datasets, which might be due to the node GNN implementation of ISDEA+. These tasks do not care much about the actual relation type as we can see from the superior performance of homogeneous GNNs on node prediction. So the additional equivariance over relations and the training loss over both negative nodes and negative relations might cause the model to focus more on the relation prediction task, while the double equivariant structural representation might hurt the performance of missing node prediction (Srinivasan & Ribeiro, 2020).

But it is important to note that the structural double equivariant ISDEA+ model excels on relation prediction and achieves much better results on Hits@1 and Hits@5 as shown in Tables 3 and 4. The performance of baseline models that is lower than random is probably because the knowledge

**Algorithm 1 Sampling Algorithm for PediaTypes.** This is a subgraph sampling code for a single graph (either training or test). It will reduce the large original graph into a connected graph of the required size.

**Require:** Raw graph triplets $\mathcal{S}^{\text{raw}}$, Raw graph node set $\mathcal{V}^{\text{raw}}$, Raw graph relation set $\mathcal{R}^{\text{raw}}$, Maximum number of nodes $N$, Maximum number of edges $M$, Maximum node degree $D$.
**Ensure:** Subgraph triplets $\mathcal{S}^{\text{sub}}$
1: $\mathcal{S}^{\text{sub}} \leftarrow \emptyset$
2: $\mathcal{V}^{\text{sub}} \leftarrow \emptyset$
3: $\mathcal{R}^{\text{sub}} \leftarrow \emptyset$
4: Create an empty queue $Q$.
5: Get the node $v_0$ with the highest degree in the raw graph.
6: $Q$.add($v_0$)
7: $\mathcal{V}^{\text{sub}} \leftarrow \mathcal{V}^{\text{sub}} \cup \{v_0\}$
8: **while** $|Q| > 0$ **do**
9:     $u \leftarrow Q$.pop()
10:     **if** $|\mathcal{V}^{\text{sub}}| \geq N$ or $|\mathcal{V}^{\text{sub}}| \geq M$ **then**
11:         **continue**
12:     **end if**
13:     $\mathcal{B} = \{(v, r, u)|(r, v) \in \mathcal{R}^{\text{raw}} \times \mathcal{V}^{\text{raw}}\} \cup \{(u, r, v)|(r, v) \in \mathcal{R}^{\text{raw}} \times \mathcal{V}^{\text{raw}}\}$
14:     **if** $|\mathcal{B}| > D$ **then**
15:         Uniformly select $D$ triplets from $\mathcal{B}$ as $\mathcal{B}'$
16:     **else**
17:         $\mathcal{B}' \leftarrow \mathcal{B}$
18:     **end if**
19:     **for** $(i, r, j) \in \mathcal{B}'$ **do**
20:         **if** $i = u$ **then**
21:             $Q$.add($j$)
22:             $\mathcal{V}^{\text{sub}} \leftarrow \mathcal{V}^{\text{sub}} \cup \{j\}$
23:         **else**
24:             $Q$.add($i$)
25:             $\mathcal{V}^{\text{sub}} \leftarrow \mathcal{V}^{\text{sub}} \cup \{i\}$
26:         **end if**
27:         $\mathcal{S}^{\text{sub}} \leftarrow \mathcal{S}^{\text{sub}} \cup \{(i, r, j)\}$
28:     **end for**
29: **end while**

Table 2: **Relation & Node MRR performance on Doubly Inductive Link Prediction over Pedi-aTypes.** We report standard deviations over 5 runs. A higher value means better doubly inductive link prediction performance. "Rand" column contains unbiased estimations of the performance from a random predictor. **Both ISDEA+ and DEq-InGram consistently achieve better results than the baselines.** N/A*: Not available due to constant crashes.

(a) **Relation prediction** $(i, ?, j)$ **performance in %. Higher ↑ is better.**

| Models | EN-FR | FR-EN | EN-DE | DE-EN | DB-WD | WD-DB | DB-YG | YG-DB |
|---|---|---|---|---|---|---|---|---|
| Rand | $8.86_{\pm00.00}$ | $8.86_{\pm00.00}$ | $8.86_{\pm00.00}$ | $8.86_{\pm00.00}$ | $8.86_{\pm00.00}$ | $8.86_{\pm00.00}$ | $8.86_{\pm00.00}$ | $8.86_{\pm00.00}$ |
| GAT | $8.04_{\pm00.25}$ | $7.93_{\pm00.04}$ | $8.17_{\pm00.08}$ | $8.12_{\pm00.09}$ | $8.06_{\pm00.15}$ | $7.90_{\pm00.12}$ | $8.12_{\pm00.21}$ | $8.17_{\pm00.16}$ |
| GIN | $8.07_{\pm00.09}$ | $8.09_{\pm00.05}$ | $8.07_{\pm00.13}$ | $8.07_{\pm00.11}$ | $8.03_{\pm00.20}$ | $7.97_{\pm00.30}$ | $7.82_{\pm00.27}$ | $7.84_{\pm00.14}$ |
| GraphConv | $7.92_{\pm00.16}$ | $7.97_{\pm00.12}$ | $8.07_{\pm00.15}$ | $8.03_{\pm00.05}$ | $8.14_{\pm00.04}$ | $7.98_{\pm00.18}$ | $8.04_{\pm00.24}$ | $7.84_{\pm00.13}$ |
| NBFNet | $10.25_{\pm01.24}$ | $9.53_{\pm00.85}$ | $8.15_{\pm01.21}$ | $4.32_{\pm00.26}$ | $10.33_{\pm02.45}$ | $8.97_{\pm01.24}$ | $9.29_{\pm01.38}$ | $14.54_{\pm04.76}$ |
| RMPI | $12.45_{\pm01.90}$ | $12.10_{\pm02.71}$ | $11.69_{\pm04.37}$ | $10.28_{\pm01.28}$ | N/A* | $8.54_{\pm02.70}$ | $17.89_{\pm12.22}$ | $6.53_{\pm02.16}$ |
| InGram | $50.03_{\pm05.32}$ | $26.31_{\pm08.27}$ | $21.32_{\pm07.84}$ | $29.81_{\pm14.21}$ | $48.70_{\pm10.06}$ | $38.81_{\pm03.10}$ | $29.94_{\pm13.28}$ | $32.26_{\pm13.97}$ |
| DEq-InGram (Ours) | $\mathbf{73.38}_{\pm05.77}$ | $\underline{41.61}_{\pm10.12}$ | $\underline{46.86}_{\pm09.11}$ | $\underline{40.56}_{\pm14.80}$ | $\mathbf{80.74}_{\pm04.47}$ | $\underline{66.06}_{\pm02.91}$ | $\underline{39.51}_{\pm16.76}$ | $\underline{49.10}_{\pm05.43}$ |
| ISDEA+ (Ours) | $\underline{72.96}_{\pm00.77}$ | $\mathbf{65.73}_{\pm00.58}$ | $\mathbf{59.95}_{\pm03.91}$ | $\mathbf{84.71}_{\pm01.11}$ | $\underline{71.47}_{\pm00.31}$ | $\mathbf{71.47}_{\pm00.69}$ | $\mathbf{66.48}_{\pm06.75}$ | $\mathbf{67.36}_{\pm00.43}$ |

(b) **Node prediction** $(i, k, ?)$ **performance in %. Higher ↑ is better.**

| Models | EN-FR | FR-EN | EN-DE | DE-EN | DB-WD | WD-DB | DB-YG | YG-DB |
|---|---|---|---|---|---|---|---|---|
| Rand | $8.86_{\pm00.00}$ | $8.86_{\pm00.00}$ | $8.86_{\pm00.00}$ | $8.86_{\pm00.00}$ | $8.86_{\pm00.00}$ | $8.86_{\pm00.00}$ | $8.86_{\pm00.00}$ | $8.86_{\pm00.00}$ |
| GAT | $51.43_{\pm00.25}$ | $49.48_{\pm01.51}$ | $26.22_{\pm00.44}$ | $25.45_{\pm01.23}$ | $16.87_{\pm00.59}$ | $34.66_{\pm00.33}$ | $37.22_{\pm00.29}$ | $45.96_{\pm00.29}$ |
| GIN | $53.72_{\pm03.45}$ | $52.03_{\pm03.38}$ | $34.60_{\pm07.43}$ | $37.27_{\pm09.42}$ | $20.75_{\pm07.22}$ | $40.37_{\pm08.20}$ | $35.80_{\pm01.36}$ | $44.77_{\pm00.92}$ |
| GraphConv | $63.72_{\pm01.76}$ | $57.77_{\pm01.09}$ | $48.18_{\pm00.96}$ | $45.18_{\pm00.15}$ | $22.49_{\pm00.76}$ | $50.30_{\pm02.80}$ | $38.71_{\pm00.55}$ | $50.54_{\pm00.42}$ |
| NBFNet | $69.22_{\pm02.44}$ | $\mathbf{74.01}_{\pm01.41}$ | $63.49_{\pm02.44}$ | $38.86_{\pm02.55}$ | $41.26_{\pm02.58}$ | $64.02_{\pm01.25}$ | $38.13_{\pm01.11}$ | $52.30_{\pm02.09}$ |
| RMPI | $63.02_{\pm02.94}$ | $43.72_{\pm05.65}$ | $44.82_{\pm02.93}$ | $46.84_{\pm05.36}$ | N/A* | $46.33_{\pm08.76}$ | $\mathbf{43.00}_{\pm03.70}$ | $\underline{53.72}_{\pm01.84}$ |
| InGram | $71.23_{\pm01.73}$ | $55.67_{\pm05.65}$ | $55.94_{\pm02.76}$ | $61.15_{\pm01.42}$ | $34.50_{\pm08.47}$ | $57.05_{\pm03.73}$ | $26.36_{\pm04.73}$ | $\mathbf{56.23}_{\pm01.56}$ |
| DEq-InGram (Ours) | $\mathbf{78.45}_{\pm00.89}$ | $\underline{68.59}_{\pm04.30}$ | $66.13_{\pm01.48}$ | $\mathbf{70.32}_{\pm01.58}$ | $44.71_{\pm08.98}$ | $\underline{69.23}_{\pm02.53}$ | $\underline{35.67}_{\pm03.92}$ | $48.07_{\pm08.76}$ |
| ISDEA+ (Ours) | $\underline{74.95}_{\pm01.56}$ | $57.17_{\pm01.70}$ | $\mathbf{74.38}_{\pm00.66}$ | $\underline{62.62}_{\pm00.22}$ | $\mathbf{63.21}_{\pm00.69}$ | $\mathbf{70.58}_{\pm00.42}$ | $34.79_{\pm00.49}$ | $40.71_{\pm01.75}$ |

they learn from one dataset is not able to correctly transform to another dataset, while our double equivariant model architecture is able to perform this hard doubly inductive link prediction over both new nodes and new relation types. We also note that in the Hits@1 and Hits@5 Tables 3 and 4, there are cases where DEq-InGram has higher variances than the original InGram while achieving much better average performance. This is because due to the random initialization, InGram performs poorly on the much harder Hits@1 and Hits@5 performance compared to Hits@10. In some seeds of the runs, DEq-InGram successfully improves the performance of InGram, but there are still seeds of runs that DEq-InGram still performs similar to InGram. Thus, it results in DEq-InGram having much better average results while also with higher standard deviations.

### F.1.3 WIKITOPICS: TESTING SELF-SUPERVISED PRE-TRAINED ZERO-SHOT META-LEARNING CAPABILITIES

As discussed in Section 5.2, the WikiTopics dataset is created from the WikiData-5M (Wang et al., 2021b) (under CC0 1.0 license). Each node in the graphs of this dataset represents an entity described by an existing Wikipedia page, and each relation type corresponds to a particular relation between the entities, such as "director of" or "designed by". The node and relation type indices are codenames that start with the prefix "Q" and "P" respectively, which are devoid of semantic meaning. Nevertheless, WikiData-5M (Wang et al., 2021b) provides aliases for all nodes and relation types that map their indices to textual descriptions, and we use these textual descriptions to group the relation types into 11 different topic groups, or domains (we do not however provide these textual descriptions to the models per the specification of the doubly inductive link prediction task). In total, WikiData-5M (Wang et al., 2021b) contains 822 relation types. We create WikiTopics datasets from all 822 relation types, which comprise graphs with as many as 66 relation types. Each graph has a disjoint set of relation types from all other graphs. Table 5 shows the 11 topics/domains of the WikiTopics dataset, each corresponding to a distinct KG with distinct relation types.

To control the overall size of the graphs in WikiTopics, we downsample $10,000$ nodes for each domain from the subgraph consisting of only the triplets with the relation types belonging to that domain. We adopt the Forest Fire sampling procedure with burning probability $p = 0.8$ (Leskovec & Faloutsos, 2006) implemented in the Little Ball of Fur Python package (Rozemberczki et al., 2020). We then split the downsampled domain KG into 90% observable triplets and 10% querying triplets to

Table 3: **Relation & Node Hits@1 performance on Doubly Inductive Link Prediction over PediaTypes.** We report standard deviations over 5 runs. A higher value means better doubly inductive link prediction performance. "Rand" column contains unbiased estimations of the performance from a random predictor. **Both ISDEA+ and DEq-InGram consistently achieve better results than the baselines.** N/A*: Not available due to constant crashes.

(a) **Relation prediction** $(i, ?, j)$ **performance in %. Higher ↑ is better.**

| Models | EN-FR | FR-EN | EN-DE | DE-EN | DB-WD | WD-DB | DB-YG | YG-DB |
|---|---|---|---|---|---|---|---|---|
| Rand | $1.96_{\pm00.00}$ | $1.96_{\pm00.00}$ | $1.96_{\pm00.00}$ | $1.96_{\pm00.00}$ | $1.96_{\pm00.00}$ | $1.96_{\pm00.00}$ | $1.96_{\pm00.00}$ | $1.96_{\pm00.00}$ |
| GAT | $1.07_{\pm00.14}$ | $1.01_{\pm00.01}$ | $1.03_{\pm00.03}$ | $1.11_{\pm00.09}$ | $1.07_{\pm00.14}$ | $0.99_{\pm00.21}$ | $0.96_{\pm00.16}$ | $1.09_{\pm00.25}$ |
| GIN | $1.01_{\pm00.03}$ | $0.95_{\pm00.08}$ | $1.03_{\pm00.06}$ | $1.10_{\pm00.06}$ | $0.96_{\pm00.15}$ | $1.00_{\pm00.15}$ | $0.92_{\pm00.15}$ | $0.83_{\pm00.17}$ |
| GraphConv | $0.91_{\pm00.03}$ | $0.97_{\pm00.06}$ | $1.05_{\pm00.14}$ | $1.01_{\pm00.03}$ | $1.09_{\pm00.07}$ | $0.91_{\pm00.04}$ | $0.94_{\pm00.22}$ | $0.88_{\pm00.20}$ |
| NBFNet | $4.43_{\pm01.24}$ | $3.62_{\pm01.01}$ | $2.49_{\pm01.23}$ | $0.51_{\pm00.18}$ | $4.18_{\pm02.17}$ | $2.80_{\pm00.83}$ | $1.63_{\pm00.89}$ | $7.30_{\pm05.01}$ |
| RMPI | $3.92_{\pm02.08}$ | $4.04_{\pm01.83}$ | $3.37_{\pm02.20}$ | $2.13_{\pm00.79}$ | N/A* | $2.39_{\pm02.35}$ | $7.36_{\pm09.03}$ | $0.91_{\pm00.92}$ |
| InGram | $35.19_{\pm07.73}$ | $12.40_{\pm07.55}$ | $8.45_{\pm06.57}$ | $16.46_{\pm16.33}$ | $33.66_{\pm12.09}$ | $25.69_{\pm03.88}$ | $14.24_{\pm12.00}$ | $15.83_{\pm12.59}$ |
| DEq-InGram (Ours) | $\mathbf{65.26}_{\pm10.23}$ | $\underline{26.90}_{\pm12.97}$ | $\underline{36.80}_{\pm11.16}$ | $\underline{25.34}_{\pm18.48}$ | $\mathbf{75.00}_{\pm06.42}$ | $\mathbf{60.35}_{\pm02.56}$ | $\underline{24.28}_{\pm14.29}$ | $\underline{30.82}_{\pm10.43}$ |
| ISDEA+ (Ours) | $\underline{58.43}_{\pm01.29}$ | $\mathbf{48.68}_{\pm00.96}$ | $\mathbf{37.29}_{\pm05.11}$ | $\mathbf{75.08}_{\pm01.99}$ | $\underline{57.05}_{\pm00.92}$ | $\underline{56.00}_{\pm01.17}$ | $\mathbf{59.36}_{\pm07.96}$ | $\mathbf{49.41}_{\pm00.85}$ |

(b) **Node prediction** $(i, k, ?)$ **performance in %. Higher ↑ is better.**

| Models | EN-FR | FR-EN | EN-DE | DE-EN | DB-WD | WD-DB | DB-YG | YG-DB |
|---|---|---|---|---|---|---|---|---|
| Rand | $1.96_{\pm00.00}$ | $1.96_{\pm00.00}$ | $1.96_{\pm00.00}$ | $1.96_{\pm00.00}$ | $1.96_{\pm00.00}$ | $1.96_{\pm00.00}$ | $1.96_{\pm00.00}$ | $1.96_{\pm00.00}$ |
| GAT | $31.80_{\pm00.64}$ | $30.19_{\pm02.30}$ | $10.23_{\pm00.96}$ | $8.68_{\pm01.69}$ | $7.98_{\pm00.89}$ | $16.26_{\pm00.34}$ | $26.09_{\pm00.47}$ | $33.06_{\pm00.29}$ |
| GIN | $34.59_{\pm04.64}$ | $34.57_{\pm05.26}$ | $17.69_{\pm07.91}$ | $20.74_{\pm10.01}$ | $12.42_{\pm06.59}$ | $23.10_{\pm09.67}$ | $23.72_{\pm01.62}$ | $32.26_{\pm01.89}$ |
| GraphConv | $47.48_{\pm02.60}$ | $40.37_{\pm01.52}$ | $31.96_{\pm01.02}$ | $28.46_{\pm00.13}$ | $12.53_{\pm00.34}$ | $24.12_{\pm00.80}$ | $37.05_{\pm00.51}$ |  |
| NBFNet | $\underline{64.17}_{\pm02.68}$ | $\mathbf{69.68}_{\pm01.63}$ | $\underline{57.50}_{\pm02.66}$ | $32.26_{\pm02.81}$ | $\underline{34.56}_{\pm02.54}$ | $\underline{59.70}_{\pm01.38}$ | $\mathbf{33.32}_{\pm01.11}$ | $\mathbf{47.47}_{\pm02.08}$ |
| RMPI | $48.27_{\pm03.74}$ | $26.92_{\pm04.87}$ | $27.38_{\pm03.09}$ | $29.60_{\pm04.77}$ | N/A* | $34.81_{\pm08.97}$ | $\underline{33.29}_{\pm03.20}$ | $42.14_{\pm02.87}$ |
| InGram | $60.00_{\pm02.06}$ | $41.59_{\pm06.37}$ | $39.05_{\pm02.99}$ | $\underline{45.44}_{\pm01.69}$ | $22.06_{\pm08.10}$ | $42.54_{\pm04.50}$ | $13.47_{\pm03.50}$ | $20.09_{\pm04.96}$ |
| DEq-InGram (Ours) | $\mathbf{69.46}_{\pm01.12}$ | $\underline{57.65}_{\pm05.54}$ | $51.93_{\pm01.88}$ | $\mathbf{57.06}_{\pm01.96}$ | $32.12_{\pm09.51}$ | $57.84_{\pm03.28}$ | $20.49_{\pm03.35}$ | $33.01_{\pm08.87}$ |
| ISDEA+ (Ours) | $62.17_{\pm02.38}$ | $44.67_{\pm01.92}$ | $\mathbf{63.36}_{\pm00.78}$ | $47.78_{\pm00.48}$ | $\mathbf{51.76}_{\pm00.86}$ | $\mathbf{60.15}_{\pm00.69}$ | $20.74_{\pm00.28}$ | $26.24_{\pm02.03}$ |

Table 4: **Relation & Node Hits@5 performance on Doubly Inductive Link Prediction over PediaTypes.** We report standard deviations over 5 runs. A higher value means better doubly inductive link prediction performance. "Rand" column contains unbiased estimations of the performance from a random predictor. **Both ISDEA+ and DEq-InGram consistently achieve better results than the baselines.** N/A*: Not available due to constant crashes.

(a) **Relation prediction** $(i, ?, j)$ **performance in %. Higher ↑ is better.**

| Models | EN-FR | FR-EN | EN-DE | DE-EN | DB-WD | WD-DB | DB-YG | YG-DB |
|---|---|---|---|---|---|---|---|---|
| Rand | $9.80_{\pm00.00}$ | $9.80_{\pm00.00}$ | $9.80_{\pm00.00}$ | $9.80_{\pm00.00}$ | $9.80_{\pm00.00}$ | $9.80_{\pm00.00}$ | $9.80_{\pm00.00}$ | $9.80_{\pm00.00}$ |
| GAT | $9.08_{\pm00.39}$ | $8.63_{\pm00.25}$ | $9.47_{\pm00.18}$ | $9.20_{\pm00.24}$ | $8.95_{\pm00.36}$ | $8.63_{\pm00.29}$ | $9.58_{\pm00.50}$ | $9.16_{\pm00.23}$ |
| GIN | $9.09_{\pm00.16}$ | $9.31_{\pm00.15}$ | $9.18_{\pm00.28}$ | $9.23_{\pm00.34}$ | $9.12_{\pm00.12}$ | $8.85_{\pm00.56}$ | $8.53_{\pm00.66}$ | $8.61_{\pm00.34}$ |
| GraphConv | $8.97_{\pm00.66}$ | $8.74_{\pm00.26}$ | $9.23_{\pm00.11}$ | $8.82_{\pm00.10}$ | $9.17_{\pm00.29}$ | $9.11_{\pm00.50}$ | $9.01_{\pm00.72}$ | $8.73_{\pm00.15}$ |
| NBFNet | $12.94_{\pm01.77}$ | $12.46_{\pm01.40}$ | $8.56_{\pm01.67}$ | $2.68_{\pm00.72}$ | $13.44_{\pm04.02}$ | $11.74_{\pm03.02}$ | $11.95_{\pm03.78}$ | $20.37_{\pm05.90}$ |
| RMPI | $16.39_{\pm04.15}$ | $15.76_{\pm04.58}$ | $15.86_{\pm08.05}$ | $12.56_{\pm02.70}$ | N/A* | $8.91_{\pm03.51}$ | $24.25_{\pm19.24}$ | $4.98_{\pm03.08}$ |
| InGram | $67.15_{\pm05.04}$ | $37.86_{\pm14.41}$ | $30.99_{\pm11.82}$ | $40.00_{\pm13.02}$ | $65.80_{\pm09.59}$ | $51.66_{\pm03.57}$ | $43.27_{\pm19.30}$ | $51.54_{\pm26.09}$ |
| DEq-InGram (Ours) | $\underline{83.23}_{\pm05.64}$ | $\underline{59.83}_{\pm11.57}$ | $\underline{54.30}_{\pm08.25}$ | $\underline{57.65}_{\pm15.74}$ | $\underline{87.08}_{\pm02.55}$ | $\underline{70.79}_{\pm03.80}$ | $\underline{51.45}_{\pm29.14}$ | $\underline{75.85}_{\pm07.26}$ |
| ISDEA+ (Ours) | $\mathbf{93.79}_{\pm00.20}$ | $\mathbf{91.21}_{\pm00.25}$ | $\mathbf{94.04}_{\pm01.21}$ | $\mathbf{96.64}_{\pm00.13}$ | $\mathbf{90.83}_{\pm02.29}$ | $\mathbf{93.97}_{\pm00.45}$ | $\mathbf{74.27}_{\pm06.29}$ | $\mathbf{92.75}_{\pm00.40}$ |

(b) **Node prediction** $(i, k, ?)$ **performance in %. Higher ↑ is better.**

| Models | EN-FR | FR-EN | EN-DE | DE-EN | DB-WD | WD-DB | DB-YG | YG-DB |
|---|---|---|---|---|---|---|---|---|
| Rand | $9.80_{\pm00.00}$ | $9.80_{\pm00.00}$ | $9.80_{\pm00.00}$ | $9.80_{\pm00.00}$ | $9.80_{\pm00.00}$ | $9.80_{\pm00.00}$ | $9.80_{\pm00.00}$ | $9.80_{\pm00.00}$ |
| GAT | $78.49_{\pm00.44}$ | $74.70_{\pm00.68}$ | $42.17_{\pm00.91}$ | $42.39_{\pm00.52}$ | $20.96_{\pm00.65}$ | $57.26_{\pm00.89}$ | $46.92_{\pm00.37}$ | $59.20_{\pm00.41}$ |
| GIN | $79.96_{\pm01.88}$ | $74.33_{\pm01.16}$ | $53.97_{\pm07.61}$ | $55.89_{\pm10.06}$ | $25.05_{\pm09.23}$ | $61.94_{\pm06.71}$ | $46.56_{\pm01.37}$ | $57.48_{\pm00.35}$ |
| GraphConv | $85.21_{\pm00.63}$ | $80.67_{\pm00.30}$ | $67.76_{\pm01.19}$ | $64.97_{\pm00.43}$ | $28.37_{\pm01.41}$ | $67.36_{\pm02.37}$ | $\underline{53.79}_{\pm00.72}$ | $64.13_{\pm00.23}$ |
| NBFNet | $81.48_{\pm02.24}$ | $\mathbf{85.15}_{\pm01.06}$ | $77.62_{\pm02.41}$ | $48.73_{\pm02.59}$ | $51.52_{\pm03.21}$ | $72.18_{\pm00.90}$ | $44.01_{\pm01.40}$ | $60.34_{\pm02.28}$ |
| RMPI | $82.47_{\pm02.25}$ | $64.88_{\pm07.62}$ | $67.24_{\pm04.38}$ | $69.47_{\pm06.60}$ | N/A* | $60.11_{\pm08.77}$ | $51.57_{\pm05.03}$ | $\mathbf{66.67}_{\pm01.28}$ |
| InGram | $85.15_{\pm01.74}$ | $72.32_{\pm05.31}$ | $78.84_{\pm02.86}$ | $81.01_{\pm00.97}$ | $45.96_{\pm11.09}$ | $74.88_{\pm03.09}$ | $37.49_{\pm06.84}$ | $50.66_{\pm06.76}$ |
| DEq-InGram (Ours) | $\underline{89.62}_{\pm00.63}$ | $\underline{81.54}_{\pm02.82}$ | $\underline{84.57}_{\pm00.95}$ | $\mathbf{87.16}_{\pm01.04}$ | $\underline{57.44}_{\pm09.14}$ | $\mathbf{83.14}_{\pm01.64}$ | $\mathbf{51.77}_{\pm05.14}$ | $\underline{65.33}_{\pm09.57}$ |
| ISDEA+ (Ours) | $\mathbf{92.45}_{\pm00.73}$ | $71.24_{\pm02.13}$ | $\mathbf{89.98}_{\pm00.96}$ | $\underline{82.65}_{\pm00.79}$ | $\mathbf{76.12}_{\pm00.87}$ | $\underline{83.33}_{\pm00.46}$ | $48.04_{\pm01.78}$ | $57.92_{\pm01.47}$ |

be predicted by the models. When splitting, we ensure that the set of nodes in the querying triplets is a subset of those in the observable triplets. This way, the model is not tasked with the impossible task of predicting relation types between orphaned nodes previously unseen in the observable part of the graph. This is implemented via an iterative procedure, where we first sample a batch of missing

Table 5: The 11 different topics/domains of the WikiTopics dataset.

| Domain KG index | Abbreviation | Description |
|---|---|---|
| T1 | Art | Art and Media Representation |
| T2 | Award | Award Nomination and Achievement |
| T3 | Edu | Education and Academia |
| T4 | Health | Health, Medicine, and Genetics |
| T5 | Infra | Infrastructure and Transportation |
| T6 | Loc | Location and Administrative Entity |
| T7 | Org | Organization and Membership |
| T8 | People | People and Social Relationship |
| T9 | Science | Science, Technology, and Language |
| T10 | Sport | Sport, and Game Competition |
| T11 | Tax | Taxonomy and Biology |

| | #Nodes | # Relations | #Triplets (Obv.) | #Triplets (Qry.) | Avg. Deg. |
|---|---|---|---|---|---|
| Art | 10000 | 45 | 28023 | 3113 | 6.23 |
| Award | 10000 | 10 | 25056 | 2783 | 5.57 |
| Edu | 10000 | 15 | 14193 | 1575 | 3.15 |
| Health | 10000 | 20 | 15337 | 1703 | 3.41 |
| Infra | 10000 | 27 | 21646 | 2405 | 4.81 |
| Loc | 10000 | 35 | 80269 | 8918 | 17.84 |
| Org | 10000 | 18 | 30214 | 3357 | 6.71 |
| People | 10000 | 25 | 58530 | 6503 | 13.01 |
| Sci | 10000 | 42 | 12516 | 1388 | 2.78 |
| Sport | 10000 | 20 | 46717 | 5190 | 10.38 |
| Tax | 10000 | 31 | 19416 | 2157 | 4.32 |

Figure 6: **Statistics of WikiTopics:** We report graph statistics including the number of nodes, number of relations, observed (obv.) triplets, querying (qry.) triplets, and average degree for each graph. We also report (in & out) degree distribution on each graph at the bottom. We omit tail distribution larger than 35 since they are fairly small and almost flat.

triplets from the downsampled domain graph, then discard those that contain unseen nodes in the rest of the triplets, and repeat this process until the number of sampled triplets reaches 10% of total triplets. Figure 6 shows the **data statistics** of WikiTopics dataset.

**Self-supervised pre-trained zero-shot meta-learning tasks over WikiTopics to train on a KG of one domain and test on a KG of a completely unseen test domain**   We train the models on each of the 11 graphs for 5 random seeds, and for each trained model checkpoint, we cross-test it on all the other 10 graphs, resulting in a total of 550 statistics. We report the mean results across random seeds in heatmaps. We present a detailed results (heatmaps with values) of Node and Relation

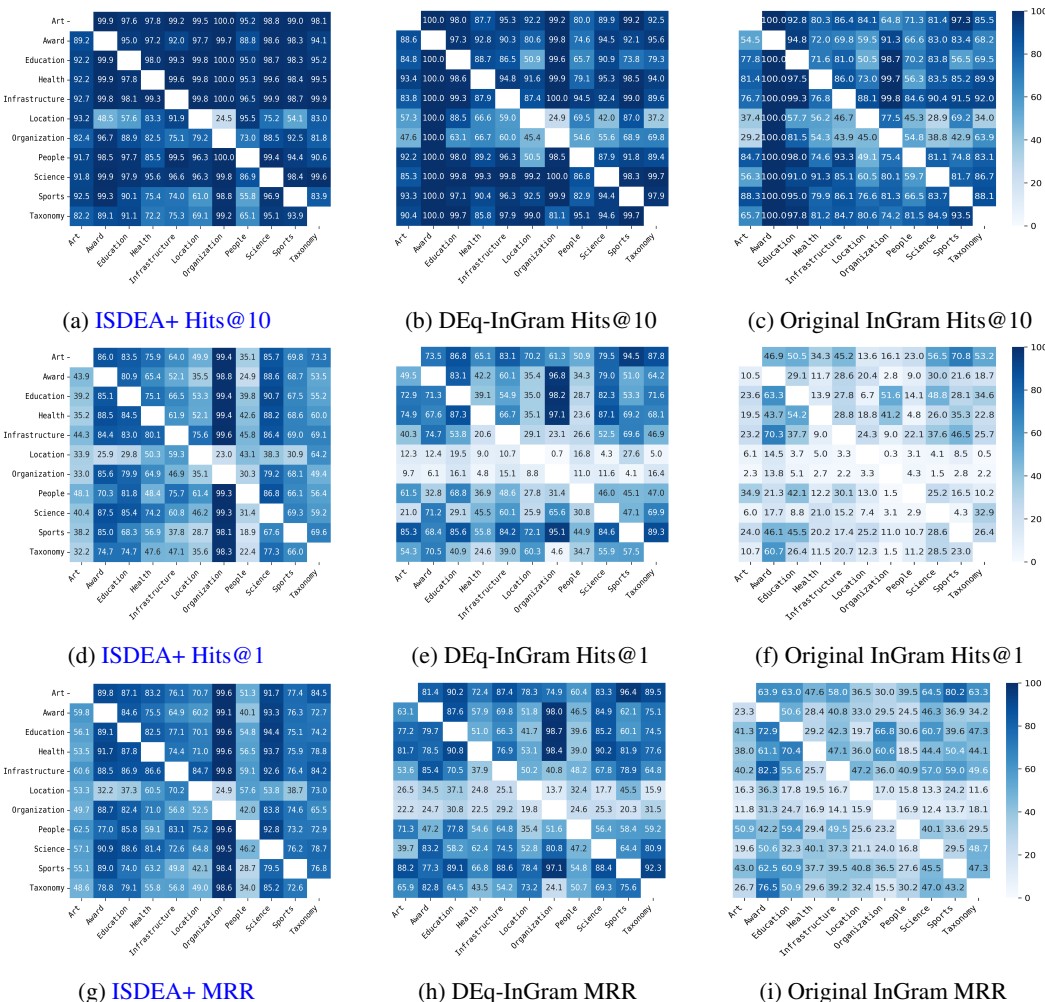

(a) ISDEA+ Hits@10  (b) DEq-InGram Hits@10  (c) Original InGram Hits@10

(d) ISDEA+ Hits@1  (e) DEq-InGram Hits@1  (f) Original InGram Hits@1

(g) ISDEA+ MRR  (h) DEq-InGram MRR  (i) Original InGram MRR

Figure 7: **Relation prediction** $(i, ?, j)$ **performance over WikiTopics** for ISDEA+, DEq-InGram, and InGram (Lee et al., 2023). Each row within each heatmap corresponds to a training graph, and each column within each heatmap corresponds to a test graph. A darker color means better performance. **Both ISDEA, DEq-InGram perform significantly better than InGram, especially for Hits@1 and MRR, whereas ISDEA+ exhibits more consistent results across different train-test scenarios than both DEq-InGram and InGram.**

Hits@10, Hits@1, and MRR for WikiTopics in Figures 7 and 8. Due to the large number of runs ($11 \times 10 = 110$ different train-test scenarios, each with 5 random seeds, resulting in a total of 550 runs) and the time constraints to run all baseline models, we perform the evaluation over only the three models (ISDEA+, DEq-InGram, and InGram) that are designed for our doubly inductive link prediction task. Figure 7 shows that for the task of predicting missing relation types $(i, ?, j)$, ISDEA+ and DEq-InGram are consistently better than InGram across all different metrics. Especially, the structural double equivariant ISDEA+ model exhibits more consistent results across different train-test scenarios than both DEq-InGram and InGram, and achieves significantly better results in Hits@1 and MRR, showcasing its ability for doubly inductive link prediction in a much harder evaluation scenario. For the task of prediction missing nodes $(i, k, ?)$ as shown in Figure 8, ISDEA+, DEq-InGram, and InGram showcase comparable performance, whereas ISDEA+ exhibits more consistent results across different train-test scenarios than both DEq-InGram and InGram. We also note that similar to the relation prediction task, ISDEA+ also exhibits the best performance in the Hits@1 metric for the node prediction task.

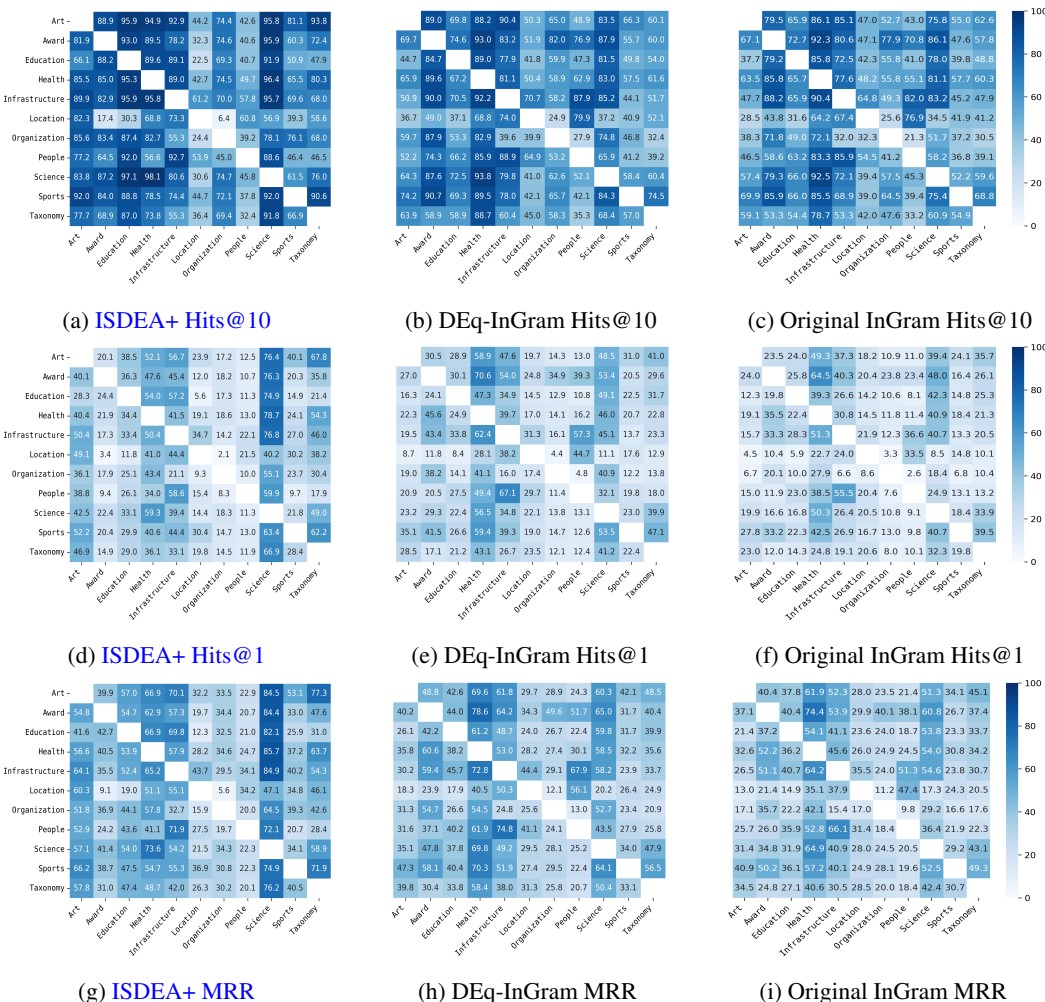

Figure 8: **Node prediction** $(i, k, ?)$ **performance over WikiTopics** for ISDEA+, DEq-InGram, and InGram (Lee et al., 2023). Each row within each heatmap corresponds to a training graph, and each column within each heatmap corresponds to a test graph. A darker color means better performance. **ISDEA+, DEq-InGram, and InGram showcase comparable performance in general, and ISDEA+ exhibits the best performance on Hits@1 in particular.**

**Self-supervised pre-trained zero-shot meta-learning tasks over WikiTopics to train on combined KGs with an increasing number of distinct domains and test on KGs with completely unseen test domains** In Section 5.2, we describe an experiment conducted with WikiTopics, focusing on the zero-shot meta-learning tasks and the effect of varying the number of training KGs. For this experiment, we selected 8 out of the 11 domains from WikiTopics. In each experimental setup, we randomly select $1, 2, 3,$ or $4$ KGs with distinct domains for training. The model's performance is then evaluated on each of the remaining KGs for testing. We repeat this process to collect performance across various combinations, enabling us to calculate an average performance gain when training on an increasing number of KGs. In addition, we compare the inductive performance of our ISDEA+ against the best-performing transductive model, which is trained and tested on the KG of the same domain. This approach helps us analyze whether ISDEA+, via meta-learning on a more diverse set of training graphs, can match or even surpass the transductive model in performance.

To select the best transductive performance, we train two models: ISDEA+ and SOTA relational GNN model NBFNet (Zhu et al., 2021) (with its original implementation) and test on graphs within the same domain, and then select the best performance from these two models as the *transductive SOTA* performance. We experimented with a total of 5 setups, and for each setup, we trained models with 3 different random seeds for each training combination.

Table 6 shows training and test domains sampled for each setup, and Tables 7 and 8 shows the Hits@10 and Hits@5 results on each test domain respectively. The boxplots shown in Figure 2 are the aggregated results of Tables 7 and 8. Similar to Figure 2, we can observe that the performance of ISDEA+ consistently increases when training on more and more domains, and finally reaching or even surpassing the performance of the best-performing transductive model on certain test domains.

Table 6: The 5 randomly sampled setups of training and test combinations of WikiTopic domain KGs (Table 5) that are used in our zero-shot meta-learning experiment. The $+$ between different domains denotes that the KGs corresponding to these domains are merged for model training.

| Setup | 1 Training Domain | 2 Training Domains | 3 Training Domains | 4 Training Domains | Test Domains |
|-------|-------------------|--------------------|--------------------|--------------------|--------------|
| 1 | Infra | Infra + Sci | Infra + Sci + Sport | Infra + Sci + Sport + Tax | Art, Award, Edu, Health |
| 2 | Award | Award + Edu | Award + Edu + Tax | Award + Edu + Tax + Sport | Art, Health, Infra, Sci |
| 3 | Sport | Sport + Infra | Sport + Infra + Edu | Sport + Infra + Edu + Health | Art, Award, Sci, Tax |
| 4 | Art | Art + Award | Art + Award + Health | Art + Award + Health + Infra | Edu, Sci, Sport, Tax |
| 5 | Health | Health + Sport | Health + Sport + Tax | Health + Sport + Tax + Art | Award, Edu, Infra, Sci |

Table 7: **Average Relation & Node Hits@10 performance of ISDEA+ on the Zero-Shot Meta-Learning task over WikiTopics on each test domain.** We also report the performance of the best transductive model. **ISDEA+ consistently achieves better and better performance when training on more and more domains, closing and even sometimes surpassing the best transductive model performance.**

(a) **Relation prediction $(i, ?, j)$ performance in %. Higher ↑ is better.**

| Training Scenario | Test on Art | Test on Award | Test on Edu | Test on Health | Test on Infra | Test on Sci | Test on Sport | Test on Tax |
|-------------------|-------------|---------------|-------------|----------------|---------------|-------------|---------------|-------------|
| Best Transductive | $95.37_{\pm00.54}$ | $99.95_{\pm00.02}$ | $98.33_{\pm00.11}$ | $99.61_{\pm00.10}$ | $99.83_{\pm00.03}$ | $99.64_{\pm00.16}$ | $94.40_{\pm07.55}$ | $87.11_{\pm16.02}$ |
| 1 training domain | $92.31_{\pm02.15}$ | $99.83_{\pm00.18}$ | $97.99_{\pm00.38}$ | $92.28_{\pm08.67}$ | $96.41_{\pm06.41}$ | $98.50_{\pm01.23}$ | $98.82_{\pm00.36}$ | $94.15_{\pm08.49}$ |
| 2 training domains | $93.61_{\pm02.00}$ | $99.81_{\pm00.07}$ | $97.66_{\pm00.55}$ | $98.64_{\pm01.54}$ | $99.25_{\pm00.30}$ | $98.78_{\pm01.20}$ | $98.43_{\pm00.52}$ | $96.15_{\pm09.00}$ |
| 3 training domains | $95.00_{\pm01.75}$ | $99.82_{\pm00.10}$ | $98.25_{\pm00.22}$ | $99.31_{\pm00.19}$ | $99.73_{\pm00.22}$ | $99.35_{\pm00.51}$ | $98.57_{\pm00.35}$ | $99.50_{\pm00.60}$ |
| 4 training domains | $96.14_{\pm00.80}$ | $99.84_{\pm00.09}$ | $98.17_{\pm00.16}$ | $99.25_{\pm00.17}$ | $99.83_{\pm00.09}$ | $99.55_{\pm00.27}$ | $99.38_{\pm00.12}$ | $99.71_{\pm00.38}$ |

(b) **Node prediction $(i, k, ?)$ performance in %. Higher ↑ is better.**

| Training Scenario | Test on Art | Test on Award | Test on Edu | Test on Health | Test on Infra | Test on Sci | Test on Sport | Test on Tax |
|-------------------|-------------|---------------|-------------|----------------|---------------|-------------|---------------|-------------|
| Best Transductive | $91.70_{\pm00.54}$ | $93.57_{\pm01.32}$ | $96.85_{\pm00.18}$ | $98.92_{\pm00.07}$ | $97.75_{\pm00.17}$ | $96.73_{\pm00.65}$ | $81.21_{\pm00.10}$ | $90.87_{\pm05.16}$ |
| 1 training domain | $88.55_{\pm04.53}$ | $85.16_{\pm05.44}$ | $94.75_{\pm01.23}$ | $89.44_{\pm09.11}$ | $81.81_{\pm08.62}$ | $95.91_{\pm00.52}$ | $66.93_{\pm14.16}$ | $82.85_{\pm12.84}$ |
| 2 training domains | $88.55_{\pm05.52}$ | $88.14_{\pm01.69}$ | $95.87_{\pm01.37}$ | $96.84_{\pm01.58}$ | $89.90_{\pm07.05}$ | $95.88_{\pm01.05}$ | $66.98_{\pm13.73}$ | $86.92_{\pm11.03}$ |
| 3 training domains | $92.26_{\pm00.71}$ | $87.93_{\pm01.23}$ | $96.85_{\pm00.53}$ | $97.27_{\pm01.22}$ | $87.63_{\pm06.73}$ | $96.33_{\pm00.54}$ | $77.19_{\pm05.87}$ | $93.67_{\pm01.25}$ |
| 4 training domains | $90.74_{\pm02.54}$ | $89.47_{\pm00.82}$ | $97.00_{\pm00.36}$ | $97.82_{\pm00.48}$ | $94.01_{\pm01.82}$ | $96.19_{\pm00.28}$ | $80.69_{\pm00.29}$ | $92.89_{\pm01.19}$ |

Table 8: **Average Relation & Node Hits@5 performance of ISDEA+ on the Zero-Shot Meta-Learning task over WikiTopics on each test domain.** We also report the performance of the best transductive model. **ISDEA+ consistently achieves better and better performance when training on more and more domains, closing and even sometimes surpassing the best transductive model performance.**

(a) **Relation prediction $(i, ?, j)$ performance in %. Higher ↑ is better.**

| Training Scenario | Test on Art | Test on Award | Test on Edu | Test on Health | Test on Infra | Test on Sci | Test on Sport | Test on Tax |
|-------------------|-------------|---------------|-------------|----------------|---------------|-------------|---------------|-------------|
| Best Transductive | $83.80_{\pm01.92}$ | $97.32_{\pm00.14}$ | $94.31_{\pm00.27}$ | $98.43_{\pm00.26}$ | $98.32_{\pm00.43}$ | $99.57_{\pm00.16}$ | $90.60_{\pm05.84}$ | $87.11_{\pm16.02}$ |
| 1 training domain | $79.93_{\pm03.42}$ | $96.08_{\pm01.68}$ | $92.83_{\pm00.99}$ | $88.99_{\pm08.63}$ | $92.02_{\pm06.72}$ | $98.45_{\pm01.29}$ | $88.40_{\pm01.23}$ | $94.14_{\pm08.48}$ |
| 2 training domains | $82.78_{\pm03.27}$ | $96.19_{\pm00.51}$ | $93.33_{\pm01.07}$ | $96.48_{\pm01.73}$ | $95.93_{\pm00.88}$ | $98.72_{\pm01.18}$ | $89.79_{\pm05.11}$ | $96.12_{\pm08.98}$ |
| 3 training domains | $84.46_{\pm03.01}$ | $96.08_{\pm00.39}$ | $92.66_{\pm01.32}$ | $96.92_{\pm00.22}$ | $96.28_{\pm00.72}$ | $99.30_{\pm00.49}$ | $86.47_{\pm01.64}$ | $99.48_{\pm00.59}$ |
| 4 training domains | $85.44_{\pm02.71}$ | $96.20_{\pm00.40}$ | $92.74_{\pm01.20}$ | $96.65_{\pm00.57}$ | $97.33_{\pm00.71}$ | $99.47_{\pm00.26}$ | $92.25_{\pm02.16}$ | $99.69_{\pm00.37}$ |

(b) **Node prediction $(i, k, ?)$ performance in %. Higher ↑ is better.**

| Training Scenario | Test on Art | Test on Award | Test on Edu | Test on Health | Test on Infra | Test on Sci | Test on Sport | Test on Tax |
|-------------------|-------------|---------------|-------------|----------------|---------------|-------------|---------------|-------------|
| Best Transductive | $85.01_{\pm00.89}$ | $75.54_{\pm03.10}$ | $86.10_{\pm00.27}$ | $95.95_{\pm00.30}$ | $96.24_{\pm00.11}$ | $95.15_{\pm00.29}$ | $67.30_{\pm00.08}$ | $85.04_{\pm06.03}$ |
| 1 training domain | $80.20_{\pm05.57}$ | $60.59_{\pm09.59}$ | $77.25_{\pm06.02}$ | $81.27_{\pm09.32}$ | $73.61_{\pm08.63}$ | $94.30_{\pm00.71}$ | $52.22_{\pm14.46}$ | $77.75_{\pm12.63}$ |
| 2 training domains | $80.55_{\pm05.80}$ | $65.41_{\pm03.45}$ | $83.19_{\pm02.46}$ | $90.89_{\pm02.09}$ | $84.92_{\pm08.23}$ | $94.36_{\pm01.03}$ | $52.88_{\pm14.05}$ | $81.17_{\pm10.75}$ |
| 3 training domains | $84.32_{\pm01.34}$ | $65.56_{\pm02.17}$ | $84.03_{\pm01.60}$ | $92.20_{\pm01.14}$ | $81.93_{\pm07.45}$ | $94.97_{\pm00.53}$ | $62.13_{\pm06.85}$ | $87.91_{\pm00.88}$ |
| 4 training domains | $83.29_{\pm02.56}$ | $68.97_{\pm01.44}$ | $83.93_{\pm01.11}$ | $91.94_{\pm02.09}$ | $88.38_{\pm04.51}$ | $94.64_{\pm00.43}$ | $66.12_{\pm00.17}$ | $86.91_{\pm01.09}$ |

### F.1.4 DOUBLY INDUCTIVE LINK PREDICTION OVER DATASETS FROM INGRAM (LEE ET AL., 2023)

Lee et al. (2023) proposed the NL-$k$, WK-$k$, and FB-$k$ benchmarks originally used to evaluate InGram's performance of reasoning over new nodes and new relation types at test time, where $k \in \{25, 50, 75, 100\}$ means that, in the test graphs, approximately $k\%$ of triplets have unseen relations. For example, the test graph of WK-100 does not contain any training relations and thus induces a doubly inductive link prediction task. Hence, we run our models (ISDEA and DEq-InGram) against InGram on these benchmarks with results shown in Table 9. We note that, however, due to the different experimental settings (as we discuss next), our results reported in Table 9 are not directly comparable to those reported in Lee et al. (2023), even though they are experimented on essentially the same datasets.

**Difference to the original data split and evaluation in InGram (Lee et al., 2023):** Different from Lee et al. (2023), which uses part of the test graph as the validation set to conduct model hyperparameter search, *our experiments consider a harder setting where the relations in test are not observed in the validation data.* Hence, to modify the NL-$k$, WK-$k$, and FB-$k$ datasets to our setting, we discard the original validation set and instead split the original training set into a new set of training and validation triplets with a ratio of 9:1. During training, the models perform self-supervised masking over the training set of triplets to create the training-time observable triplets and training-time target triplets. During validation, the entire set of the new training triplets is taken as the validation-time observable triplets, and the new validation triplets are the target triplets to predict. In addition, Lee et al. (2023) evaluate their model's node prediction performance against *all* nodes in the graph. For efficiency reasons, we evaluate the model performance by sampling without replacement 50 negative nodes for the node prediction task and sampling with replacement 50 negative relation types for the relation prediction task.

Table 9 shows the results, where we can see that ISDEA outperforms InGram on most datasets on the relation prediction task and has smaller standard deviation in general, and DEq-InGram consistently outperforms InGram on all datasets for both relation prediction and node prediction tasks. Importantly, in the dataset FB-100 which follows our doubly inductive link prediction setting with completely new nodes and new relation types in the test with the largest number of training and test relations (134 in train and 77 in test) (Lee et al., 2023), ISDEA achieves significant better results in the relation perdiction task, showcasing its ability for doubly inductive link prediction.

### F.1.5 A SYNTHETIC CASE STUDY FOR ISDEA

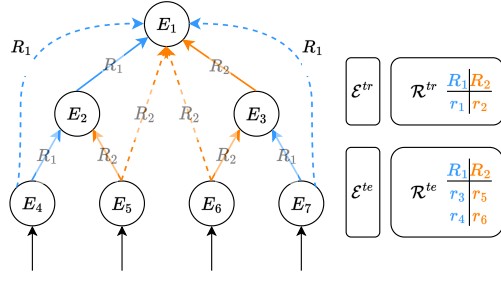

Figure 9: **Synthetic Example of FD-2:** Training and test has their own node and relation type sets: $\mathcal{V}^{(tr)} \cap \mathcal{V}^{(te)} = \emptyset$ and $\mathcal{R}^{(tr)} \cap \mathcal{R}^{(te)} = \emptyset$.

To further understand the expressive power and limitations of our proposed sturctural double equivariant model ISDEA, we create **FD-2** to empirically justify the expressivity of our proposal on tasks over both new nodes and new relation types. On FD-2, training has 127 nodes and 2 relations, while test has 254 nodes and 4 relations (more nodes and more relations).

FD-2 is constructed by only a single rule, $(E_1, R_1, E_3) \land (E_3, R_2, E_2) \Rightarrow (E_1, R_1, E_3)$ where $E_1, E_2, E_3$ and $R_1, R_2$ are all variables. As illustrated in Figure 9, The training data has only two

Table 9: **Relation & Node Hits@10 performance on Doubly Inductive Link Prediction over NL-$k$, WK-$k$, and FB-$k$ of Lee et al. (2023).** We report standard deviations over 5 runs. A higher value means better doubly inductive link prediction performance. The best values are shown in bold font, while the second-best values are underlined. **ISDEA outperforms InGram on most datasets on the relation prediction task, and DEq-InGram consistently outperforms InGram on all datasets for both relation prediction and node prediction tasks.**

(a) **Performance in % on WK-$k$ datasets. Higher ↑ is better.**

| | Relation prediction $(i, ?, j)$ | | | | Node prediction $(i, k, ?)$ | | | |
|---|---|---|---|---|---|---|---|---|
| Models | WK-25 | WK-50 | WK-75 | WK-100 | WK-25 | WK-50 | WK-75 | WK-100 |
| InGram | $58.76_{\pm13.91}$ | $84.01_{\pm03.30}$ | $80.19_{\pm04.19}$ | $58.20_{\pm11.13}$ | $76.99_{\pm07.72}$ | $70.93_{\pm02.38}$ | $78.85_{\pm04.65}$ | $66.29_{\pm03.70}$ |
| DEq-InGram (Ours) | $\underline{81.06}_{\pm22.31}$ | $\mathbf{94.85}_{\pm00.85}$ | $\mathbf{95.84}_{\pm01.54}$ | $\underline{81.83}_{\pm10.10}$ | $\underline{87.91}_{\pm05.68}$ | $82.58_{\pm01.70}$ | $89.10_{\pm02.15}$ | $79.69_{\pm03.07}$ |
| ISDEA+ (Ours) | $\mathbf{96.25}_{\pm00.86}$ | $\underline{93.10}_{\pm02.55}$ | $\underline{95.66}_{\pm00.82}$ | $\mathbf{95.00}_{\pm00.15}$ | $\mathbf{94.41}_{\pm01.61}$ | $\mathbf{90.34}_{\pm01.71}$ | $\mathbf{91.98}_{\pm01.61}$ | $\mathbf{91.22}_{\pm00.45}$ |

(b) **Performance in % on FB-$k$ datasets. Higher ↑ is better.**

| | Relation prediction $(i, ?, j)$ | | | | Node prediction $(i, k, ?)$ | | | |
|---|---|---|---|---|---|---|---|---|
| Models | FB-25 | FB-50 | FB-75 | FB-100 | FB-25 | FB-50 | FB-75 | FB-100 |
| InGram | $68.26_{\pm08.27}$ | $50.41_{\pm08.79}$ | $79.51_{\pm02.69}$ | $40.46_{\pm12.21}$ | $\underline{86.79}_{\pm00.70}$ | $73.32_{\pm06.64}$ | $86.57_{\pm00.69}$ | $71.72_{\pm06.93}$ |
| DEq-InGram (Ours) | $\underline{82.89}_{\pm03.58}$ | $\underline{76.65}_{\pm04.05}$ | $\underline{89.70}_{\pm01.14}$ | $\underline{46.88}_{\pm15.76}$ | $\mathbf{92.39}_{\pm00.30}$ | $81.08_{\pm06.98}$ | $\mathbf{92.14}_{\pm00.43}$ | $\underline{77.54}_{\pm06.36}$ |
| ISDEA+ (Ours) | $\mathbf{96.68}_{\pm00.15}$ | $\mathbf{93.55}_{\pm00.77}$ | $\mathbf{96.86}_{\pm00.58}$ | $\mathbf{97.55}_{\pm00.33}$ | $85.01_{\pm00.92}$ | $\mathbf{84.40}_{\pm00.38}$ | $\underline{86.62}_{\pm00.31}$ | $\mathbf{91.28}_{\pm00.19}$ |

(c) **Performance in % on NL-$k$ datasets. Higher ↑ is better.**

| | Relation prediction $(i, ?, j)$ | | | | Node prediction $(i, k, ?)$ | | | |
|---|---|---|---|---|---|---|---|---|
| Models | NL-25 | NL-50 | NL-75 | NL-100 | NL-25 | NL-50 | NL-75 | NL-100 |
| InGram | $64.54_{\pm16.86}$ | $64.54_{\pm12.56}$ | $80.16_{\pm04.43}$ | $70.84_{\pm08.52}$ | $\underline{89.95}_{\pm02.01}$ | $92.74_{\pm00.52}$ | $95.40_{\pm01.38}$ | $88.20_{\pm01.92}$ |
| DEq-InGram (Ours) | $\underline{83.58}_{\pm17.57}$ | $\underline{91.32}_{\pm05.60}$ | $\underline{96.01}_{\pm01.23}$ | $\underline{87.52}_{\pm10.39}$ | $\mathbf{95.03}_{\pm00.32}$ | $\mathbf{96.02}_{\pm00.34}$ | $\mathbf{97.94}_{\pm00.34}$ | $\mathbf{93.80}_{\pm01.38}$ |
| ISDEA+ (Ours) | $\mathbf{92.04}_{\pm02.67}$ | $\mathbf{93.78}_{\pm01.37}$ | $\mathbf{96.57}_{\pm01.85}$ | $\mathbf{99.72}_{\pm00.15}$ | $82.20_{\pm04.36}$ | $83.35_{\pm00.40}$ | $\underline{90.94}_{\pm00.85}$ | $\underline{90.06}_{\pm00.49}$ |

relation types $\{r_1, r_2\}$, while test data has four relation types $\{r_3, r_4, r_5, r_6\}$ which are all different from training relations. For all relation types, only $r_1, r_3, r_4$ can be used for $R_1$ assignments, and only $r_2, r_5, r_6$ can be use for $R_2$ assignments. Besides, training and test also have distinct node sets.

Each graph (training or test) is consisted by one or more tree-like structures as left side of Figure 9. In each tree-like structure, all solid edges are used as observations, and will form a complete binary tree; while all dashed edges are used as training, validation or test samples which are built by applying the only rule over all observed edges. In training, we have only one tree-like structure; while in test, we have two disconnected tree-like structures. A more detailed generation algorithm for a graph given depths of all tree-like structures is provided in Algorithm 2.

Since the structure of FD-2 does not satisfy the requirement of the spanning tree algorithm used in InGram (Lee et al., 2023), we are not able to apply InGram and DEq-InGram on FD-2. So we provide the results on FD-2 in Table 10 with all remaining baselines and ISDEA. We can see that ISDEA clearly perform better than other baselines, especially in the relation prediction task, and shows capability to perform accurately on the doubly inductive link prediction over both new nodes and new relation types, while methods like NBFNet and RMPI are not able to correctly perdict this task, even for node prediction.

### F.1.6 EXPRESSIVITY LIMITATION CASE STUDY WITH FD-2 FOR ISDEA

We now provide a FD-2 variant where we show that double equivariant representation is not expressive enough to solve a specific task. It is a simple 2-depth tree structure as shown in Figure 10. We denote node representations given by arbitrary double equivariant representation as $H_{v,r}$ where $v \in [1, 7]$ and $r \in [1, 4]$. We can easily notice that $e_4$ and $e_7$ are symmetric, $e_5$ and $e_6$ are symmetric (simply flipping blue and orange colors), thus we will expect $H_{4,1} = H_{7,2}, H_{4,2} = H_{7,1}, H_{5,1} = H_{6,2}, H_{5,2} = H_{6,1}$.

Since there is no $r_3$ and $r_4$ in observation, they are freely exchangeable with each other, thus we will also expect

$$H_{1,3} = H_{1,4},$$
$$H_{4,3} = H_{4,4} = H_{7,4} = H_{7,3},$$
$$H_{5,3} = H_{5,4} = H_{6,4} = H_{6,3}.$$

**Algorithm 2 Synthesis Algorithm for FD-2.** This is triplet generation code for a single graph (either training and test). It will provide observation and query triplets. For training, query triplets are further divided into training and validation triplets; For test, query triplets directly become test triplets.

---

**Require:** Tree depth $\{D_1, \ldots, D_M\}$, Node Labeling "Names$^{\text{nd}}$", Relation Type Labeling "Names$^{\text{rl}}$".
**Ensure:** Observation triplets $\mathcal{S}$, Query triplets $\mathcal{Q}$

```
 1: S = ∅
 2: Q = ∅
 3: n ← 0
 4: for m ← 1, . . . , M do
 5:     for d ← 1, . . . , D_m do
 6:         for v ← 2^d − 1, . . . , 2^{d+1} − 2 do
 7:             u_1 ← ⌈(v − 2)/2⌉
 8:             u_2 ← ⌈(u_1 − 2)/2⌉
 9:             if v mod 2 = 0 then                    ▷ For relation type variable R_2.
10:                 if u_1 ≥ 0 then
11:                     S.add( (Names^nd[n + v], Names^rl[2m − 1], Names^nd[n + u_1]) )
12:                 end if
13:                 if u_2 ≥ 0 then
14:                     Q.add( (Names^nd[n + v], Names^rl[2m − 1], Names^nd[n + u_2]) )
15:                 end if
16:             else                                   ▷ For relation type variable R_1.
17:                 if u_1 ≥ 0 then
18:                     S.add( (Names^nd[n + v], Names^rl[2m − 2], Names^nd[n + u_1]) )
19:                 end if
20:                 if u_2 ≥ 0 then
21:                     Q.add( (Names^nd[n + v], Names^rl[2m − 2], Names^nd[n + u_2]) )
22:                 end if
23:             end if
24:         end for
25:         n ← n + 2^d
26:     end for
27: end for
```

Table 10: **Relation & Node performance on Doubly Inductive Link Prediction over FD2.** We report standard deviations over 5 runs. A higher value means better doubly inductive link prediction performance. The best values are shown in bold font, while the second-best values are underlined. ISDEA consistently achieve better results than the baselines, especially in the Relation perdiction task. NA* due to the fact that FD2 does not satisfy the spanning tree algorithm used in InGram (Lee et al., 2023).

| Models | Relation prediction $(i, ?, j)$ | | | | Node prediction $(i, k, ?)$ | | | |
| --- | --- | --- | --- | --- | --- | --- | --- | --- |
| | MRR | Hits@1 | Hits@2 | Hits@4 | MRR | Hits@1 | Hits@2 | Hits@4 |
| GAT | $7.61_{\pm00.71}$ | $0.77_{\pm00.39}$ | $2.78_{\pm00.80}$ | $5.85_{\pm00.95}$ | $84.62_{\pm02.64}$ | $71.61_{\pm04.94}$ | $\underline{93.51}_{\pm01.03}$ | $99.72_{\pm00.27}$ |
| GIN | $8.44_{\pm00.40}$ | $1.29_{\pm00.37}$ | $3.51_{\pm00.58}$ | $\underline{7.18}_{\pm01.01}$ | $73.99_{\pm09.60}$ | $65.73_{\pm06.58}$ | $76.69_{\pm12.44}$ | $81.45_{\pm15.80}$ |
| GraphConv | $7.88_{\pm00.45}$ | $0.81_{\pm00.29}$ | $2.62_{\pm00.61}$ | $6.98_{\pm01.09}$ | $\underline{85.95}_{\pm00.77}$ | $\underline{74.52}_{\pm01.81}$ | $92.66_{\pm01.02}$ | $\mathbf{99.84}_{\pm00.15}$ |
| RMPI | $\underline{9.09}_{\pm03.18}$ | $\underline{1.94}_{\pm01.88}$ | $\underline{3.95}_{\pm04.43}$ | $7.10_{\pm05.58}$ | $21.16_{\pm05.85}$ | $9.84_{\pm05.04}$ | $16.74_{\pm06.50}$ | $27.98_{\pm09.96}$ |
| NBFNet | $6.39_{\pm02.19}$ | $1.50_{\pm02.49}$ | $1.79_{\pm02.39}$ | $2.91_{\pm02.22}$ | $21.95_{\pm04.14}$ | $14.44_{\pm04.34}$ | $18.61_{\pm04.29}$ | $26.47_{\pm04.24}$ |
| InGram | N/A* | N/A* | N/A* | N/A* | N/A* | N/A* | N/A* | N/A* |
| DEq-InGram (Ours) | N/A* | N/A* | N/A* | N/A* | N/A* | N/A* | N/A* | N/A* |
| ISDEA (Ours) | $\mathbf{44.39}_{\pm12.17}$ | $\mathbf{32.82}_{\pm12.69}$ | $\mathbf{38.71}_{\pm13.60}$ | $\mathbf{50.73}_{\pm14.04}$ | $\mathbf{90.98}_{\pm03.55}$ | $\mathbf{83.59}_{\pm06.22}$ | $\mathbf{95.69}_{\pm02.34}$ | $\underline{99.72}_{\pm00.27}$ |

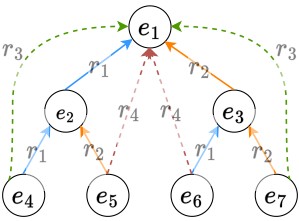

Figure 10: **Expressivity Limitation:** Relation $r_1$ and $r_2$ are always observed, while $r_3$ and $r_4$ are always querying. $r_3$ implies that relation types on the path are same, while $r_4$ implies that relation types on the path are different..

After getting all those representations, we can now focus on querying triplet representations (dashed green and red) by concatenating head and tail node representations w.r.t. relation types:

$$\Gamma_{\text{tri}}\left((e_1, r_4, e_4), \mathbf{A}\right) = H_{1,4} \parallel H_{4,4},$$
$$\Gamma_{\text{tri}}\left((e_1, r_3, e_5), \mathbf{A}\right) = H_{1,3} \parallel H_{5,3},$$
$$\Gamma_{\text{tri}}\left((e_1, r_3, e_6), \mathbf{A}\right) = H_{1,3} \parallel H_{6,3},$$
$$\Gamma_{\text{tri}}\left((e_1, r_4, e_7), \mathbf{A}\right) = H_{1,4} \parallel H_{7,4}.$$

We can notice that

$$\overbrace{H_{1,4} \parallel H_{4,4}}^{\Gamma_{\text{tri}}((e_1,r_4,e_4),\mathbf{A})} = \overbrace{H_{1,4} \parallel H_{7,4}}^{\Gamma_{\text{tri}}((e_1,r_4,e_7),\mathbf{A})} = \overbrace{H_{1,3} \parallel H_{7,3}}^{\Gamma_{\text{tri}}((e_1,r_3,e_7),\mathbf{A})} = \overbrace{H_{1,3} \parallel H_{4,3}}^{\Gamma_{\text{tri}}((e_1,r_3,e_4),\mathbf{A})},$$

$$\overbrace{H_{1,4} \parallel H_{5,4}}^{\Gamma_{\text{tri}}((e_1,r_4,e_5),\mathbf{A})} = \overbrace{H_{1,4} \parallel H_{6,4}}^{\Gamma_{\text{tri}}((e_1,r_4,e_6),\mathbf{A})} = \overbrace{H_{1,3} \parallel H_{6,3}}^{\Gamma_{\text{tri}}((e_1,r_3,e_6),\mathbf{A})} = \overbrace{H_{1,3} \parallel H_{5,3}}^{\Gamma_{\text{tri}}((e_1,r_3,e_5),\mathbf{A})}.$$

Suppose the score of $(e_u, r_c, e_v)$ utilizing such representation is $s_{u,c,v}$, we will have

$$s_{4,4,1} = s_{7,4,1} = s_{7,3,1} = s_{4,3,1},$$
$$s_{5,4,1} = s_{6,4,1} = s_{6,3,1} = s_{5,3,1}.$$

If a model can distinguish $r_3$ and $r_4$, it should at least rank node $e_7$ higher than $e_6$ given head node $e_1$ and relation $r_3$ since this is a positive triplet in training. Then, we will have $s_{7,3,1} > s_{6,3,1}$, since we already knew that $s_{7,3,1} = s_{7,4,1}, s_{6,3,1} = s_{6,4,1}$, we will also have $s_{7,4,1} > s_{6,4,1}$. This means that we rank node $e_7$ higher than node $e_6$ given head node $e_1$ and relation $r_4$, however, this is incorrect since $(e_7, r_4, e_1))$ is negative while $(e_6, r_4, e_1)$ is positive. In summary, if we use double equivariant representation for triplet scoring in this specific example, there is no way for it to correctly rank $r_3$ and $r_4$ in the same time. This shows that double equivariant representation (even the most expressive) can face challenges for doubly inductive link prediction on knowledge graph.

### F.2 Complexity Analysis for ISDEA and ISDEA+

For each layer of our method ISDEA, it can be treated as running 2 unattributed GNN $|\mathcal{R}|$ times on the knowledge graph, thus time cost is roughly $2|\mathcal{R}|$ times of adopted GNN. In our experiment, we use node representation GNNs (e.g., GIN (Xu et al., 2019a), GAT (Veličković et al., 2017), GraphConv (Morris et al., 2019)) as our GNN architecture, thus the complexity is $\mathcal{O}(|\mathcal{R}|L(|\mathcal{S}|d + |\mathcal{V}|d^2))$ where $L$ is the number of layers, $d$ is the maximum size of hidden layers, $|\mathcal{V}|$ is number of nodes and $|\mathcal{R}|$ is number of relations in the knowledge graph, and $|\mathcal{S}|$ is number of fact triplets (number of edges) in the knowledge graph.

However, for ISDEA+, since the update for each relation can be conducted independently, by using parrallelizing or a carefully designed training batch process, we can reduce the time complexity to $\mathcal{O}(L(|\mathcal{S}|d + |\mathcal{V}|d^2))$, $|\mathcal{R}|$ times faster than ISDEA.

Besides, for both positive and negative samples $(i, k, j)$, our method requires the shortest distance between any two nodes without considering $(i, k, j)$, which can be achieved from the Dijkstra or Floyd algorithm, where we will pre-split the graph in ISDEA+ and simplify the computation of calculating distances to only one time rather than repeating it in each batch as implemented in ISDEA.

### F.3 Ablation study for ISDEA

Since a part of negative samplings is drawn by uniformly corrupting objects (without loss of generality), it is very likely that corrupted objects are far way from the subject while the true object is close to the subject. Then, the distance feature can help predict in such cases. However, the shortest distance feature will not provide any additional information if we corrupt the relation type. Under such a scenario, shortest distance itself may provide some features to achieve good ranking performance in inductive link prediction on knowledge graph, thus we want to know if shortest distance feature augmentation contributes to the performance gain. We perform an ablation study for ISDEA with or without distance on doubly inductive link prediction over PediaTypes.

As shown in Table 11, even if the shortest distance is excluded from our model, our model still performs quite well and is better than most other baselines in the doubly inductive link prediction on PediaTypes. Especially, as we anticipate, the distance feature is more helpful in the node prediction task than the relation prediction task. Thus, we can say that double equivariant node representation itself is enough to provide good performance on doubly inductive link prediction.

### F.4 Limitations and Impacts for ISDEA and ISDEA+

ISDEA and ISDEA+ excels both in synthetic and real-world benchmarks. However, the simplification from pairwise to node embeddings limits its expressivity. In Appendix F.1.5, we give a synthetic counterexample how this could be an issue in some knowledge graphs. We also do not envision a direct negative social impact of our work.

## G  Future Work

As addressed in the main paper, our implemented architecture has a few limitations, which could be addressed in future work. First, ISDEA+ has high pre-processing cost. This high time cost is introduced by using shortest distances whose computation is of the same complexity as enclosed subgraph. However, our ablation studies show that shortest distances is not a dominant factor in our model for real-world tasks, thus it is possible that shortest distances can be replaced by other heuristics that can be efficiently extracted.

Second, our specific implementation ISDEA happens to have high training and inference costs, since it relies on repeating GNNs for each relation. Thus, complexity ISDEA of scales linearly w.r.t. number of relations, which is often a large number in real-world knowledge base, e.g., Wikipedia. However, fully equivariance over all relations can be too strong, and we may only want partial equivariance which may reduce the cost. We resolve this issue by introduing ISDEA+.

Table 11: **Relation & Node performance on Doubly Inductive Link Prediction over PediaTypes for ISDEA with/without Shortest Distances.** We report standard deviations over 5 runs. A higher value means better doubly inductive link prediction performance. **Even without the shortest distance as an augmented feature, our proposal still achieves comparable results, especially in the relation prediction task.**

(a) **Relation prediction** $(i, ?, j)$ **performance in %. Higher ↑ is better.**

| Dataset | | MRR | Hits@1 | Hits@5 | Hits@10 |
|---|---|---|---|---|---|
| EN-FR | ISDEA w/ Distance | $70.06_{\pm02.01}$ | $61.46_{\pm00.79}$ | $82.11_{\pm04.01}$ | $84.94_{\pm05.00}$ |
| | ISDEA w/o Distance | $68.65_{\pm00.41}$ | $60.34_{\pm00.53}$ | $80.17_{\pm00.99}$ | $82.80_{\pm01.73}$ |
| FR-EN | ISDEA w/ Distance | $69.01_{\pm00.57}$ | $58.18_{\pm00.14}$ | $83.19_{\pm01.73}$ | $84.75_{\pm02.51}$ |
| | ISDEA w/o Distance | $67.74_{\pm01.15}$ | $56.35_{\pm01.53}$ | $83.07_{\pm00.75}$ | $86.23_{\pm00.56}$ |
| EN-DE | ISDEA w/ Distance | $78.38_{\pm04.04}$ | $68.00_{\pm06.41}$ | $92.39_{\pm00.83}$ | $95.26_{\pm00.63}$ |
| | ISDEA w/o Distance | $76.52_{\pm01.32}$ | $67.66_{\pm02.37}$ | $87.49_{\pm00.87}$ | $88.47_{\pm00.64}$ |
| DE-EN | ISDEA w/ Distance | $88.82_{\pm00.28}$ | $84.83_{\pm00.29}$ | $93.59_{\pm00.53}$ | $94.23_{\pm00.71}$ |
| | ISDEA w/o Distance | $88.94_{\pm00.92}$ | $84.76_{\pm00.49}$ | $93.98_{\pm01.74}$ | $94.73_{\pm01.98}$ |
| DB-WD | ISDEA w/ Distance | $65.89_{\pm04.71}$ | $57.51_{\pm05.40}$ | $75.95_{\pm03.89}$ | $82.22_{\pm02.44}$ |
| | ISDEA w/o Distance | $70.66_{\pm07.05}$ | $63.36_{\pm05.30}$ | $79.87_{\pm10.19}$ | $82.96_{\pm11.89}$ |
| WD-DB | ISDEA w/ Distance | $72.57_{\pm00.73}$ | $62.72_{\pm01.24}$ | $86.10_{\pm01.26}$ | $88.87_{\pm02.94}$ |
| | ISDEA w/o Distance | $67.98_{\pm02.14}$ | $60.83_{\pm01.55}$ | $76.65_{\pm03.14}$ | $77.90_{\pm03.09}$ |
| DB-YG | ISDEA w/ Distance | $75.88_{\pm01.58}$ | $69.12_{\pm02.40}$ | $85.80_{\pm01.23}$ | $91.42_{\pm01.79}$ |
| | ISDEA w/o Distance | $75.42_{\pm00.35}$ | $69.17_{\pm01.13}$ | $84.86_{\pm01.58}$ | $88.78_{\pm02.36}$ |
| YG-DB | ISDEA w/ Distance | $74.04_{\pm00.47}$ | $66.68_{\pm00.81}$ | $83.36_{\pm01.55}$ | $85.34_{\pm01.49}$ |
| | ISDEA w/o Distance | $74.22_{\pm01.56}$ | $66.97_{\pm01.63}$ | $83.62_{\pm01.85}$ | $85.73_{\pm02.66}$ |

(b) **Node prediction** $(i, k, ?)$ **performance in %. Higher ↑ is better.**

| Dataset | | MRR | Hits@1 | Hits@5 | Hits@10 |
|---|---|---|---|---|---|
| EN-FR | ISDEA w/ Distance | $53.92_{\pm00.26}$ | $43.03_{\pm00.25}$ | $64.45_{\pm00.24}$ | $76.28_{\pm00.50}$ |
| | ISDEA w/o Distance | $45.12_{\pm00.41}$ | $34.04_{\pm00.36}$ | $56.61_{\pm00.48}$ | $63.46_{\pm00.76}$ |
| FR-EN | ISDEA w/ Distance | $57.68_{\pm00.68}$ | $47.38_{\pm00.28}$ | $67.24_{\pm01.32}$ | $77.51_{\pm01.46}$ |
| | ISDEA w/o Distance | $42.52_{\pm00.91}$ | $30.41_{\pm01.17}$ | $54.94_{\pm00.22}$ | $65.29_{\pm00.20}$ |
| EN-DE | ISDEA w/ Distance | $50.30_{\pm02.08}$ | $35.41_{\pm02.25}$ | $68.80_{\pm01.90}$ | $82.24_{\pm00.94}$ |
| | ISDEA w/o Distance | $45.16_{\pm00.76}$ | $30.26_{\pm00.76}$ | $62.59_{\pm00.57}$ | $76.98_{\pm00.63}$ |
| DE-EN | ISDEA w/ Distance | $51.33_{\pm00.40}$ | $37.12_{\pm00.31}$ | $68.20_{\pm00.53}$ | $81.80_{\pm00.68}$ |
| | ISDEA w/o Distance | $43.67_{\pm00.32}$ | $28.97_{\pm00.25}$ | $60.36_{\pm00.54}$ | $74.95_{\pm00.51}$ |
| DB-WD | ISDEA w/ Distance | $45.75_{\pm00.66}$ | $35.59_{\pm00.73}$ | $54.83_{\pm00.90}$ | $66.69_{\pm01.01}$ |
| | ISDEA w/o Distance | $40.26_{\pm03.77}$ | $30.59_{\pm03.89}$ | $48.15_{\pm03.68}$ | $59.43_{\pm03.76}$ |
| WD-DB | ISDEA w/ Distance | $51.64_{\pm00.60}$ | $40.56_{\pm01.72}$ | $62.60_{\pm02.55}$ | $75.19_{\pm03.12}$ |
| | ISDEA w/o Distance | $45.94_{\pm00.14}$ | $35.17_{\pm00.30}$ | $56.89_{\pm00.33}$ | $66.46_{\pm00.55}$ |
| DB-YG | ISDEA w/ Distance | $41.72_{\pm01.64}$ | $27.70_{\pm01.95}$ | $55.21_{\pm01.07}$ | $72.87_{\pm01.03}$ |
| | ISDEA w/o Distance | $32.71_{\pm00.60}$ | $17.69_{\pm00.39}$ | $47.82_{\pm00.60}$ | $66.92_{\pm01.90}$ |
| YG-DB | ISDEA w/ Distance | $48.21_{\pm01.06}$ | $35.29_{\pm01.67}$ | $61.87_{\pm01.30}$ | $76.41_{\pm01.52}$ |
| | ISDEA w/o Distance | $37.52_{\pm00.79}$ | $23.10_{\pm00.76}$ | $53.34_{\pm00.88}$ | $68.43_{\pm01.62}$ |

Third, ISDEA+ has expressivity limitation. This limitation is related to former two cost issues since it is caused by compromising most-expressive pairwise representation to node-wise representation due to time cost. Thus if we can reduce the cost, we may be able to use more expressive graph encoder.

Finally, although we show ISDEA+ representations can capture UQER Horn clauses, there is no algorithm to create UQER Horn clauses from ISDEA+ representations. This topic is known as *explainability* which is important in graph machine learning community. We leave such an algorithm as another future work other than optimization.

