# OpenReview forum: "Double Equivariance for Inductive Link Prediction for Both New Nodes and New Relation Types"
_ICLR.cc/2024/Conference — Submitted to ICLR 2024_

### Official Review · Reviewer_1Qir · 2023-10-28

**Soundness:** 3 good
**Presentation:** 3 good
**Contribution:** 3 good
**Rating:** 6
**Confidence:** 4

**Summary:**

This paper investigates inductive link prediction for both new entities and new relations. It proposes an inductive structural double equivariant architecture that decomposes a knowledge graph into subgraphs containing different relations and encodes and aggregates them in the same way to eliminate the use of relation embeddings. The paper also constructs two datasets based on OpenEA and Wikidata5M. Extensive experimental results demonstrate the strong performance of ISDEA.

**Strengths:**

S1. This paper addresses an important and challenging task.

S2. It introduces a novel framework that avoids reliance on relationship embeddings.

S3. Good reproducibility - the paper provides code and detailed experimental settings.

**Weaknesses:**

W1. The proposed framework requires significant preprocessing and expensive encoding costs. This may be attributed to three factors: preprocessing costs, encoding for each relation, and separate scoring for each candidate entity.

W2. The 1 vs. 50 evaluation poses a risk as negative samples obtained from negative sampling are mostly easily distinguishable. This setup may not be sufficient to cover real-world scenarios.

W3. While ISDEA appears suitable for relation prediction, its performance on node prediction is not very good.

**Questions:**

Q1. As shown in Table 1(b), ISDEA's performance is not good and, in some datasets, even receives the lowest scores. Can you explain the reasons for this?

Q2. I am concerned about the efficiency of the proposed framework. Could you report training and inference times on some datasets?

Q3. It should be clarified that the multilingual KGs in the OpenEA library share the same schema. So many of the relations in these KGs overlap.

---

> ### Author Response · Authors · 2023-11-19
>
> ## Official Response to Reviewer 1Qir
>
> We are grateful to the reviewer 1Qir for recognizing the significance of our work. Your thoughtful questions and constructive feedback are highly valued. Please refer to the Common Reviewer Response for the most important revisions in our submission. Here are our detailed responses to your comments:
>
> **Comment 1. The proposed framework requires significant preprocessing and expensive encoding costs.**
>
> **A1:** In Appendix F.2 and G, we have discussed the limitation of our proposed ISDEA architecture due to its computation cost. The primary source of complexity in ISDEA is the process of enumerating across all relations and computing distances between every pair of nodes. However, we believe separate scoring for each candidate entity is commonly used by existing literatures such as DistMult (Yang et al., ICLR 2015), RGCN (Schlichtkrull et al., ESWC 2018), GraIL (Teru et al., ICML 2020), etc..
>
> To address this, as discussed in the Common Reviewer Response, we have developed ISDEA+, which improves the time complexity by $R$ over ISDEA (where $R$ is the number of relation types). In our experiments, we observe ISDEA+ is between 20x to 120x faster than ISDEA (old).
>
> **Comment 2. Only using 50 negative samples may not be sufficient to cover real-world scenarios.**
>
> **A2:** The choice was purely due to computational constraints of our proposed ISDEA (old) since we wanted at least 5 runs for each experiment. We have simply followed the procedure proposed by GraIL (Teru et al., ICML 2020), which is among the first works to apply GNNs for inductive link prediction over new nodes (as GraIL, we perform evaluation against 50 random negative samples). This choice allows some balance between computational cost and meaningful performance insights.
>
> In our rebuttal, we introduced an improved version of ISDEA, named ISDEA+. This enhancement allows us to handle larger negative samples during evaluation. Here are the results we have managed to gather in the limited time available. In particular, we rerun the node prediction task on all tasks of PediaTypes with 500 random negative node samples, and collected the Hits@10 results in the following table:
>
> | Models     | EN-FR          | FR-EN          | EN-DE          | DE-EN          | DB-WD           | WD-DB          | DB-YG          | YG-DB           |
> |------------|----------------|----------------|----------------|----------------|-----------------|----------------|----------------|-----------------|
> | NBFNet     | $65.73\pm3.17$ | $73.36\pm1.84$ | $58.93\pm2.89$ | $32.15\pm3.14$ | $33.75\pm2.87$  | $61.64\pm1.36$ | $34.07\pm1.49$ | $51.00\pm1.90$ |
> | InGram     | $67.42\pm2.42$ | $48.76\pm8.10$ | $46.84\pm4.07$ | $54.08\pm1.35$ | $24.87\pm10.38$ | $49.41\pm5.41$ | $15.69\pm4.11$ | $23.73\pm7.12$  |
> | DEq-InGram | $76.33\pm1.27$ | $64.84\pm6.17$ | $59.67\pm2.43$ | $66.30\pm2.42$ | $35.70\pm12.10$ | $65.55\pm4.04$ | $24.11\pm4.73$ | $37.76\pm12.01$ |
> | ISDEA+     | $68.27\pm3.62$ | $50.76\pm2.57$ | $66.83\pm0.48$ | $53.52\pm0.67$ | $56.97\pm0.85$  | $65.75\pm0.78$ | $20.72\pm0.35$ | $26.67\pm1.76$  |
>
>
> From these results, we can observe that even with an enlarged number of negative samples, we still observe the same model behaviors as negative sample size 50. The NBFNet model is still able to achieve relatively good performance in the node prediction task while its model architecture is not able to capture the relationship among the new relation types. Both of our proposed models, ISDEA+ (and DEq-InGram), still achieve better performance than the baseline NBFNet and InGram in most scenarios.
>
> **Comment 3. The performance of ISDEA on node prediction is not good.**
>
> **A3:** Please see new results of ISDEA+ in Section 5.1 of the revised manuscript, which now outperforms DEq-InGram in nearly all scenarios. The improvement in performance of DEq-InGram over InGram confirms our theory that double equivariance allows neural networks to perform zero-shot double inductive link prediction tasks.
>
> PS: The node prediction performance aligns with findings from Jambor et al., EACL 2021, where a model's success in node prediction tasks may not fully reflect its capabilities in predicting the type of link, but simply that of being able to predict homogeneous links (in a way agnostic to its type). For instance, RMPI and NBFNet have good performance at predicting nodes but significantly worse performance at predicting edge types.

---

> ### Author Response · Authors · 2023-11-19
>
> **Comment 4. What is the training and inference times for ISDEA, and how is the efficiency of the architecture?**
>
> **A4:** Here we present a run time comparison for ISDEA+, ISDEA, DEq-InGram, InGram and NBFNet on the node prediction task on EN-FR in our PediaTypes dataset, and a much larger transductive FB15K-237 dataset. Due to varied mini-batch training sizes for different models, we report the training time per minibatch step (i.e., per gradient update step) for the training time comparison. Here are the results:
>
> **Training and inference on the EN-FR in PediaTypes**
> - 4962 entities, 122 relations, 30876 train triplets, 3326 validation triplets, 2485 test triplets
>
> | Model      | Training Time per Minibatch Step | Test Inference Time | Max GPU Memory Usage |
> |------------|----------------------------------|---------------------|----------------------|
> | ISDEA+     | 0.05s                            | 6.53s               | 1.9GB                |
> | ISDEA      | 2.56s                            | 359.45s             | 9.8GB                |
> | Deq-InGram | 0.09s                            | 10.39s              | 3.0GB                |
> | InGram     | 0.09s                            | 5.60s               | 3.0GB                |
> | NBFNet     | 0.02s                            | 5.14s               | 5.5GB                |
>
> **Training and inference on the transductive FB15k-237 dataset**
> - 14541 entities, 237 relations, 272115 train triplets, 17535 validation triplets, 20466 test triplets
>
> | Model      | Training Time per Minibatch Step  | Test Inference Time | Max GPU Memory Usage |
> |------------|-----------------------------------|---------------------|----------------------|
> | ISDEA+     | 0.08s                             | 27.60s              | 5GB                  |
> | ISDEA      | N/A                               | N/A                 | OOM                  |
> | Deq-InGram | 0.57s                             | 99.67s              | 13GB                 |
> | InGram     | 0.57s                             | 62.60s              | 13GB                 |
> | NBFNet     | 0.07s                             | 62.23s              | 14GB                 |
>
> We can see that ISDEA+ significantly improves ISDEA, and is among the fastest models with least memory consumption.
>
>
> **Comment 5. It should be clarified that multilingual KGs in the OpenEA library share the same schema. So many of the relations in these KGs overlap.**
>
> **A5:** We appreciate your observation on this aspect. Indeed, the multilingual KGs within the OpenEA library consist of several pairs of knowledge graphs, each representing similar domains but in different languages. For example, entities like “http://dbpedia.org/resource/E678522” in English and “http://fr.dbpedia.org/resource/E415873” in French represent the same concept, although with different linguistic labels. It is accurate that these KGs predominantly use English for relation labels, which causes an overlap in relations. However, in our experimental setup, we treat relations as if they were in different languages and do not leverage this overlapping information during model training. Moreover, in other KGs used in our PediaTypes and WikiTopics datasets, such overlapping relations are not present. We have included this clarification in the revised submission, in the description of our datasets. Thank you for pointing this out.
>
>
> # Summary
>
> We want to thank the reviewer for their time and insightful suggestions! We hope our answers can address your concerns well.
>
> We also prudently ask you to reconsider our work if the concerns are addressed. With the new zero-shot meta-learning results and the significant performance boost of our new method ISDEA+, we hope the reviewer will reconsider their score.
>
> Overall, we believe our work makes important contributions in inductive learning on knowledge graphs, and proposes a new paradigm and novel architecture ISDEA+. Thank you for your efforts again!
>
> **References**
>
> [1] Yang et al., "Embedding entities and relations for learning and inference in knowledge bases." ICLR 2015
>
> [2] Schlichtkrull et al., "Modeling Relational Data with Graph Convolutional Networks." ESWC 2018
>
> [3] Teru et al. "Inductive Relation Prediction by Subgraph Reasoning." ICML 2020
>
> [4] Jambor et al. "Exploring the limits of few-shot link prediction in knowledge graphs." EACL 2021

---

### Official Review · Reviewer_VGiN · 2023-10-30

**Soundness:** 3 good
**Presentation:** 4 excellent
**Contribution:** 3 good
**Rating:** 8
**Confidence:** 4

**Summary:**

The paper introduces a theoretical framework for inductive link prediction over multi-relational graphs (knowledge graphs) where both entities and relations are unseen at test time. The framework includes the concepts of double permutation equivariance (to node permutation and edge type permutation) and its slight relaxation of distributionally double equivariance (to incorporate another existing model into the framework). Further, the authors introduce the first GNN implementation of the proposed framework – ISDEA as a double equivariant model, and DEq-InGram as a distributionally-double equivariant version of InGram. Experimentally, the authors devise a handful of new datasets and run experiments on relation prediction $(i, ?, k)$ and node prediction $(i, r, ?)$ tasks.

**Strengths:**

**S1.** Overall, I think it is a solid work that lays important theoretical foundations for the hardest of inductive link prediction tasks - dealing with both new entities and relations at test time requires more effort beyond learning relation embeddings. This is highly relevant for modern graph learning tasks, especially in low-data regimes without input node features.

**S2.** The experimental agenda is convincing - a handful of newly proposed datasets with relation prediction and entity prediction tasks. Perhaps the experimental section could have been even stronger if the evaluation was performed on all nodes/relations in the inference graph instead of 50 random negatives, but the authors acknowledge it is the scalability issues of the ISDEA model (not the framework in general) that are likely to be addressed in the future work.

**Weaknesses:**

The following ones are not the critical weaknesses but rather several discussion points I’d invite the authors to elaborate on:

**W1.** The formalization in Section 2 assumes the existence of bijections (nodes-to-nodes, relations-to-relations) in training and test graphs. Basically, the framework posits the double equivariance only when training and test graphs have exactly the same number of nodes and edge types - which practically does not happen very often. On the other hand, the constructed datasets PediaTypes and WikiTopics all have different numbers of nodes and relations at training and test time (so there is no bijection possible). Could you please comment on the seeming discrepancy between the theory and what is measured in the experiments?

**W2.** Section 5.2: “_relatively easier task of node prediction_” - I do not quite agree with this statement. The results might suggest the task is easier simply because you take 50 random negatives among _thousands_ of nodes in the inference graph, so those negatives are likely to be _easy_ negatives. On the other hand, the number of relations in the datasets is 50-150 in PediaTypes and <50 in WikiTopics, so the negative relation samples are likely to be harder. It was found that evaluation on small number of negative entities overestimates the performance, so I would hypothesize the numbers (and task impression) would change when the architecture would scale to ranking all nodes in the inference graph.

**Questions:**

**Q1.** What are the input features to standard GNN architectures reported in the experiments under GraphConv / GAT / GIN? Initialization of nodes with all ones or with random vectors?

**Q2.** Since DEq-InGram is distributionally double equivariant (by means of averaging several runs with different random relation vectors initializations), would averaging NBFNet results across several runs with random relation initialization count as distributionally double equivariant as well?

**Q3.** The distributionally double equivariant idea posits equivariance in expectation, of which the easiest implementation is averaging over several runs (if we talk about drawing samples of relation vectors). Drawing parallels to group-equivariant CNNs, it is possible to achieve equivariance via augmentations such as frame averaging. I wonder if any such “augmentation” or frame averaging is possible within the double equivariance framework. If so, it might be a good idea to clearly state in the paper that distributionally-double equivariance is different from frame averaging

---

> ### Author Response · Authors · 2023-11-19
>
> ## Official Response to Reviewer VGiN
>
> We are grateful to reviewer VGiN for seeing the significance of our work. Your insightful questions and valuable feedback are deeply appreciated. Please refer to the Common Reviewer Response for the most important revisions in our submission. Below, we provide comprehensive responses to your comments:
>
> **Comment 1. The theorem of double equivariant representation seems to only apply to the graphs with the same number of nodes and edges, however, it is not common in real-world graphs and is also not the setting considered in the experiments section. How to explain the discrepancy between the theory and experiments? What is the goal of the experiments?**
>
> **A1:** Thank you for the insightful question, which allows us to clarify our method (we added a comment to the revised submission). In all our WikiTopics datasets the training and test datasets have **different** number of relations (Figure 6 in the Appendix F.1.3). ISDEA+ works with a different number of relations between train and test. There is no need to have a bijective mapping between the relations in training and test. GNNs are node-equivariant and work on test graphs with varying numbers of nodes. ISDEA+ is double (relation+node) equivariant and works on test graphs with varying numbers of nodes and relation types. Isomorphic pairs are used to help us understand the property of ISDEA+, but ISDEA+ can be applied to non-isomorphic graphs. In standard graph tasks, a GNN tends to make similar embeddings for nodes that have similar structure (they need not be isomorphic) as long as doing that does not negatively affect the prediction task. And we showcase ISDEA+ also has such capabilities.
>
> **Comment 2. The statement of “node prediction task is relatively easy” might be invalid. It can depend on the negative sample size. The current negative sample size of 50 is not enough to showcase the statement.**
>
> **A2:** Thank you for raising this important point! That is true. We will qualify our point, which is essentially similar to Jambor et al., EACL 2021, where a model's success in node prediction tasks may not fully reflect its capabilities in predicting the type of link, but simply that of being able to predict homogeneous links (in a way agnostic to its type). For instance, RMPI and NBFNet have good performance at predicting nodes but significantly worse performance at predicting edge types.
>
> In our rebuttal, we introduced an improved version of ISDEA (old), named ISDEA+. This enhancement allows us to handle larger negative samples during evaluation. Here are the results we have managed to gather in the limited time available. In particular, we rerun the node prediction task on all tasks of PediaTypes with 500 random negative node samples, and collected the Hits@10 results in the following table:
>
>
> | Models     | EN-FR          | FR-EN          | EN-DE          | DE-EN          | DB-WD           | WD-DB          | DB-YG          | YG-DB           |
> |------------|----------------|----------------|----------------|----------------|-----------------|----------------|----------------|-----------------|
> | NBFNet     | $65.73\pm3.17$ | $73.36\pm1.84$ | $58.93\pm2.89$ | $32.15\pm3.14$ | $33.75\pm2.87$  | $61.64\pm1.36$ | $34.07\pm1.49$ | $51.00\pm1.90$ |
> | InGram     | $67.42\pm2.42$ | $48.76\pm8.10$ | $46.84\pm4.07$ | $54.08\pm1.35$ | $24.87\pm10.38$ | $49.41\pm5.41$ | $15.69\pm4.11$ | $23.73\pm7.12$  |
> | DEq-InGram | $76.33\pm1.27$ | $64.84\pm6.17$ | $59.67\pm2.43$ | $66.30\pm2.42$ | $35.70\pm12.10$ | $65.55\pm4.04$ | $24.11\pm4.73$ | $37.76\pm12.01$ |
> | ISDEA+     | $68.27\pm3.62$ | $50.76\pm2.57$ | $66.83\pm0.48$ | $53.52\pm0.67$ | $56.97\pm0.85$  | $65.75\pm0.78$ | $20.72\pm0.35$ | $26.67\pm1.76$  |
>
>
> From these results, we can observe that even with an enlarged number of negative samples, we still observe the same model behaviors as negative sample size 50. The NBFNet model is still able to achieve relatively good performance in the node prediction task while its model architecture is not able to capture the relationship among the new relation types. Both of our proposed models, ISDEA+ (and DEq-InGram), still achieve better performance than the baseline NBFNet and InGram in most scenarios.
>
> **Comment 3. What are the input features to standard GNN architectures reported in the experiments under GraphConv / GAT / GIN?**
>
> **A3:** Thank you for this important question. We discussed this at the bottom of page 5: since our graphs in PediaTypes and WikiTopics do not have node features, we initialize them as all-one features to feed into all model architectures.

---

> > ### Author Response · Authors · 2023-11-19
> >
> > **Comment 4. Will average NBFNet results across several runs with random relation initialization be considered as distributionally double equivariant?**
> >
> > **A4:** The short answer is yes, but the model would essentially only be able to predict that a relation exists between u and v, not the type of relation. Our results with NBFNet show essentially that. The averaged NBFNet can predict a link exists (node prediction) but can't predict the relation type.
> >
> > **Comment 5. How does achieving double equivariant representations through randomized relation embedding initialization connect to frame averaging and data augmentation?**
> >
> > **A5:** Thank you for this insightful question! Indeed, frame averaging (Puny et al., ICLR 2022) or the Reynold’s operator (Murphy et al., ICML 2019) would be an expensive way to achieve double-equivariant representations for any permutation sensitive function $f$. This is mathematically represented as $\bar{f} = \frac{1}{N!R!}\sum_{\phi\in \mathbb{S}_N, \tau\in \mathbb{S}_R} \tau^{-1}\circ\phi^{-1}\circ f(\phi\circ\tau\circ \mathbf{A})$. Such an approach is most-expressive if $f$ is most-expressive. However, the computational cost becomes prohibitively high for large graphs (unless one can find a small frame). Data augmentation can be performed via a modification of pi-SGD (Murphy et al., ICML 2019) where we would need to augment with two permutation groups (one for nodes, another for relations). The final method would be, however, only approximately double equivariant at best. We imagine the cost of training would be quite expensive due to the large number of data augmentation transformations needed. It is, however, an intriguing research direction that could be worth exploring in future work. Thank you!
> >
> > # Summary
> >
> > We want to thank the reviewer for their support, their time, and the insightful suggestions! We hope we have addressed your questions adequately. Your questions about the different number of relations in train and test inspired us to perform an interesting meta-learning task, where pre-training on an increasing number of KGs of distinct domains (e.g., Sports, Arts, Corporations) improves the zero-shot performance of the model on a different test KG domain (e.g. Health) (with new relationship types) to the point ISDEA+ performance is often comparable to SOTA models trained transductively on a KG domain (e.g., Health) and tested on the same training KG over held-out edges. Prior to this, our benchmark considered only training on one domain and testing on another domain. We wanted to show that training on more (and diverse) domains improves ISDEA+ pre-trained zero-shot performance on new KG of a different domain (with new relation types not seen in training).
> >
> > **References**
> >
> > [1] Jambor et al. "Exploring the limits of few-shot link prediction in knowledge graphs." EACL 2021
> >
> > [2] Puny et al. "Frame Averaging for Invariant and Equivariant Network Design." ICLR, 2022
> >
> > [3] Murphy et al. "Relational Pooling for Graph Representations." ICML 2019

---

### Official Review · Reviewer_Uo9n · 2023-10-31

**Soundness:** 2 fair
**Presentation:** 1 poor
**Contribution:** 2 fair
**Rating:** 3
**Confidence:** 4

**Summary:**

This paper introduces the task of  "doubly inductive link prediction'", where the objective is to be able to make inductive prediction on both novel nodes and novel relation types, which are not encountered during training. This is a highly challenging task, especially because the authors do not allow the use of any additional context regarding the unknown relations.   The authors propose a general framework ISDEA to generate "double permutation-equivariant" representations and further explore ways to augment the existing InGram architecture with "distributionally double equivariant positional embeddings". Two new real-world datasets are proposed for benchmarking "doubly inductive link prediction" and experiments are carried out to validate the theoretical findings.

**Strengths:**

- **Problem and setup**: Inductive link prediction is a very important task and authors generalise this task to also predict novel relation types. The paper provides an approach for modeling equivariant representations of nodes and relations.
- **Motivation and study**: A clear motivation, including the study of different architectures.
- **Benchmarking**: New benchmarking datasets are introduced and assessed against prior methods, establishing a new context.

**Weaknesses:**

- **Presentation and formal writing**: The writing of the paper is problematic and concepts are often unclear:
  - The text is very repetitive and contains many redundancies (i.e., the contribution of the paper is highlighted three times in the first page with paraphrased sentences), but when it comes to formal definitions, it does not make a rigorous treatment (see below).
  - Figure 1: This is crowded and does not explain much to me: why are the relations typed using conjunctions at this point? How does the logical description given in the beginning of page 3 in any way correspond to this figure?
  - Multigraph: Authors seem to suggest a multigraph is more general than a knowledge graph. It is unclear to me what authors specifically mean by this? If they mean a directed, multi-relational graph then this is nothing more than a knowledge graph. Heterogenous networks are special instances with single relation types allowed between nodes etc.
  - Doubly inductive: The naming is somewhat problematic, because the inductive prediction is either on the relation or on one of the entities at a time, but not both according to Def 1.
   - Isomorphic triplets: The definition of multigraph isomorphism and triplet isomorphism is a very odd one. I have no idea why, e.g., (Hans, Grand $\land$ Father, Bob) in train and (Hanna, Granny $\land$ Mother) should be considered isomorphic (and at this point we still do not know the role of logical conjunction in defining the relations). This is essential because everything builds on this notion of "isomorphism" which is completely unjustified.
  - The paper is very hard to parse in general: in many cases, the statements of the results appear ambiguous to me, including the ones in the appendix.

- **New architectures**: The new architectures introduced in the paper appear to be somewhat incremental. IDSEA is a variant of DSS-GNN operating on relation-induced subgraphs, whereas DEq-InGram is a simple modification of InGram with bagging.

- **Empirical findings**: IDSEA seems to perform consistently worse than DEq-InGram in the task node prediction on PediaTypes, which does not seem to match what the theory suggests and is not being discussed in the paper.

- **Train and test distribution**: The paper predominantly focuses on scenarios where the train and test graphs share a similar distribution. However, there exists a range of tasks involving unseen nodes and relations where the distribution significantly differs between the training and testing phases.  Further experimental validation on these tasks is required.

**Questions:**

Please refer to my review for clarifications and some more questions  here:

- In the experiments, why do the authors not compare with standard relational GNNs such as RGCN, CompGCN, NBFNets, etc?

- What are the differences between ISDEA, DEq-InGram, and InGram in terms of their runtime?

- Since both DEq-InGram and InGram produce distributionally double equivariant representations, why is there a substantial performance gap between these models on both datasets?

---

> ### Author Response · Authors · 2023-11-19
>
> ## Official Response to Reviewer Uo9n
> We express our gratitude to reviewer Uo9n. Your feedback is greatly appreciated. Please refer to the Common Reviewer Response for the most important revisions in our submission. We believe there was a misunderstanding due to an unintended interpretation of Figure 1, which cascaded throughout the paper making it hard to understand. Our revised submission changes Figure 1 to hopefully clear that source of misunderstanding (we also modified the text to further clarify our work). We have also made architectural changes to ISDEA (now called ISDEA+ in the revision) to make it significantly more scalable (e.g., from between 20x to 120x faster than the original ISDEA in our experiments). Please let us know if the revised version addresses the reviewer's concerns. Below we consider all questions point-by-point:
>
> **Comment 1. Writing. Introduction.**
>
> **A1:** Thank you for the feedback. We have revised the introduction to make it less repetitive.
>
> **Comment 2. Figure 1 is hard to understand, what does the relation conjunction mean and how does it correspond to the logical induction statements in the top of page 3?**
>
> **A2:** Great feedback! Thank you! In the revised version we have changed Figure 1 to help clarify the task. Now it shows the graph being trained solely on the Sports Wikipedia KG and tested on the Corporations Wikipedia KG. Note that the relations observed on the domain Sports are all different from the relations on the domain Corporations. This pre-trained zero-shot task better illustrates our task. We also point the reviewer to the updated benchmark task in Section 5.2, where we describe our newly proposed pre-trained zero-shot meta-learning task. We relegated the logical connections to Appendix B, where we discuss the connection between ISDEA+ and universally quantified Horn clauses. We removed the connection to UQER formulas from the main paper to improve clarity.
>
> **Comment 3. Why claim multigraphs are more general than knowledge graphs?**
>
> **A3:** In our revised submission we describe the graphs simply as knowledge graphs and wrote a note that the approach could be adapted to multilayer networks and similar types of graphs.
>
> **Comment 4. The naming of “Doubly inductive task” is problematic since in test the prediction is either on one of the relations or one of the entities.**
>
> **A4:** Thank you for the feedback. We believe there is a misunderstanding of the term "doubly inductive link prediction task" (Definition 2.1). The inductive setting of our task aims to predict missing links in test graphs with both completely new nodes and completely new relation types, as seen in our PediaType and WikiTopic datasets (and illustrated in Figure 1). We have debated whether the term *fully inductive* would be more adequate, but we can see that also leading to even further potential misunderstandings. The task is expressed mathematically that the set of training and test nodes are disjoint, $\mathcal{V}^{(te)} \cap \mathcal{V}^{(tr)} = \emptyset$, and that the set of training and test relation types are disjoint $\mathcal{R}^{(te)} \cap \mathcal{R}^{(tr)} = \emptyset$. Perhaps the misunderstanding originates from our last sentence in Definition 2.1, “we aim to predict both missing relations for the given head and tail nodes $(i, ?, j)$ and missing nodes for a given relation $(i, k, ?)$.", which we now removed in the revision to avoid this misunderstanding.
>
> **Comment 5. The definition of multigraph isomorphism and triplet isomorphism is unclear, why (Hans, Grand ∧ Father, Bob) in train and (Hanna, Granny ∧ Mother) should be considered isomorphic? And what does the conjunction between relation types mean?**
>
> **A5:** Thank you for pointing out this source of confusion! We have now changed Figure 1 to avoid this confusion in the revised submission. Figure 1 now describes a zero-shot task between training on the Sports wikipedia KG (KG restricted to the domain Sports) and inductively predicting links (zero-shot, no retraining, no side-information) on the Corporations wikipedia KG (KG restricted to the domain Corporations). The definition of isomorphism in the context of our theory involves a bijective mapping between the nodes and relation types of two KGs, such that the same structural relationships are preserved. Figure 1 illustrates a Sports KG and a Corporations KG that do not share relation types. In our datasets, training and test KGs even have different numbers of relation types. A double equivariant model like ISDEA+ automatically deals with distinct relation types and distinct numbers of relation types between train and test.

---

> ### Author Response · Authors · 2023-11-19
>
> **Comment 6. The new architectures introduced in the paper are somewhat incremental, ISDEA is a variant of DSS-GNN operating on relation-induced subgraphs, whereas DEq-InGram is a simple modification of InGram with bagging.**
>
> **A6:** Thank you for your feedback. It seems that there is a misunderstanding. The original DSS-GNN is not double equivariant, since it is a method for creating representations for bags of subgraphs over a single group equivariance (node permutations). Moreover, we have now made the method ISDEA+ between 20x and 120x faster, which took significant effort. We also show that double equivariance is a key property for the task of zero-shot link prediction without side information. Inspired by one of Reviewer’s VGiN questions, we added new experimental results to the main paper, where we expanded our proposed benchmark to better explain the power of double equivariant models in a self-supervised pre-trained zero-shot meta-learning task. Please refer to ''Common Reviewer Response'' and for Section 5.2 and Appendix F.1.3 in the revised submission.
>
> There is also a small misunderstanding about DEq-InGram: We perform Monte Carlo averaging, not bootstrapping aggregation (the averaging is not over bootstrapping). This averaging provably makes the model double equivariant, which significantly improves its performance. The new results show that ISDEA+ is still significantly more consistent and better than even DEq-InGram. We have made this point more salient in the text.
>
> **Comment 7. Why does ISDEA perform consistently worse than DEq-InGram in the task node prediction on PediaTypes? Does it contradict the theory?**
>
> **A7:** Please see new results of ISDEA+ in Section 5.1 of the revised manuscript, which now outperforms DEq-InGram in nearly all scenarios. The improvement in performance of DEq-InGram over InGram confirms our theory that double equivariance allows neural networks to perform zero-shot double inductive link prediction tasks.
>
> PS: The node prediction performance aligns with findings from Jambor et al., EACL 2021, where a model's success in node prediction tasks may not fully reflect its capabilities in predicting the type of link, but simply that of being able to predict homogeneous links (in a way agnostic to its type). For instance, RMPI and NBFNet have good performance at predicting nodes but significantly worse performance at predicting edge types.
>
> **Comment 8. In experiments, the training graphs and test graphs share similar distributions. How will it perform if it is not the case?**
>
> **A8:** Thank you for your question. We are uncertain what the reviewer means by "similar distributions". Assuming the reviewer is referring to the distribution of graphs, the WikiTopics dataset presents a scenario where we intentionally train the model on graphs from one domain and test it on graphs from a completely different domain, such as training on Art and testing on Sports. Arts and Sports not only have nearly no common relation types, they also have a different number of relation types. Hence, their distribution should be different. Please let us know if this answers your concerns.
>
> **Comment 9. Why not compare with standard relational GNNs such as RGCN, CompGCN, NBFNet, etc?**
>
> **A9:** We thank the reviewer for pointing out these works, but again there seems to be a misunderstanding that we hope will be cleared with the new Figure 1 in the revised submission. Please note that RGCN, CompGCN and NBFNet cannot perform the task where they are trained on, say, Arts domains, and as asked to zero-shot predict links on, say, Sport domains. Please refer to the second paragraph in Section 4 and Appendix E for the detailed distinction between the GNN-based inductive approaches and our method. We have added CompGCN in our related work section in the appendix.
>
> The doubly inductive link prediction task (Definition 2.1) involves both entirely new nodes and new relation types in the test graph. In contrast, the mentioned papers concentrate on different tasks. Specifically, RGCN and CompGCN were initially designed for transductive tasks, while NBFNet is designed for inductive link prediction for only new nodes. Following InGram, we modified the SOTA relational GNN model NBFNet with minimal changes to handle unseen relation types at test time, whose results are shown in Table 1 and Appendix F.1.2. The results demonstrate that our proposed method significantly outperforms the adaptation of NBFNet used by InGram, highlighting the unique challenges and efficiency of our approach in this task setting.

---

> ### Author Response · Authors · 2023-11-19
>
> **Comment 10. What are the differences between ISDEA, DEq-InGram, and InGram in terms of their runtime?**
>
> **A10:** Here we present a run time comparison for ISDEA+, ISDEA, DEq-InGram, InGram and NBFNet on the node prediction task on EN-FR in our PediaTypes dataset, and a much larger transductive FB15K-237 dataset. Due to varied mini-batch training sizes for different models, we report the training time per minibatch step (i.e., per gradient update step) for the training time comparison. Here are the results:
>
> **Training and inference on the EN-FR in PediaTypes**
>
> -   4962 entities, 122 relations, 30876 train triplets, 3326 validation triplets, 2485 test triplets
>
> | Model | Training Time per Minibatch Step | Test Inference Time | Max GPU Memory Usage |
> |------------|----------------------------------|---------------------|----------------------|
> | ISDEA+ | 0.05s | 6.53s | 1.9GB |
> | ISDEA(old) | 2.56s | 359.45s | 9.8GB |
> | Deq-InGram | 0.09s | 10.39s | 3.0GB |
> | InGram | 0.09s | 5.60s | 3.0GB |
> | NBFNet | 0.02s | 5.14s | 5.5GB |
>
> **Training and inference on the transductive FB15k-237 dataset**
>
> -   14541 entities, 237 relations, 272115 train triplets, 17535 validation triplets, 20466 test triplets
>
> | Model | Training Time per Minibatch Step | Test Inference Time | Max GPU Memory Usage |
> |------------|-----------------------------------|---------------------|----------------------|
> | ISDEA+ | 0.08s | 27.60s | 5GB |
> | ISDEA(old) | N/A | N/A | OOM |
> | Deq-InGram | 0.57s | 99.67s | 13GB |
> | InGram | 0.57s | 62.60s | 13GB |
> | NBFNet | 0.07s | 62.23s | 14GB |
>
> We can see that ISDEA+ significantly improves ISDEA, and is among the fastest model with least memory consumption.
>
> **Comment 11. Since both DEq-InGram and InGram produce distributionally double equivariant representations, why is there a substantial performance gap between these models on both datasets?**
>
> **A11:** There appears to be a misunderstanding. InGram is a distributionally double equivariant representation (as per Lemma 3.2) and is sensitive to permutations. In contrast, DEq-InGram, is an (approximate) double-equivariant representation constructed based on Theorem 2.8, which is designed to be (approximately) permutation invariant. This difference in handling permutations contributes to the substantial performance gap between the two models. These results underscore the significance of double equivariant representations in doubly inductive link prediction tasks. Still, the new ISDEA+ consistently outperforms InGram and DEq-InGram in the revised submission.
>
>
> # Summary
>
> We want to thank the reviewer for their time and insightful suggestions! We hope our answers can address your concerns well.
>
> We also prudently ask you to reconsider our work if the concerns are addressed. With the new zero-shot meta-learning results and the significant performance boost of our new method ISDEA+, we hope the reviewer will reconsider their score.
>
> Overall, we believe our work makes important contributions in inductive learning on knowledge graphs, and proposes a new paradigm and novel architecture ISDEA+; we would appreciate your reconsideration on this point. Thank you for your efforts again!
>
> **References**
>
> [1] Jambor et al. "Exploring the limits of few-shot link prediction in knowledge graphs." EACL 2021

---

### Official Review · Reviewer_ZXyk · 2023-11-01

**Soundness:** 4 excellent
**Presentation:** 3 good
**Contribution:** 2 fair
**Rating:** 5
**Confidence:** 4

**Summary:**

This paper aims to address the so-called doubly inductive link prediction task, where both new nodes and new relation types can be found solely in test time. To this end, author proposes two different models, ISDEA and DEq-InGram, which all abides by the equivariance requirement. Finally, experiment results show the new method beats baseline empirically.

**Strengths:**

The result of the paper seems sound, the author provides the reader with many theorems and proofs for its theory and they seem plausible to me.

The experiments are good, including many baselines, and the empirical result shows that the new method is in general better (though it falls behind the baseline in some settings).

**Weaknesses:**

The design of ISDEA is very straightforward, however, it is purely brutal force and has very high complexity. I have checked the statistics of the dataset used for experiment evaluation and found these two newly crafted datasets are significantly smaller than commonly used datasets, like FB15k or even its subset FB15k-237. I believe one major motivation for the setting for inductive learning is to allow for scalability towards a larger knowledge graph, yet the model design seems to be in the opposite direction.

**Questions:**

What is the largest knowledge graph that can be computed by ISDEA, for example, with GPU memory of 32 GB?


Can the isomorphism requirement be reduced to some WL test to reduce the complexity yet maintain decent empirical results?

---

> ### Author Response · Authors · 2023-11-19
>
> ## Official Response to Reviewer ZXyk
>
> We appreciate your efforts and insightful comments! We would like to extend our sincere thanks to reviewer ZXyk for acknowledging the theoretical framework and comprehensive empirical results presented in our work. Your insightful inquiries and constructive feedback are immensely valuable to us. Please refer to the Common Reviewer Response for the most important revisions in our submission. To address your concerns, we provide point-by-point responses below.
>
> **Comment 1. The design of ISDEA is purely brute force and has very high complexity.**
>
> **A1:** Please see the new architecture of ISDEA (denoted ISDEA+) in our revised submission. The primary source of computation complexity in the old ISDEA is the process of enumerating across all $R$ relations and computing distances between every pair of nodes. To address this, as discussed in the Common Reviewer Response, we have developed ISDEA+, which improves the time complexity by $R$ over ISDEA (where $R$ is the number of relation types). In our experiments, we observe ISDEA+ is between 20x to 120x faster than ISDEA (old) .
>
> **Comment 2. The newly crafted datasets are significantly smaller than commonly used datasets, such as FB15K and its subset FB15K-237.**
>
> **A2:** Thank you for your question! It seems there's a bit of confusion regarding the size of the datasets. Our datasets, PediaTypes and WikiTopics, were created by selectively sampling from the OpenEA library (Sun et al., VLDB 2020) and WikiData-5M (Wang et al., TACL 2021). This was done specifically to test the model's proficiency in doubly inductive link prediction. While the transductive knowledge graph completion dataset FB15K-237 has a greater number of triplets compared to our datasets, the inductive versions of commonly used datasets like FB15K-237, WN18RR, and NELL995, which are employed in models like GraIL (refer to Table 13 in Teru et al., ICML 2020) and NBFNet (see Table 11 in Zhu et al., NeurIPS 2021), are generally smaller than our graphs. For instance, as illustrated in Appendix F.1.3, each graph in our WikiTopics dataset comprises 10,000 nodes, with a minimum of 12,516 and a maximum of 80,269 training triplets. In contrast, the inductive version of FB15K-237 used in NBFNet (Table 11 in Zhu et al., NeurIPS 2021) contains only between 5,226 and 33,916 training triplets, which is significantly fewer than our datasets.
>
> In the revised version, we have added new zero-shot meta-learning experiments over larger graphs formed by combining up to three domains in training. These new zero-shot meta-learning experiments amount to up to training graphs with ~150,000 triplets.
>
> **Comment 3.  What is the largest knowledge graph that can be computed by ISDEA, for example, with GPU memory of 32 GB?**
>
> **A3:** Thank you for the great questions! We use the commonly-used transductive FB15K-237 KG to test the memory consumptions for our model along with run time comparison. Here is the result:
>
> **Training and inference on the transductive FB15k-237 dataset**
> - 14541 entities, 237 relations, 272115 train triplets, 17535 validation triplets, 20466 test triplets
>
> | Model      | Training Time per Minibatch Step  | Test Inference Time | Max GPU Memory Usage |
> |------------|-----------------------------------|---------------------|----------------------|
> | ISDEA+     | 0.08s                             | 27.60s              | 5GB                  |
> | ISDEA      | N/A                               | N/A                 | OOM                  |
> | Deq-InGram | 0.57s                             | 99.67s              | 13GB                 |
> | InGram     | 0.57s                             | 62.60s              | 13GB                 |
> | NBFNet     | 0.07s                             | 62.23s              | 14GB                 |
>
> Indeed, we have found that the original implementation of ISDEA (old) encountered an out of memory (OOM)  issue when training on this graph, while the new architecture of ISDEA+ can be trained with only 5GB max GPU memory usage, significantly better than InGram, DEq-InGram, and NBFNet. Due to time constraints, we did not test on larger graphs, but the model should be able to manage a much larger dataset on a 32GB GPU. Overall, we find that ISDEA+ is among the fastest models with least memory consumption.

---

> > ### Author Response · Authors · 2023-11-19
> >
> > **Comment 4. Can the isomorphism requirement be reduced to some WL test to reduce the complexity?**
> >
> > **A4:** That is an interesting suggestion! Thankfully, we were able to significantly speed up ISDEA+ (up to 120x faster). Addressing your question, we believe a WL test feature could hurt performance. In standard graph tasks, a GNN tends to make similar embeddings for nodes that have similar structure (they need not be isomorphic) as long as doing that does not negatively affect the prediction task. WL-tests output very distinct colors to nodes that are not isomorphic (due to the use of hash functions), even if they are structurally very similar. Moreover, there are currently no WL-type tests for double equivariant graphs.
> >
> > To this end, we propose ISDEA+ as an architecture to achieve double invariant triplet representations and double equivariant graph representations. One of the properties of the double invariant triplet representation is that it will output the same representation for isomorphic triplets (Definition 2.4). However, the model can also be applied on test graphs that are non-isomorphic to the training graphs, as shown in Section 5.
> >
> > # Summary
> >
> > We want to thank the reviewer for their time and insightful suggestions! We hope our answers can address your concerns well.
> >
> > We also prudently ask you to reconsider our work if the concerns are addressed. With the new zero-shot meta-learning results and the significant performance boost of our new method ISDEA+, we hope the reviewer will reconsider their score.
> >
> > Overall, we believe our work makes important contributions in inductive learning on knowledge graphs, and proposes a new paradigm and novel architecture ISDEA+; we would appreciate your reconsideration on this point. Thank you for your efforts again!
> >
> > **References**
> >
> > [1] Sun et al. "A benchmarking study of embedding-based entity alignment for knowledge graphs." VLDB 2020
> >
> > [2] Wang et al. "A unified model for knowledge embedding and pre-trained language representation." TACL 2021
> >
> > [3] Teru et al. "Inductive Relation Prediction by Subgraph Reasoning." ICML 2020
> >
> > [4] Zhu, et al. "Neural bellman-ford networks: A general graph neural network framework for link prediction." NeurIPS 2021

---

### Author Response · Authors · 2023-11-19
**Common Reviewer Response**

# Common Reviewer Response


Dear reviewers,

We are excited to share these updates. We sincerely appreciate the time and effort you have invested in reviewing our paper, and the great feedback we received. We have dedicated ourselves to improve the work based on your feedback and update our submission with changes marked in blue; it was an incredible amount of work but we are excited to share with you some significant enhancements that we believe address several of the concerns raised:

## ISDEA+ (20x to 120x faster than ISDEA in our experiments).

In order to address the scalability concerns, we have now developed an improved version of our ISDEA model (named ISDEA+ in the revised version). ISDEA+ improves the computational complexity by $R$ over ISDEA (where $R$ is the number of relation types). ISDEA+ has time complexity $O(L(Nd^2+|E|d))$, where $L$ is number of layers, $R$ is the number of relations, $N$ is the number of nodes, $|E|$ is the number of observed triplets in the graph, $d$ is the maximum size of hidden layers. While ISDEA's complexity was $O(LR(Nd^2+|E|d))$. This means ISDEA+ is between 20x to 120x faster than ISDEA in our experiments. ISDEA+ is also significantly faster at inference time than InGram, while consuming less than half of InGrams's memory (e.g., 5GB for ISDEA+ vs 13GB for InGram on transductive FB15k-237 dataset). The key to this development lies in ISDEA+'s improved message-passing scheme, which updates node representations independently for each relation type. Thus, we can design a training process where each gradient step considers only one relation vs all others. And ISDEA+ still produces double equivariant representations. Moreover, ISDEA+ not only accelerates the model but also significantly improves its performance in both node prediction and relation prediction tasks (see the updated results in Section 5 of our revised submission). We describe the changes in ISDEA+ in Section 3.1 and Appendix C. We believe these improvements address most of the raised concerns about performance and scalability w.r.t. ISDEA.


## New pre-trained zero-shot meta-learning task: Showcasing the power of double equivariant models.

Inspired by a comment of reviewer VGiN, we restructured one of our benchmarks to better showcase the power of double equivariant models: This benchmark is a pre-trained zero-shot meta-learning task, where the model is initially trained on a (diverse set of) KG(s) from distinct domains (e.g., Education, Health, Sports). The essence of this training is to imbue the model with the ability to abstract and generalize across domains. In test, the model is evaluated zero-shot on a KG of a completely new domain (e.g. Health). This setup challenges the model to apply its learned meta-knowledge to a new, unseen domain, demonstrating its capacity for cross-domain generalization and adaptation without prior exposure to the specific domain of the test KG. Our experiments show that ISDEA+ zero-shot performance increases with the number of self-supervised training KGs on diverse domains, to the point that it is sometimes better than transductive training on the test KG domain itself (e.g., the state-of-the-art baseline trains on Taxonomy KG to predict held out links on Taxonomy KG itself). These new results are detailed in Section 5.2 and Appendix F.1.3.

---

### Author Response · Authors · 2023-11-22
**Comment to AC and SAC**

Dear AC and SAC,

In our rebuttal and submission revision, we believe we have meticulously addressed all concerns raised. This was a significant effort (as the AC and SAC can attest from our answers). Please let us know if the reviewers have further questions, we would be happy to answer them.

PS: We understand the last day of reviewer-author interactions come close to a major US holiday. Hence, we address the reviewers, AC and SAC. Our sincere appreciation extends to all reviewers, the AC, and SAC for your dedication and invaluable contributions to the ICLR community during this period.

Thank you,
Paper 8054 Authors

---

### Meta-Review · Area_Chair_8DBv · 2023-12-06

**Metareview:**

The paper addresses doubly inductive link prediction, where both nodes and relation types can be novel.

Reviewers acknowledge that this is a challenging task and the proposed method does have merits.

However, the originally submitted version has significant flaws and the revision is so extensive that most reviewers as well as myself don’t have much confidence in the paper, and it's unreasonable to ask reviewers to re-do the review again for the entire paper. As a result, I believe another round of review is needed and I encourage the authors to ensure the quality of their paper in the first place.

**Justification For Why Not Higher Score:**

The originally submitted version has significant flaws and the revision is so extensive that most reviewers as well myself don’t have much confidence in the paper, and it's unreasonable to ask reviewers to re-do the review again for the entire paper. As a result, I believe another round of review is needed and I encourage the authors to ensure the quality of their paper in the first place.

**Justification For Why Not Lower Score:**

N/A

---

### Decision · Program_Chairs · 2024-01-16

Reject